# Brachiopod genome unveils the evolution of BMP signalling in bilaterian body patterning

Thomas D. Lewin[1], Tosuke Sakagami[1], Keisuke Shimizu[2], Li-Jung Kao[1], Yi-Ling Chiu[1], Isabel Jiah-Yih Liao[1], Mu-En Chen[1], Kanako Hisata[3], Kazuyoshi Endo[4], Noriyuki Satoh [3], Peter W. H. Holland [5], Yue Him Wong[6] ✉ & Yi-Jyun Luo [1] ✉

The molecular control and ancestral state of dorsal–ventral patterning within spiralians remain unclear due to the remarkable diversity across species. Here we present a chromosome-level genome for the brachiopod *Lingula anatina* and apply functional transcriptomics to study dorsal–ventral patterning under BMP signalling control. We uncover asymmetrical activation of BMP signalling at the dorsal side of the gastrula, governed by ventral *chordin* expression and a 'seesaw' of BMP ligands. Using small-molecule drugs and recombinant proteins, we show that high BMP activity inhibits genes typically associated with neural patterning during gastrula and larval stages, similar to deuterostomes and non-spiralian protostomes. Our findings suggest deep conservation of this mechanism across all three major bilaterian clades, supported by striking similarities in BMP-regulated gene sets between brachiopods and *Xenopus*. We argue that the spiralian ancestor retained the ancestral bilaterian mechanism, although downstream network components have undergone developmental system drift.

The integration of bone morphogenetic protein (BMP) signalling into the patterning of the dorsal–ventral axis is a cornerstone of bilaterian embryonic development. The last common ancestor of bilaterians likely established this axis through an interplay between BMP signalling molecules and their antagonists, such as chordin[1–3]. In chordates, this interaction creates a BMP gradient, dividing the ectoderm into two distinct domains, a dorsal neural domain (low BMP) and a ventral epidermal domain (high BMP), in a process known as neural induction[4,5]. In the classical vertebrate model *Xenopus*, this gradient is regulated by the Spemann–Mangold organiser, a signalling centre that produces a cocktail of BMP antagonists, including chordin[4,6,7] and noggin[8,9]. In the fruit fly *Drosophila* and other invertebrates, the BMP gradient is completely inverted, with dorsal Dpp/Bmp and ventral Sog/chordin, but neural domain formation is still induced at the low BMP end of the gradient[10–13]. This reversed pattern is considered to represent the bilaterian ancestral state shared by protostomes and ambulacrarian deuterostomes[14–16]–echinoderms and hemichordates– with a 180 degree axis inversion occurring in chordates[15,17].

The action of several BMP ligands including Bmp2/4, Bmp5–8 and Admp (anti-dorsalizing morphogenetic protein), combines to regulate the process of dorsal–ventral patterning. These proteins function as homodimers but more potently as heterodimers[18–22], binding to type I and type II serine/threonine kinase receptors and resulting in the phosphorylation of the receptor-regulated Smad1/5[23–26]. Nuclear-localised phosphorylated Smad1/5 then binds with the common mediator Smad4 to act as a transcription factor, executing a transcriptional response to the BMP signal[27,28]. Notably, it is the expression domain of the antagonist chordin, rather than those of the BMP ligands themselves, that is considered to be the primary determinant of the signalling gradient[16,29]. By binding BMP ligands and preventing their interaction with cell surface receptors, chordin effectively attenuates BMP signalling[29–32]. While *chordin* is expressed at the low end of the

[1]Biodiversity Research Center, Academia Sinica, Taipei, Taiwan. [2]Research Institute for Global Change, Japan Agency for Marine Science and Technology, Kanagawa, Japan. [3]Marine Genomics Unit, Okinawa Institute of Science and Technology Graduate University, Okinawa, Japan. [4]Department of Earth and Planetary Science, Graduate School of Sciences, The University of Tokyo, Tokyo, Japan. [5]Department of Biology, University of Oxford, Oxford, UK. [6]Institute for Advanced Study, Shenzhen University, Shenzhen, China. ✉e-mail: timwong@szu.edu.cn; yjluo@as.edu.tw

BMP gradient, BMP ligand distribution varies significantly by taxon[16], with *bmp2/4* typically expressed at the opposite end to *chordin*[3]. In *Drosophila*, where BMP signalling is maximal dorsally, *dpp*, the ortholog of *bmp2/4*, is restricted to the dorsal side[33], while *screw*, the ortholog of *bmp5–8*, is expressed ubiquitously[34]. In contrast, several distantly related animals, including *Xenopus*, the sea urchin *Paracentrotus* and even the sea anemone *Nematostella*, exhibit what has been denoted as a BMP 'seesaw', with BMP ligands produced at both the high and low ends of the signalling gradient[35]. In *Xenopus*, where BMP signalling is maximal ventrally, *Bmp4* and *Bmp7* are expressed ventrally but *Bmp2* and *Admp* are dorsal[5,35–37]. In *Paracentrotus*, where BMP signalling is maximal dorsally, *bmp2/4* and *admp1* are coexpressed together with *chordin* on the ventral side, while *admp2* is expressed dorsally[16,38]. In *Nematostella*'s directive axis, which may or may not be homologous to the dorsal–ventral axis of bilaterians[29,39–43], the BMP ligands *dpp* and *bmp5*–8 are also co-expressed with *chordin* at the low BMP signalling end, with another BMP ligand, *gdf5*-like, produced at the opposite end[39–41]. The bimodal production of BMP ligands and associated negative feedback mechanisms, known as the BMP seesaw, is proposed to facilitate the production of a robust and precise gradient via self-regulatory mechanisms[35]. However whether this arrangement was present in the ancestor of bilaterians or has evolved several times independently is currently uncertain[29].

Aside from the dorsal–ventral axis inversion in chordates and the variability in BMP ligand expression, the ancestral BMP-mediated patterning mechanism remains highly conserved across diverse members of two of the three main groups of bilaterians: deuterostomes and ecdysozoans[44,45]. However, within Spiralia, a diverse group of protostomes including brachiopods, molluscs, annelids and platyhelminths that constitutes the third major bilaterian clade, there appears to be no consistent role for BMP signalling in dorsal–ventral axis patterning[46]. Spiralians display diverse patterning systems including region-specific BMP signalling[47–49] and even complete absence of dorsal–ventral BMP signalling[50]. For instance, perturbations to BMP signalling in the annelid *Capitella teleta* did not affect dorsal–ventral axis formation or central nervous system development but resulted in abnormal left–right asymmetries[48,51]. In the mollusc *Crepidula fornicata*, BMP signal perturbations affected neural and epidermal cell fates in the head region but not the trunk[47]. In another annelid, *Chaetopterus pergamentaceus*, activin/nodal signalling but not BMP signalling is the key specifier of the dorsal–ventral axis[50,52]. Indeed, the key BMP antagonist chordin has been lost many times independently within annelids[53] and other spiralians like platyhelminths[54]. In some species, such as the molluscs *Lottia peitaihoensis* (also known as *Lottia goshimai*) and *Ilyanassa obsoleta*, BMP signals involved in dorsal–ventral patterning are present but have a positive effect on neurogenesis[55,56], opposite to that in ecdysozoans and deuterostomes. In the early-branching annelid *Owenia fusiformis*, BMP signals do not affect neurogenesis[57], suggesting potential independent effects of BMP signals on dorsal–ventral fate and neural induction. In light of these experiments, the ancestral function of the BMP signalling pathway in spiralian development remains unclear[51].

In this context, the focus of most spiralian studies on only two phyla, molluscs and annelids, limits our understanding. Broader sampling across spiralian phyla is essential to clarify the ancestral role of BMP signalling within this group and beyond. To this end, we aimed to study dorsal–ventral patterning in brachiopods, a phylum of marine spiralians with calcified dorsal–ventral shells that is relatively understudied compared to other spiralian phyla. Though previous studies have indicated that the brachiopods *Novocrania anomala* and *Terebratalia transversa* may use a BMP–chordin network for dorsal–ventral patterning in a similar manner to ecdysozoans and deuterostomes[58], understanding of this process remains incomplete. As a result, this group of animals makes ideal subjects for comprehensive studies of the BMP–chordin axis to gain broader understanding of evolutionary scenarios regarding the ancestral role of BMP signalling and dorsal–ventral patterning within spiralians.

Here we present a chromosome-level genome for the brachiopod *Lingula anatina*. We then investigate dorsal–ventral patterning during *L. anatina* embryonic development using a combination of small-molecule drugs, recombinant proteins, functional transcriptomics and in situ hybridisation. Specifically, we (i) characterise the BMP pathway gene repertoire; (ii) map the temporal and spatial expression of BMP ligands, antagonists and modulators during embryogenesis; (iii) identify the sites of BMP signal activation by these expression gradients; (iv) elucidate the role of asymmetric BMP signal activation in dorsal–ventral axis patterning; and (v) determine how BMP signalling influences neural domain formation.

## Results

### Chromosome-level assembly of the brachiopod genome

To explore the functions of BMP signalling in the brachiopod *L. anatina*, we first sequenced its genome to chromosome level using PacBio HiFi (405-fold coverage) and Hi-C (171 million paired reads) technologies (Supplementary Fig. 1, Supplementary Data 1 and 2). The resulting assembly is 329.3 Mb in length with a 36.5% GC content and consists of 31.7% repetitive elements (Supplementary Fig. 2, Supplementary Data 3 and 4). There have been recent expansions of repeats, particularly long terminal repeat elements, in the genome of *L. anatina* and other Lophophorata members (Supplementary Fig. 2). The assembly has a scaffold N50 of 30.1 Mb, a substantial improvement over the previous scaffold-level genome with a scaffold N50 of 0.5 Mb[59], and is comparable in quality to other high-quality spiralian genomes[53,60–62] (Fig. 1a and Supplementary Data 5).

The majority of the assembly consists of 10 chromosome-scale scaffolds accounting for 97.8% of the total length, consistent with previous karyotype observations in *L. anatina* ($n = 10$)[63]. Gene prediction and annotation using newly generated RNA sequencing data resulted in highly complete gene models with a BUSCO score of 98.1% (Metazoa *odb10*) (Supplementary Fig. 1 and Supplementary Data 6–10). These gene models comprise 29,458 transcripts corresponding to 24,330 unique protein-coding genes with extensive redundancy reduction achieved compared to the previous assembly (Supplementary Fig. 3 and Supplementary Data 3). Gene family analysis reveals a gene content that is more stable than those of other Lophotrochozoans (Supplementary Fig. 4 and Supplementary Data 11). Notably, we found that longer chromosomes have a higher density of protein-coding genes and a lower density of repeats (Supplementary Fig. 5). Phylogeny construction using exclusively genomic data (Supplementary Data 12) shows brachiopods to be closely related to phoronids and bryozoans, as previously suggested[64–71] (Fig. 1b). This result is robust to the inclusion of transcriptome data for the entoprocts *Barentsia gracilis* and *Loxosomella nordgaardi*[72] and the cycliophoran S*ymbion pandora*[73] (Supplementary Fig. 6).

### BMP pathway components in spiralian genomes

The absence of high-quality genomes has hindered accurate assessment of gene duplications and losses in numerous spiralian groups, especially in understudied lineages such as brachiopods, phoronids, bryozoans and nemerteans. To reconstruct the evolution of the BMP signalling pathway, we annotated BMP pathway components in *L. anatina* and 13 additional spiralian genomes based on reported pathway genes in model systems such as *Xenopus*[5], *Drosophila*[74] and sea urchins[75] and sought to characterise the evolution of BMP components through gene duplication and loss analysis (Fig. 1c, Supplementary Data 13 and 14)[76]. In *L. anatina*, the BMP ligands *bmp2/4*, *bmp3*, *bmp5–8* and *admp* were all present as single copies (Supplementary Fig. 7). Of the serine/threonine kinase receptors that receive BMP signals, *bmpr1* and *bmpr2* were single-copy while there was a partial duplication of

*acvr1* (Supplementary Fig. 8). Interestingly, the BMP signalling mediator *smad1/5* was duplicated, while *smad2/3*, *smad4* and *smad6/7* were single-copy (Supplementary Fig. 9). We also annotated 11 BMP signalling modulators, including *chordin*, *cv2* and *bambi*; of these, all were single-copy in *L. anatina* except *noggin*-like, which was not found (Supplementary Figs. 10–17). Thus, *L. anatina* shows a conserved BMP signalling toolkit, with most genes single-copy, just two duplications, and the loss of *noggin*-like, providing a reference point for comparison with other spiralians.

Characterising the BMP pathway gene sets across, Spiralia we found that the main ligands and modulators associated with dorsal–ventral patterning are highly conserved in the Lophotrochozoa, with *bmp2/4*, *bmp5–8*, *admp* and *chordin* present in all sampled species except specific annelid lineages[53] (Fig. 1c). In contrast, other BMP components show remarkable variation considering their developmental importance. This is epitomised by bryozoans, which possess numerous lineage-specific duplications and losses to many key components at a scale comparable to known divergent groups like

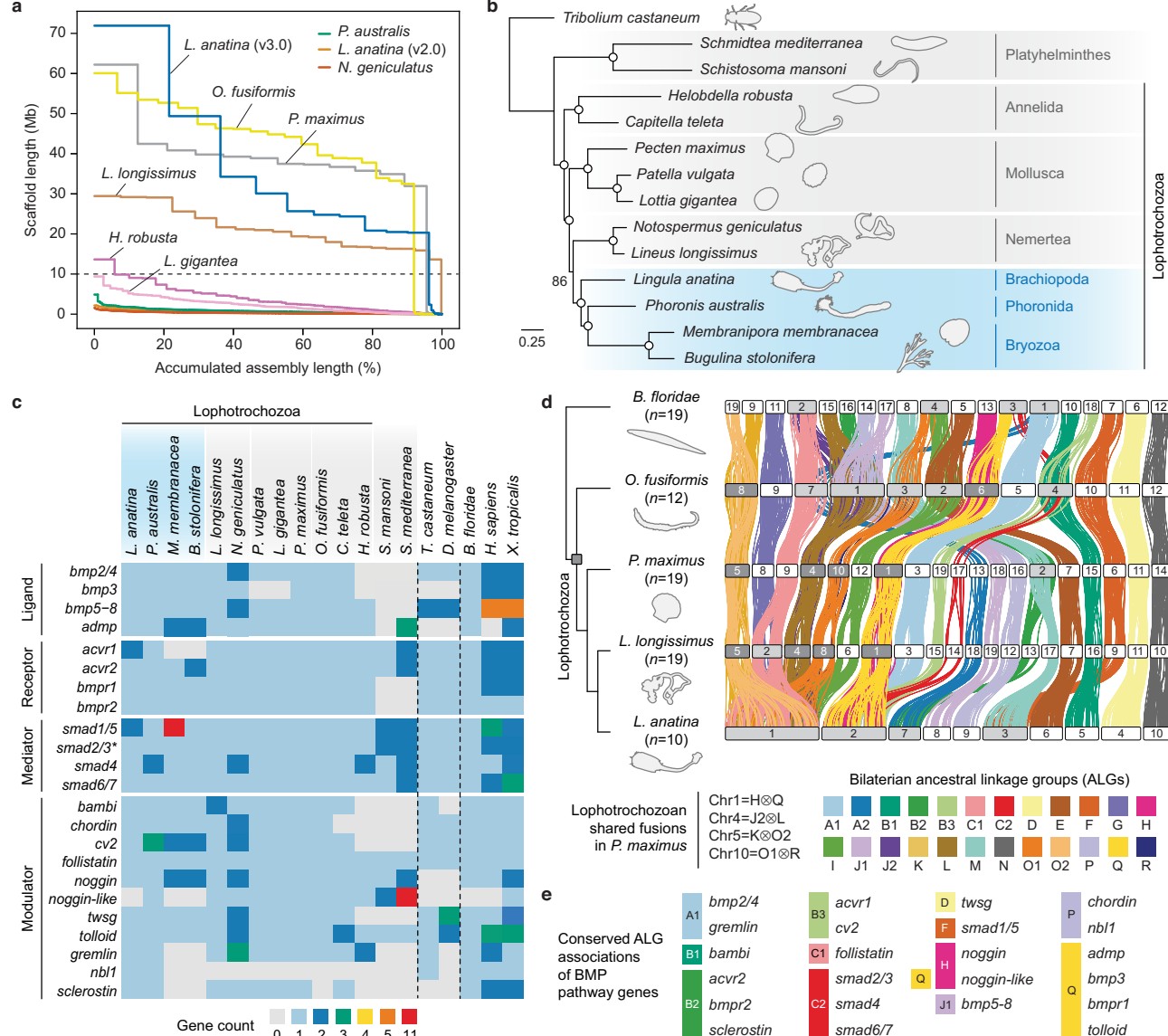

**Fig. 1 | Brachiopod genome reveals BMP signalling pathway evolution.**
**a** Assembly quality comparison across selected spiralian genomes. The *L. anatina* assembly presented in this study (v3.0) comprises 10 chromosome-scale scaffolds. Horizontal dashed line at 10 Mb distinguishes scaffold-level genomes from chromosome-level genomes. **b** Maximum likelihood phylogeny of selected spiralians inferred from 109 single-copy orthologous genes using the LG + F + R5 substitution model and 1000 bootstrap replicates. Open circles denote 100% bootstrap support. The Lophophorata clade, highlighted in blue, comprises Brachiopoda, Phoronida and Bryozoa. **c** Heatmap showing BMP signalling pathway gene repertoires in bilaterians. Vertical dashed lines separate the three main bilaterian clades: spiralians, ecdysozoans and deuterostomes. The asterisk marks *smad2/3*, which is a canonical TGF-β pathway component that can be used by BMPs under specific circumstances. **d** Upper: chromosome-scale gene linkage observed between amphioxus *Branchiostoma floridae*, annelid *Owenia fusiformis*, scallop

*Pecten maximus*, nemertean *Lineus longissimus*, and brachiopod *L. anatina*. Horizontal bars denote chromosomes. Dark grey chromosomes indicate ancestral spiralian fusion events while light grey chromosomes represent lineage-specific fusion events. Vertical lines between chromosomes link the genomic positions of orthologous genes. Lines are colour-coded based on bilaterian ancestral linkage groups (ALGs)[64]. The *L. anatina* genome has undergone extensive rearrangements with lineage-specific fusion events resulting in chromosomes 1, 2, 3 and 7. Notably, the expansive *L. anatina* chromosome 1 equates to six distinct chromosomes in *P. maximus*. Lower: summary of the four lophotrochozoan shared fusions (H⊗Q, J2⊗L, K⊗O2 and O1⊗R) and colour code for bilaterian ALGs. **e** BMP signalling pathway genes are consistently associated with the same ALG (coloured blocks) across the five species shown in (**d**), with the exception of *noggin*-like, which is always found on chromosomes associated with both ALGs H and Q. Source data are provided as a Source data file.

platyhelminths[77]. Two copies each of *admp, cv2* and *noggin* are present and we failed to find *acvr1, noggin*-like and any of the three DAN family genes. We also found 11 copies of *smad1/5* in the bryozoan *Membranipora membranacea*. Dynamic duplications and losses are not limited to bryozoans: the phoronid *Phoronis australis* has two copies of *smad4* and three of *cv2*; the nemertean *Lineus longissimus* has two copies of *bambi*; two mollusc assemblies lack *bmp3* and two annelid assemblies lack *sclerostin*. Interestingly, we identified *nbl1* in *L. anatina* and *P. australis* but no other spiralian assemblies. Numerous gene duplicates are identified in the nemertean *Notospermus geniculatus* but based on BUSCO duplication scores (Supplementary Fig. 1), the use of a low-quality draft assembly, and the lack of duplicates in the nemertean *L. longissimus*, these are likely attributable to false haplotypic duplications[78,79]. This finding highlights the need for high-quality chromosome-level genomes for gene content analyses. Overall, while *L. anatina* maintains a relatively conserved set of BMP pathway-related genes, repertoires exhibit notable evolutionary variation across other spiralians, indicating that BMP gradients may have evolved to accommodate diverse developmental modes and environmental contexts across various spiralian lineages.

## Conserved syntenic associations of BMP pathway genes

We next questioned whether the developmental importance of BMP genes as an ancestral mechanism for body patterning in bilaterians has affected the evolution of their genomic position. To explore this hypothesis, we analysed genome-scale linkage conservation, focusing on the macrosyntenic positions of BMP genes in five bilaterians: the brachiopod *L. anatina*, the mollusc *Pecten maximus*[61], the nemertean *L. longissimus*[62], the annelid *Owenia fusiformis*[53] and the chordate *Branchiostoma floridae*[80]. We first identified the correspondence between chromosomes and bilaterian ancestral linkage groups (ALGs)[81], which are conserved to varying extents in spiralians[82,83], then characterised the events of chromosome fusion and fission (Fig. 1d, Supplementary Data 15 and 16). Strikingly, our analysis reveals extensive lineage-specific fusion-with-mixing in *L. anatina*, with chromosomes 1, 2, 3 and 7 all the products of fusion events (Fig. 1d). The giant 71.9 Mb *L. anatina* chromosome 1 consists of nine bilaterian ALGs (C1, G, I, J2, K, L, O1, O2 and R) and corresponds to six separate *P. maximus* chromosomes (Fig. 1d and Supplementary Fig. 18). Further comparison with the annelid *O. fusiformis* and the nemertean *L. longissimus* suggests independent fusion events in brachiopods and annelids (Fig. 1d and Supplementary Fig. 18).

We next identified the genomic locations of BMP pathway components to assess their associations with an ALG. Our results show remarkable conservation: in each of the five selected bilaterian genomes, all 23 annotated BMP components are invariably linked to the same ALG (Fig. 1e and Supplementary Data 17). This exceeds the usual ALG conservation rate found in randomly selected genes (chi-square test; $P = 0.003$) (Supplementary Data 18), suggesting a higher tendency for BMP pathway genes to maintain their specific ALG associations. Even in species whose genomes have undergone extensive interchromosomal rearrangements, like that of the bryozoan *M. membranacea*[84,85], 16 of 17 BMP pathway gene families present have conserved ALG associations. This suggests that BMP-related gene linkages are highly stable but not completely fixed, even in lineages where extensive interchromosomal rearrangements have occurred. This evolutionary conservation of the genomic locations of BMP pathway components may reflect the preservation of regulatory programs across large evolutionary spans.

To determine if the conservation of ALG associations is a characteristic unique to BMP pathway components, we extended our analysis to other genes with conserved, ancient roles in development. All 12 Wnt ligands found in both spiralians and chordates demonstrated 100% association with a single ALG across the five studied species, a rate significantly higher than the average ALG conservation

rate (chi-square test; $P = 0.038$) (Supplementary Data 19). In addition, the Hox gene cluster is consistently associated with bilaterian ALG B2 (e.g. chromosome 16 in *B. floridae*) in all analysed species. This persistent conservation of developmental genes' genomic locations from brachiopods to chordates, despite widespread chromosomal rearrangements, implies evolutionary constraints on the mobility of these genes among ALGs. We propose that there is a stronger selective pressure at crucial developmental loci to maintain associations with regulatory elements in nearby genomic regions, thereby preventing their translocation and resulting in highly conserved macrosynteny.

## Brachiopod embryos exhibit a BMP seesaw with dorsal BMP signal activation

The use of the BMP–chordin signalling pathway for dorsal–ventral axis patterning is a conserved feature of the bilaterian body plan (Fig. 2a). This pathway operates through Smad1/5 phosphorylation, leading to nuclear translocation and regulation of downstream BMP signalling genes (Fig. 2b). To explore BMP–chordin pathway gene expression during *L. anatina* development, we quantified the expression of BMP pathway genes in an RNA sequencing time course[59] (Fig. 2c, d, Supplementary Fig. 19 and Supplementary Data 20). *Smad1/5* and *bmp5–8* are expressed maternally with their expression persisting through early development and into the larval stages (Fig. 2d). Although the *L. anatina* genome contains two copies of the *smad1/5* gene (Supplementary Fig. 9), only one of these copies is active during early development (Fig. 2d), while the other is most highly expressed in the gonad (Supplementary Fig. 20). Core BMP pathway genes *bmp2/4, bmp3* and *chordin* show early upregulation in the blastula, while modulators *bambi* and *cv2* increase during the gastrula stage. The expression of *admp* peaks later at the larval stage. BMP receptors are ubiquitously expressed, indicating that their expression pattern plays a non-crucial role in the timing of BMP pathway activation (Fig. 2d). Interestingly, modulators like *follistatin, gremlin, noggin* and *nbl1* show negligible expression in these stages, suggesting that they are not involved in dorsal–ventral patterning.

To elucidate the role of BMP signalling during *L. anatina* embryonic development, we first performed in situ hybridisation for the four BMP signalling ligands (*bmp2/4, bmp3, bmp5–8* and *admp*) and their main antagonist, *chordin* (Fig. 2e). Gene expression was assayed at three developmental stages: blastula, late gastrula and larva. In the blastula, *chordin* is expressed in the vegetal endomesoderm, *bmp3* is localised to the dorsal ectoderm, and *bmp5–8* is expressed near-ubiquitously, but most highly in the dorsal ectoderm. *Bmp2/4* is expressed to a low level in the endomesoderm. At the gastrula, there is a BMP seesaw of strong dorsal *bmp2/4* and *bmp3* expression and equally strong opposing ventral *admp* expression, in the same location as *chordin*. Expression of *bmp2/4, bmp3* and *bmp5–8* is also observed in the endomesoderm. In the larva, which is the product of significant reorganisation post-gastrulation (Supplementary Fig. 21), *bmp2/4* is expressed in the centre of the tentacle and the gut, while *bmp3* and *bmp5–8* are also expressed in the gut. *Admp* is expressed in the lateral part of the tentacle while *chordin* expression is near-absent but may persist at a low level in the same domain as *admp*.

We next investigated how these signalling ligand and antagonist expression domains translate to localised or asymmetrical activation of the BMP signalling pathway during *L. anatina* development. We assessed BMP pathway activity by detecting the presence or absence of phosphorylated Smad1/5 (pSmad1/5), which serves as a readout of BMP signalling. Using an antibody targeting a conserved PHNPISSVS phosphopeptide sequence present in metazoan Smad1/5 proteins (Supplementary Fig. 22), we examined the distribution of pSmad1/5 across a developmental time course. From the 1-cell to 128-cell stages, we detected no pSmad1/5 nuclear signal (Fig. 2d, f–n and Supplementary Fig. 23). Expression of the BMP ligand *bmp2/4* is very low

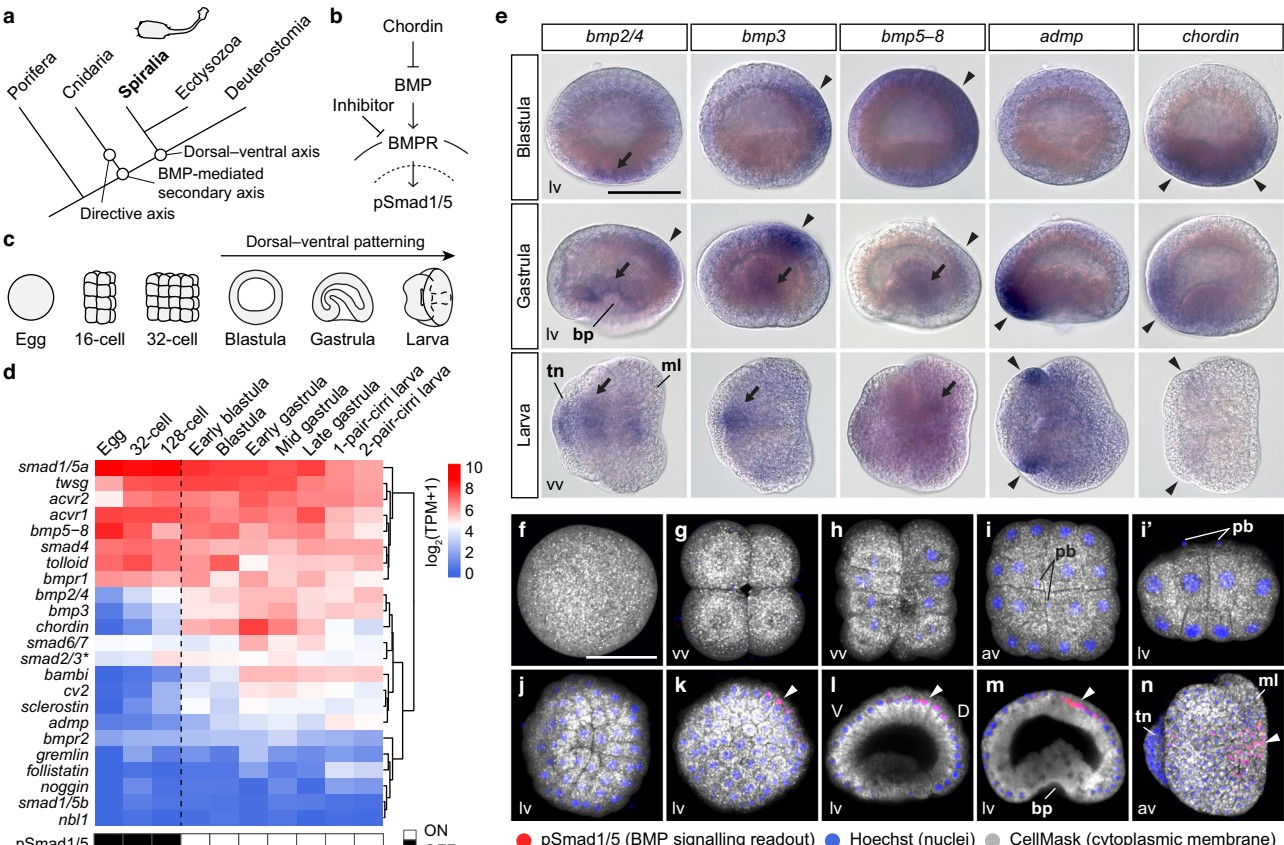

**Fig. 2 | Brachiopod embryos express all essential BMP pathway components and exhibit a BMP seesaw. a** Schematic illustration showing the evolution of the directive and dorsal−ventral axes in animals. The use of a BMP signalling gradient to pattern the dorsal−ventral axis is an ancestral feature of the bilaterian body plan. Whether the directive axis of cnidarians and dorsal−ventral axis of bilaterians are homologous is unknown. **b** Canonical BMP signalling pathway regulated by chordin and mediated through a heterotetramer of BMPR type I and type II receptors and resulting in phosphorylation of Smad1/5 (pSmad1/5). Solid line represents the plasma membrane, while dashed line denotes the nuclear envelope. BMP receptors are located on the surface of the plasma membrane. **c** Schematic of *L. anatina* embryonic development. Dorsal−ventral patterning begins at the blastula stage. **d** Expression profiles of *L. anatina* BMP signalling ligands, receptors, mediators and modulators. The appearance of nuclear pSmad1/5 signals is represented by a transition from black to white rectangles. The asterisk marks *smad2/3,* which is a canonical TGF-β pathway component that can be used by BMPs under specific circumstances. TPM, transcripts per million. **e** Spatial expression patterns of BMP ligands and antagonists in the *L. anatina* blastula, gastrula and larva. Arrowheads mark ectodermal gene expression. Complete arrows denote endomesodermal gene expression. **f**−**n** Immunostaining of pSmad1/5 (red) at early embryonic stages shows signals with asymmetrical nuclear localisation (arrowheads). Nuclei are labelled with Hoechst 33342 (blue). Cytosol is counterstained with CellMask deep red (grey). Embryonic stages: 1-cell (**f**); 4-cell (**g**); 16-cell (**h**); 32-cell (**i** and **i′**); 64-cell (**j**); blastula (**k**); early gastrula (**l**); late gastrula (**m**); early larva, ventral to the left (**n**). vv vegetal view, av animal view, lv lateral view, pb polar body, bp blastopore, tn tentacle, ml mantle lobe, V ventral, D dorsal. Scale bar, 50 μm. Source data are provided as a Source data file.

before the early blastula stage (Fig. 2d), suggesting that BMP signalling is not activated until after the cleavage stages. However, we did not directly test the effects of BMP signal manipulation at this stage, and further experimentation is required to confirm this. Nuclearized pSmad1/5 appears abruptly in the blastula, where it is asymmetrically localised at the animal pole towards the dorsal side (Fig. 2k). This occurrence aligns with the upregulation of *bmp2/4, bmp3* and *chordin* (Fig. 2d) observed in RNA-seq data, suggesting an importance of these proteins to the regulation of this signal. The presence of pSmad1/5 then persists on the dorsal side through the gastrula and early larval stages (Fig. 2l−n), demonstrating dorsal activation of the BMP signalling pathway. This pattern is consistent with those found in other protostomes and ambulacrarian deuterostomes, indicating the presence of a deeply conserved dorsal−ventral asymmetry of BMP pathway activation in spiralians.

**Asymmetric BMP activation regulates dorsal−ventral patterning**
To investigate the role of asymmetric BMP activation, we perturbed BMP signalling with receptor inhibitors and recombinant BMP proteins

to block or enhance its activity, respectively. Treatments were applied either from early blastula to late gastrula, when dorsal−ventral molecular features first appear, or extended to the one-pair-cirri larva, after axis patterning is established (Fig. 3a and Supplementary Data 21). In control late gastrulae, pSmad1/5 staining was restricted to the dorsal side (Fig. 3b, c). Inhibition of BMP receptors with LDN-193189 (LDN)[86] or K02288 (K02)[87], collectively referred to herein as BMP(−), abolished pSmad1/5 signal entirely (Fig. 3d, e). Conversely, treatment with recombinant mouse BMP4 protein (BMP)[88], referred to as BMP(+), expanded nuclear pSmad1/5 across the whole gastrula (Fig. 3f), validating the efficacy of the manipulations. pSmad1/5 staining confirmed that both inhibitors and recombinant proteins remained effective until the larval stage (Supplementary Fig. 24). Notably, both BMP(−) and BMP(+) treatments disrupt gastrulation. BMP(−) treatment results in a shortened archenteron (Fig. 3d) or a complete delay in invagination when using a stronger BMP inhibitor (Fig. 3e). In contrast, BMP(+) treatment delays invagination but promotes ingression (Fig. 3f). These observations show the importance of the BMP gradient in gastrula development.

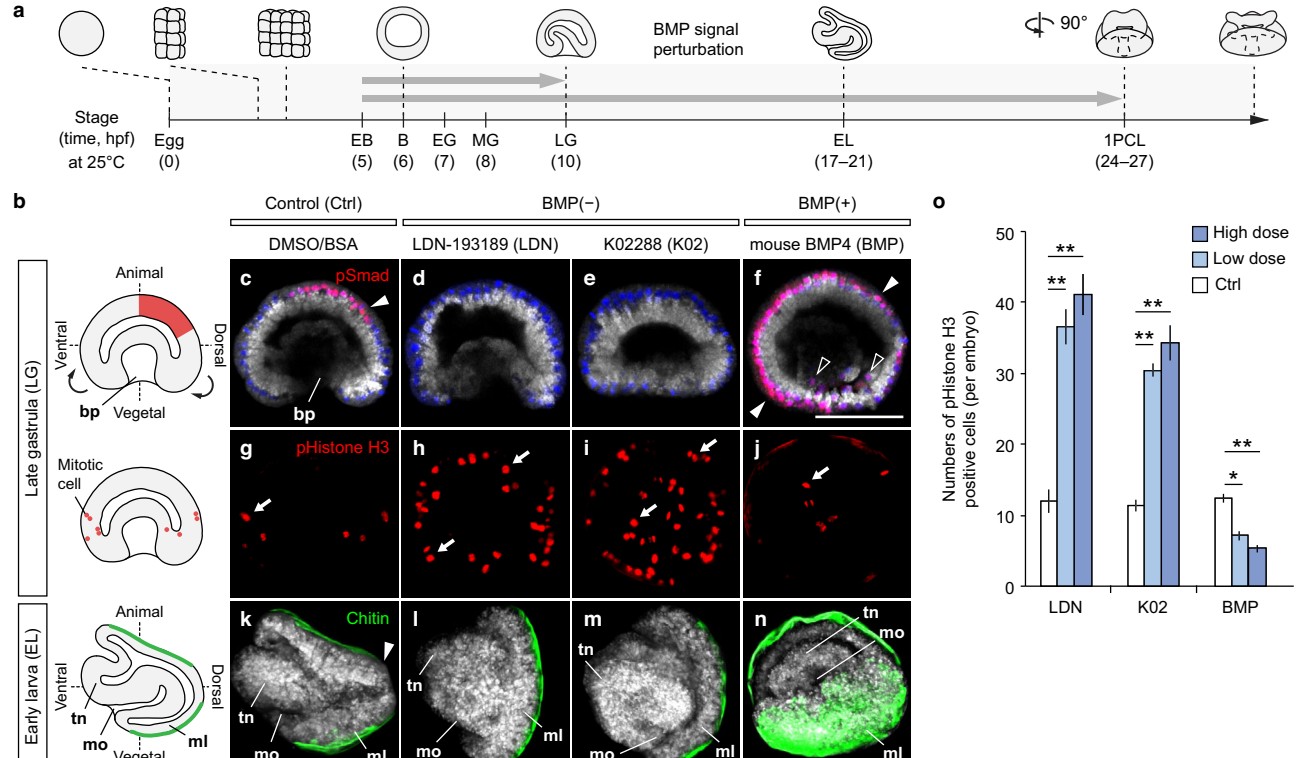

**Fig. 3 | Asymmetric BMP signalling regulates brachiopod dorsal−ventral patterning and cell proliferation. a** Developmental timeline of *L. anatina* (hours post-fertilisation, hpf) and BMP manipulation windows. Embryos were treated with inhibitors (LDN-193189 and K02288) or recombinant BMP4. EB early blastula, B blastula, EG early gastrula, MG mid-gastrula, LG late gastrula, EL early larva, 1PCL one-pair-cirri larva, 1PCL is rotated 90° relative to EL. **b** Body axes and gastrulation movements. bp blastopore, tn tentacle, mo mouth, ml mantle lobe. pSmad1/5 (red) marks the BMP gradient; mitotic cells (pHistone H3) are shown as red circles; embryonic shells (chitin) are green. **c–f** Immunostaining of pSmad1/5 antibody (red) in optical sectioned late gastrulae. High-dose treatments are shown unless noted. BMP inhibitor treatment disrupted gastrulation with K02288 causing a more severe phenotype. Nuclei are labelled with Hoechst 33342 (blue). Cytosol is counterstained with CellMask deep red (grey). Nuclearized pSmad1/5 signals are marked by arrowheads and empty arrowheads in blastodermal and ingressed mesenchymal cells, respectively. Scale bar, 50 μm. **g–j** Immunostaining of pHistone H3 antibody (red). Mitotic cells are indicated by arrows. **k–n** Staining of chitin with a chitin-binding probe (green). Chitin staining marks the embryonic shells. The arrowhead marks the dorsal edge of the larva, which lacks chitin staining in the control (**k**). Larvae are shown in lateral view (**k**, **l**), with a slight rotation in (**n**) to bring the ventral side partially into view. **o** Statistics of cell proliferation under BMP signalling perturbation (*n* = 4−5 embryos). High dose (LDN-193189, 8 μM; K02288, 500 nM; mBMP4, 200 ng/mL); low dose (LDN-193189, 4 μM; K02288, 250 nM; mBMP4, 100 ng/mL); data are presented as mean ± SEM; asterisks, unpaired two-sided t-tests for comparing between control and treatments (*$P < 0.05$; **$P < 0.01$). Exact *P*-values: LDN-193189, 0.0059/0.0075; K02288, 0.0002/0.0042; mBMP4, 0.0265/0.0045 (low dose/high dose). Source data are provided as a Source data file.

We next explored the influence of BMP signalling on cellular proliferation and the cell cycle by quantifying gastrula cells marked with the mitosis-specific phospho-histone H3. BMP(−) treatments significantly increased proliferative cell numbers compared to the control, whereas BMP(+) embryos showed a decrease in mitotic cells ($P < 0.05$) (Fig. 3g–j, o). This is consistent with dose-dependent effects of BMP signals reported from vertebrate development, where low doses promote proliferation while high doses inhibit proliferation and induce terminal differentiation[89–91], and suggests that a similar effect may be exerted during *L. anatina* development.

During *L. anatina* development, the embryo develops a mantle fold which is subsequently split into two mantle lobes that go on to form the two shells (Fig. 3b)[92]. We assessed the necessity of BMP signalling to this process by manipulating the BMP pathway and observing embryonic shell formation with a chitin-binding probe. In control embryos, the larval shell is marked by chitin staining across both mantle lobes, excluding the dorsal edge (Fig. 3k). In contrast, BMP(−) embryos showed fused shell fields and a single, unseparated shell, indicated by continuous chitin staining, including at the dorsal edge (Fig. 3l, m). This suggests that the depressions that split the mantle fold into two lobes[92] during normal development have failed to form properly. Meanwhile, BMP(+) embryos exhibited extended shell domains, with the chitinous shell extending ventrally, resulting in the

tentacle and mouth being encased inside the shell (Fig. 3n). The lack of a clear opposite effect on shell formation in BMP(−) and BMP(+) embryos may be because the effects of BMP signalling on mantle lobe and shell formation are a product of interference with embryonic morphogenetic processes (e.g. the formation of the furrows that split the mantle fold) rather than a direct promotion or inhibition of shell formation. The disturbance of the distribution of proliferative cells may also contribute to this process. Overall, these experiments indicate that dorsal BMP signalling is essential for splitting the developing larval shell into two distinct valves[59] and, more broadly, maintaining embryonic structural integrity and proper morphogenesis.

## BMP signals control neural and cell cycle gene expression

We further explored the role of BMP signalling in *L. anatina* development through functional transcriptomics using RNA sequencing on control, BMP(−) and BMP(+) embryos at both the late gastrula and one-pair-cirri larval stages (Supplementary Data 22–24). Differential gene expression and hierarchical clustering analyses identified gene sets regulated by BMP signalling. Genes downregulated under BMP(−) treatment and upregulated under BMP(+) treatment were classified as BMP-upregulated whereas genes showing the opposite pattern were categorised as BMP-downregulated. At the late gastrula stage, this

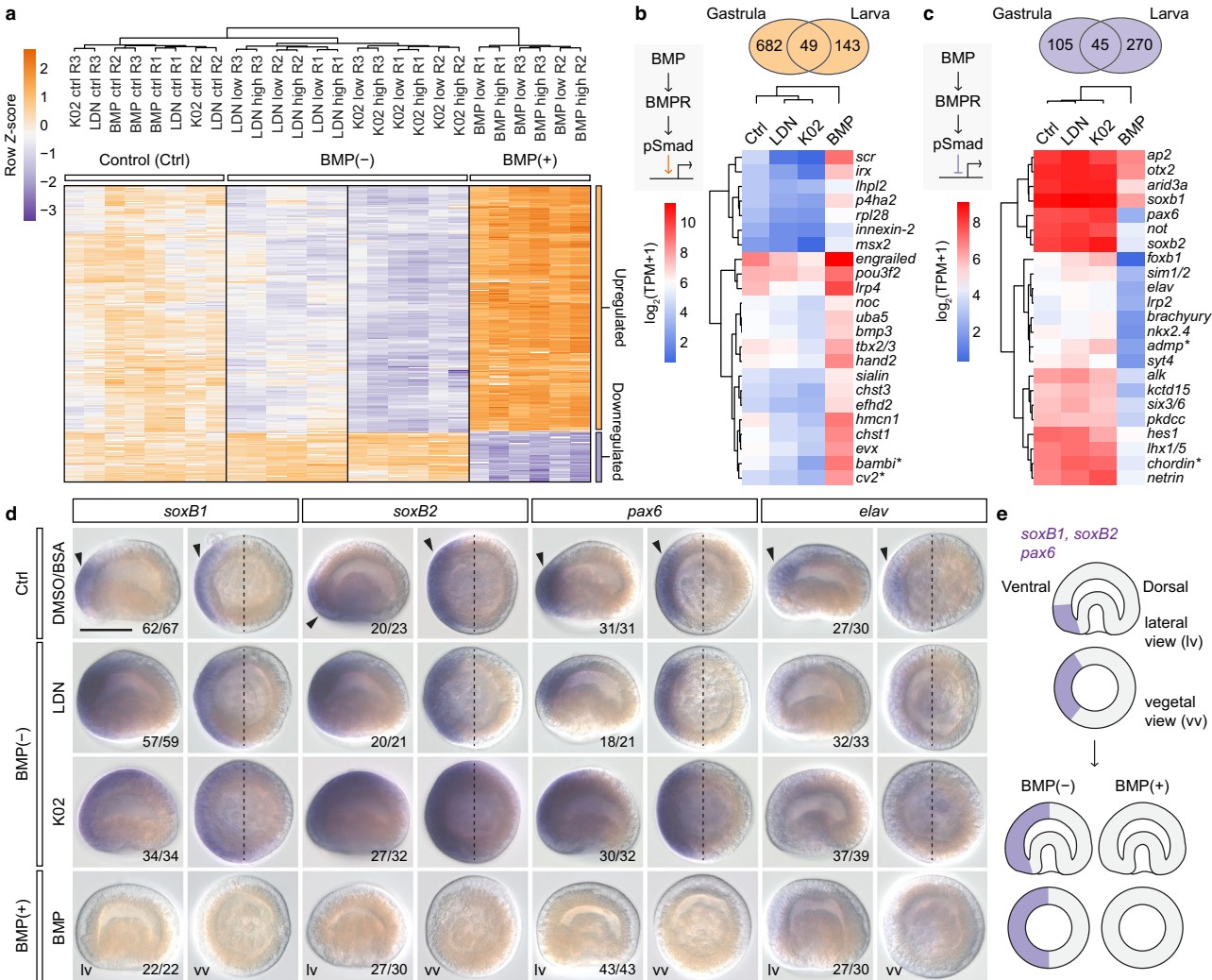

**Fig. 4 | BMP signals suppress the expression of genes involved in neurogenesis.**
**a** Hierarchical clustering heatmap of 881 differentially expressed genes at the late gastrula stage under the manipulation of BMP signals. Embryos were treated from the early blastula stage (5 h post-fertilisation, hpf) to the late gastrula stage (10 hpf). The *Z*-score scale represents log-transformed and mean-subtracted transcripts per million (TPM). BMP-upregulated (731, orange) and downregulated (150, purple) gene groups are labelled on the right. Columns represent samples and rows represent genes. ctrl control, R biological replicate. **b, c** Restricted dataset of genes with strong evidence for being regulated by BMP signalling in the *L. anatina* embryo. Venn diagrams show the overlap of BMP-upregulated (**b**) and BMP-downregulated (**c**) genes in the two experiments we conducted (late gastrula and early larva). Only genes identified in both the late gastrula and early larva experiments (49 genes for BMP-upregulated and 45 genes for BMP-downregulated) were used for further analysis and the heatmap below. Heatmaps show expression profiles of selected BMP-upregulated (**b**) and BMP-downregulated (**c**) genes at the late gastrula stage. The BMP-downregulated gene set contains many genes known for their neural expression in model organisms, such as *elav* and *soxb1*. Asterisks highlight *Xenopus* ventral centre and Spemann–Mangold organiser genes. **d** In situ hybridisation experiments showing the response to manipulation of BMP signals of genes associated with neural domain expression, *soxb1*, *soxb2*, *pax6* and *elav*. In control embryos, all four genes are expressed ventrally (arrows), directly opposite the dorsal BMP maximum. BMP(−) treatment results in dorsal expansion of neural gene expression domains. BMP(+) treatment results in the elimination of *soxb1*, *soxb2* and *pax6* expression while *elav* expression is reduced but still present. lv lateral view, vv vegetal view. Scale bar, 50 μm. **e** Schematic illustration of *soxb1*, *soxb2* and *pax6* expression in control and manipulated embryos. Source data are provided as a Source data file.

produced a set of 881 genes with strongly significant expression changes (fold-change > 4; *P* < 0.001) (Fig. 4a and Supplementary Fig. 25). Results were consistent across both independent BMP inhibitor molecules (LDN and K02), suggesting that the observed phenotypes were due to BMP signal disruption rather than off-target effects of a specific inhibitor.

We used gene ontology (GO) analysis to characterise these gene sets (Supplementary Data 25–28). Among BMP-downregulated genes, 'nervous system development' was one of the most highly enriched terms (Supplementary Fig. 25). To investigate this further, we sought to use a highly robust conservative dataset of genes for which we had strong evidence for regulation by BMP signalling. To this end, we restricted the dataset to only genes that were BMP-regulated in both

our late gastrula and early larval stage experiments, resulting in a total of 49 BMP-upregulated and 45 BMP-downregulated genes (Fig. 4b, c and Supplementary Fig. 26). In this dataset, the BMP-downregulated targets were dominated by genes demonstrating either exclusively neural expression or known to be expressed in neurons in specific developmental contexts in other model systems, including *elav*, which may be a universal neural marker[93], *foxb1*, *lhx1/5*, *netrin*, *nkx2.4*, *otx2*, *pax6*, *six3/6*, *sim1/2*, *soxb1* and *soxb2*[94–101] (Fig. 4c, Supplementary Figs. 27 and 28).

To validate the transcriptomic data, we examined expression of four key neural markers—*soxb1, soxb2, pax6* and *elav*—by in situ hybridisation in control, BMP(+) and BMP(−) embryos (Fig. 4d and Supplementary Data 29). In control embryos, all four genes were

expressed ventrally, directly opposite the dorsal BMP maximum (Fig. 4d, top row). For *soxb1*, *soxb2* and *pax6*, inhibition of BMP signalling caused a dorsal expansion of expression domains, with stronger effects under K02 compared to LDN treatment (Fig. 4d, second and third rows). In contrast, BMP(+) treatment abolished detectable expression of these three genes (Fig. 4d, bottom row). *Elav* expression was slightly reduced under BMP(+) and showed a slight, though less clear, upregulation with K02 (Fig. 4d). This is consistent with the RNA-seq data, where the effect on *elav* is less pronounced than other neural genes. Overall, these data show that key neural genes are expressed in the same location as *chordin*, diametrically opposed to the high point of BMP signal activation. Moreover, their expression domains expand when BMP signalling is inhibited and are eliminated when it is over-activated, confirming that BMP signals restrict the expression of neural genes. This finding, supported by both transcriptomic and in situ hybridisation data, contrasts with a recent study that reported an unusual positive effect of BMP signals on neural domain formation in spiralians[55,56], but is consistent with the observed inhibitory effect of BMP in the neuroectoderm of non-spiralian protostomes[102] and deuterostomes[15,45]. Our results suggest that inhibition of neural development by elevated BMP levels is a mechanism shared between spiralians, ecdysozoans and deuterostomes, despite the evolutionary diversification of embryological development modes in molluscs[55] and annelids[48].

In addition, at the gastrula stage, each of the 12 most statistically significant GO terms for BMP-downregulated genes relates to processes involved in cell cycle regulation (Supplementary Data 30), including 'DNA replication', 'chromosome organisation' and 'cell cycle'. BMP-downregulated genes include *atr*, *cdc16*, *ctf18*, *pcna* and *pole*, all of which encode essential components of the cellular DNA replication machinery[103–108]. All six *mcm2–7* genes, encoding the core components of the eukaryotic replicative helicase complex[109] are also BMP-downregulated. Conversely, no GO terms associated with BMP-upregulated genes relate to DNA replication or the cell cycle (Supplementary Data 25). This is consistent with the above result that the presence of mitotic cells was reduced in BMP(+) embryos and increased in BMP(−) embryos, and supports the contention that BMP signals inhibit cellular proliferation at this time point. A role for BMPs in regulating proliferation is well-known but the extent and direction of effect is highly context-dependent[110].

## Deep conservation of the dorsal–ventral patterning pathway across bilaterians

We next investigated whether the downstream effects of BMP signalling on gene expression that we observed in *L. anatina* show similarities to those of other bilaterians. We selected the vertebrate model *Xenopus* for this comparison, as the role of BMP signalling in dorsal–ventral patterning is well characterised[5] and its phylogenetic position within the deuterostomes means that evolutionary inferences can be made that span the entire Bilateria. In *Xenopus,* a ventral signalling centre produces the BMP ligand Bmp4[111] plus feedback inhibitors like Cv2[112], Twisted-gastrulation[113–115] and Bambi[116], while the dorsal Spemann–Mangold organiser secretes Admp[117] and a set of BMP antagonists that includes chordin[4,6,7], Follistatin[118] and noggin[8,9]. These antipodal signalling centres construct the dorsal–ventral BMP gradient that patterns the axis, with the neural inducers secreted from the Spemann–Mangold organiser causing the formation of neural tissue on the dorsal side.

We found that several brachiopod homologues of key *Xenopus* dorsal–ventral patterning genes were up- or down-regulated at the RNA level in the above BMP signal manipulation experiments. Specifically, BMP signalling in *L. anatina* downregulates homologues of the *Xenopus* Spemann–Mangold organiser genes *chordin* and *admp*, while upregulating homologues of the *Xenopus* ventral centre genes *bambi* and *cv2* (Fig. 4b, c).

The parallel in regulatory logic between *Xenopus* and *L. anatina* suggests a conserved BMP signalling network architecture (Fig. 5a). We next investigated whether the spatial expression patterns of these genes in *Xenopus* are also conserved in *L. anatina*. In situ hybridisation experiments show that *bambi* and *cv2* are dorsally localised in *L. anatina* gastrulae, while *chordin* and *admp* are found ventrally (Fig. 5b). This is consistent with their expression in *Xenopus* when the axis inversion is taken into account. Notably, both *chordin* and *admp* were expressed at the boundary of the ventral ectoderm and endo-mesoderm, similar to the expression pattern observed in the cephalochordate *B. floridae*[15] (Fig. 5b). *Bambi* and *cv2* were expressed in domains of active BMP signals on the dorsal side. BMP(−) treatment causes expansion of *chordin* and *admp* expression domains and the near absence of *bambi* and *cv2* expression. In contrast, BMP(+) treatment results in the near absence of expression of *admp* and *chordin* while *bambi* and *cv2* are expressed universally (Fig. 5b). These experiments support our transcriptomic results showing that BMP signalling upregulates *bambi* and *cv2* while downregulating *chordin* and *admp*. This result is also consistent with regulatory interactions observed in *Xenopus,* where BMP signalling represses *chordin*[5] and *admp*[117,119] while activating *bambi*[116,120] and *cv2*[5,121]. Overall, we found conservation of *Xenopus* dorsal–ventral patterning gene regulatory logic and gene expression patterns in the brachiopod *L. anatina* (Fig. 5c, d), indicating deep conservation of these pathways in spiralians.

Despite this remarkable conservation, several *Xenopus* Spemann–Mangold organiser and ventral centre genes (e.g. *twsg*, *noggin* and *follistatin*) do not appear in our restricted differentially expressed lists (Fig. 4b, c). We hypothesised that this may reflect either our highly conservative approach to differentially expressed gene identification or genuine divergence in the role of these genes between species. To investigate this, we studied the expression of 15 key *Xenopus* Spemann–Mangold organiser and ventral centre genes under each manipulation condition. Intriguingly, many of these genes are differentially expressed in response to BMP signalling in *L. anatina*, but not in the predicted direction. For instance, the Spemann–Mangold organiser gene *gremlin* is BMP-upregulated when it might be expected to be downregulated like *chordin* and *admp*, and vice versa for *twsg* (Fig. 5e). These findings, while requiring further validation, suggest that although the central BMP–chordin axis is conserved, its downstream effects are evolutionarily variable. Accordingly, a phylostratigraphy approach using transcriptome age index (TAI) analysis[122] shows that increasing the level of BMP signalling significantly decreases the evolutionary age of the transcriptome during both gastrula and larval stages of development (two-sided t-test $P < 0.05$; Fig. 5f, g and Supplementary Data 31–33). This result suggests that activating the BMP pathway in *L. anatina* leads to the upregulation of evolutionarily young genes. Indeed, our findings show that during the developmental time course, the late blastula and early gastrula stages—following BMP signal activation—exhibit the highest TAI, indicating the youngest transcriptome (Supplementary Fig. 29). This aligns with transcriptome ages at similar stages in zebrafish development[122]. When combined with our BMP signal inhibition experiments, the findings suggest that BMP signals are involved in a gastrulation process that is marked by increased expression of genes that emerged relatively recently in evolution.

## Discussion

Elucidating the evolution of axial patterning mechanisms within bilaterian animals remains a key goal of evolutionary developmental biology. Within this group, spiralians stand out as having highly diverse mechanisms of dorsal–ventral patterning that are only beginning to be understood. Our study aims to expand on current knowledge by investigating a relatively understudied spiralian phylum, the brachiopods. We find BMP signal readout at the dorsal side of the *L. anatina*

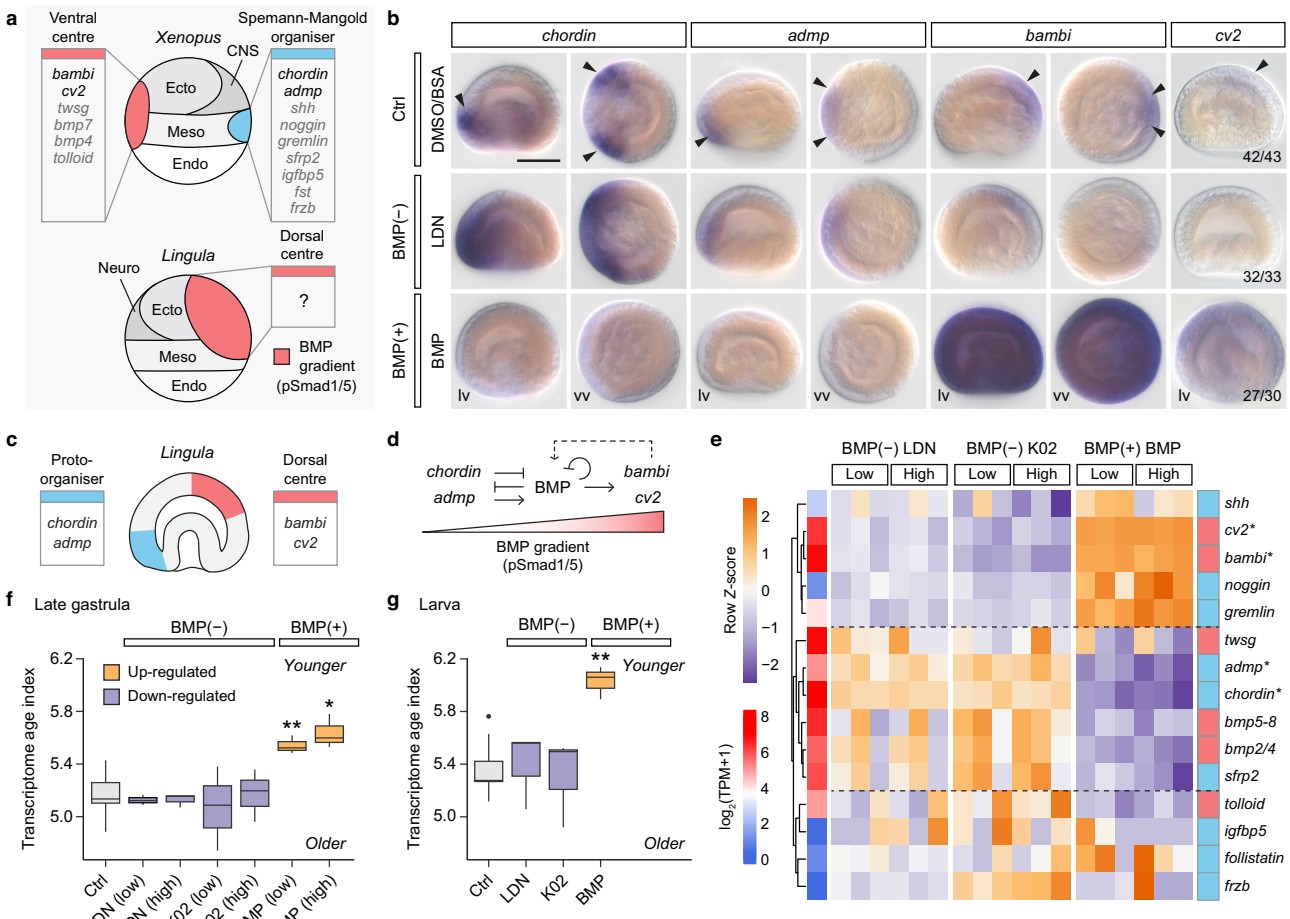

**Fig. 5 | Brachiopods exhibit BMP-mediated patterning similar to that of the vertebrate *Xenopus*. a** Top: schematic illustration of proteins secreted by ventral and dorsal (Spemann–Mangold organiser) signalling centres in the *Xenopus* gastrula. Bottom: illustration of BMP signal activation in the *L. anatina* gastrula corresponding to the ventral centre in *Xenopus*. Ecto ectoderm, meso mesoderm, endo endoderm. **b** The expression profile of *Xenopus* dorsal genes (*chordin* and *admp*) and ventral genes (*bambi* and *cv2*) under the manipulation of BMP signalling in *L. anatina*. lv lateral view, vv vegetal view. Scale bar, 50 μm. **c** Schematic illustration of the expression domains of BMP signalling genes at the *L. anatina* late gastrula stage. **d** Proposed molecular network of the BMP signalling genes in a BMP–chordin axis generating a BMP gradient (red triangle). Dashed line indicates an unknown interaction between the modulators and BMP signals. **e** Expression profile of orthologues of key *Xenopus* BMP pathway genes in the *L. anatina* gastrula under BMP signalling manipulation. *Xenopus* ventral centre genes (red) and Spemann–Mangold organiser genes (blue) labelled on the right. Asterisks highlight genes within the *Xenopus* ventral centre and the Spemann–Mangold organiser that BMP signals similarly regulate in brachiopods. Blue/red heatmap on the left-hand side shows mean expression of the genes across the experimental conditions (Log$_2$[TPM + 1]). Transcriptome age index of *L. anatina* late gastrula embryos (**f**) and larvae (**g**) under conditions of BMP signal manipulation. Asterisks show significant differences from control samples at $P < 0.05$ (*) and $P < 0.01$ (**) as determined by two-sided t-tests ($n = 3$ batches). Box plots show the median (centre line), first and third quartiles (box bounds), and whiskers extend to 1.5 times the interquartile range. Exact $P$-values are reported in Supplementary Data 33. A higher transcriptome age index indicates an evolutionarily younger transcriptome. TPM, transcripts per million. Source data are provided as a Source data file.

gastrula, consistent with ecdysozoan protostomes like *Drosophila*[10–12] and ambulacrarian deuterostomes, such as sea urchins[16,45] and hemichordates[14,45]. Asymmetric activation of the BMP signalling pathway is governed by ventral *chordin* expression and a BMP ligand seesaw, with ventrally localised Admp and dorsally localised Bmp3. BMP seesaws have been identified in cnidarians[39–41], acoels[123] (which may be the sister group to other bilaterians), deuterostomes[16,29,38–43] and even planarians[77], but not in lophotrochozoans like molluscs and annelids. The identification of such a set-up in a lophotrochozoan phylum, the brachiopods, when combined with data from other lineages, suggests that the last common ancestor of spiralians and even bilaterians may have utilised a BMP seesaw.

Though opposing sources of BMP ligands are broadly conserved, the identities of the ligands themselves vary markedly between lineages. This is reminiscent of development system drift (DSD), where changes in the molecular underpinnings of developmental processes occur over time through chance rather than by selection for phenotypic divergence[124]. Genikhovich et al.[29] argued that in the

BMP–chordin axis the location of *chordin* expression alone is sufficient to define gradient of BMP signalling, as it shuttles BMP ligands away from the side of its expression, making the expression domains of BMP ligands irrelevant. This would explain the extensive variation between species and is supported by manipulations of the *Drosophila* embryo, where the mis-expression of the usually-dorsal BMP ligand Dpp in a *dpp*-null embryo with an even-skipped stripe 2 enhancer resulted in a normal BMP gradient[125]. However, the fact that spatially-localised seesaw expression of BMP ligands is found in distantly related lineages despite over half a billion years of divergence points to some kind of functional role.

In addition to the shared BMP seesaw, other similarities between gene expression localisation and regulatory interactions in *L. anatina* and *Xenopus*, particularly those regarding the Spemann–Mangold organiser, are striking. In the early-branching chordate amphioxus, such similarities have been taken, along with transplantation experiments[126], as evidence that the organiser was present at the base of the chordates[15,127] or the deuterostomes[38], not just in vertebrates.

Moreover, a transplanted blastopore lip is capable of axis induction and organiser activity not only in vertebrates or chordates, but in animals as distantly related as sea anemones[128,129]. In light of the extension of these molecular similarities to *L. anatina,* we consider it a possibility that these features have been inherited and maintained from a bilaterian–cnidarian ancestor that possessed a 'proto-organiser'. Transplantation experiments testing whether certain regions of *L. anatina* embryos possess organiser capabilities should therefore be a priority, although in any case homology would be difficult to assess. Irrespective of the presence or absence of an organiser in *L. anatina*, the results of this study suggest deep evolutionary conservation between brachiopods and deuterostomes that supports the hypothesis that a classical BMP-mediated dorsal–ventral patterning system was present in the ancestor of spiralians despite their derived mechanism of early embryonic development (Fig. 6).

A role of BMP signals that is closely linked to but potentially independent from dorsal–ventral patterning is neural induction. Across both deuterostomes and ecdysozoans, BMP signals repress the formation of neural tissue[46]. One significant observation casting doubt on whether the ancestor of spiralians utilised BMP signals during early development in the same way as ecdysozoans and deuterostomes[51] was the discovery that, in some molluscs and annelids, BMP signals have no effect or even a positive effect on neural development[48,55-57]. Through functional transcriptomics of embryos with manipulated levels of BMP signalling, we found strong inhibition of genes associated with neural expression in several model organisms by the *L. anatina* BMP pathway during development. In situ hybridisation in BMP-manipulated embryos confirmed that the expression domains of these genes are restricted by BMP signals in vivo. These results suggest that the function of BMP signalling in restricting neural induction is similar in some spiralians to that in arthropods and deuterostomes. One possibility is that this function was ancestral to all three major bilaterian clades but several spiralian lineages modified or replaced it (annelids and molluscs, for instance) while it was conserved in

brachiopods. An alternative explanation recently suggested by Knabl et al.[130] is that the primary function of BMP signalling is dorsal–ventral patterning and the 'anti-neural' function observed in vertebrates, arthropods and now brachiopods is merely a 'side-effect'. If this is the case, BMP signalling may have originally promoted neurogenesis in the cnidarian–bilaterian common ancestor lacking a centralised nervous system[130]. In bilaterians, multiple lineages may subsequently have evolved convergent mechanisms that utilise it in the opposite manner to pattern the central nervous system[131]. Differences in the relationship between BMP signalling and neural induction[5] may therefore reflect general variation in nervous system organisation, potentially linked to independent evolution of central nerve cords across bilaterians[131,132].

Irrespective of the direction of change, it is clear that the role of BMP signals in neural induction and dorsal–ventral patterning shows extensive diversity across the Spiralia[42,47,48,52,55,56,133]. Why has a mechanism highly conserved from arthropods to vertebrates been repeatedly modified in this specific group? The answer likely lies in a combination of two features of spiralian development. First, spiralian development typically consists not only of spiral cleavage itself but also stereotyped specification of certain blastomeres (i.e. highly regular cell division and conserved cell lineages)[134,135]. This is intricately interlinked with axis specification and dorsal–ventral patterning. For example, the spiralian-specific D-quadrant organiser is a key contributor to dorsal–ventral axis specification[136-138], and the organising activity of the D-quadrant of the molluscs *Ilyanassa obsoleta* and *Lottia peitaihoensis* relies on Bmp2/4[55,56] (though this appears not to be the case in annelids[47,139]). Second, spiralian development is highly variable and it is clear that both spiral cleavage and stereotyped cell lineage specification have been modified and lost repeatedly[135,140,141]. Lineages exhibiting equal versus unequal cleavage, for instance, polarise their secondary axis at different times via different mechanisms[136,142] and brachiopods themselves, though phylogenetically unquestionably spiralians, do not show spiral cleavage. It follows that since spiralian development is highly variable and intertwined with dorsal–ventral

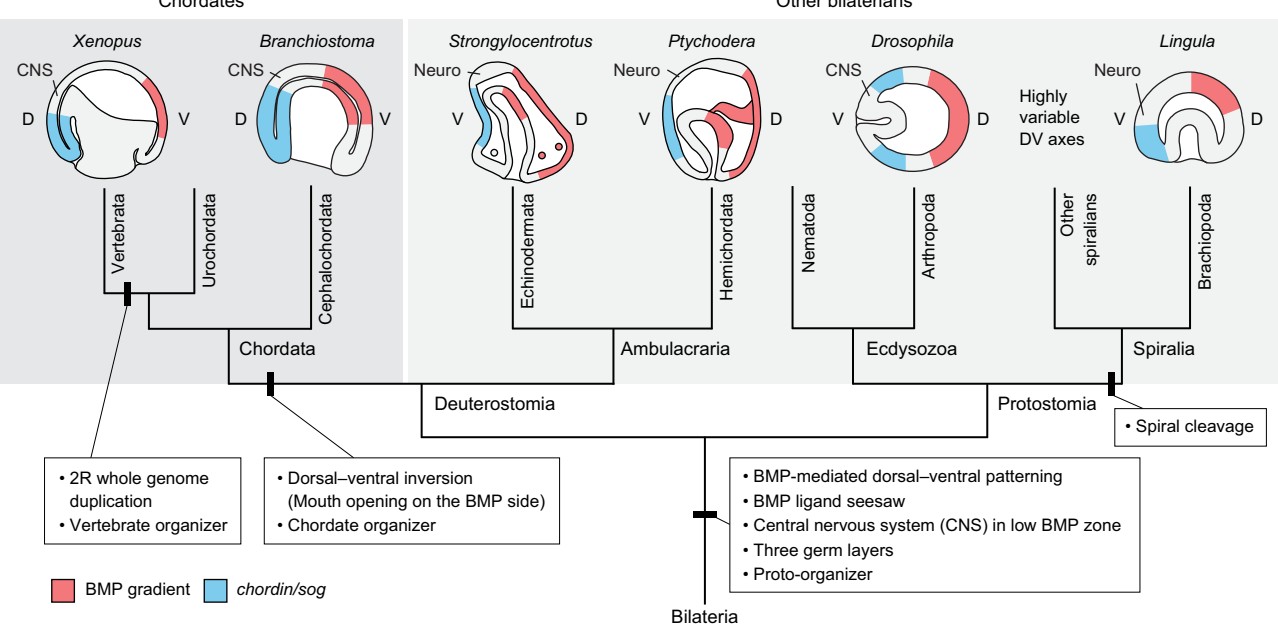

**Fig. 6 | Evolution of BMP gradients and dorsal–ventral patterning in bilaterians.** Although the spatial expression patterns of BMP ligands are variable, the domain with high BMP signalling readout (pSmad1/5, red) and the *chordin/sog* expression domain (blue) are on the dorsal (D) and ventral (V) sides, respectively, of the ecdysozoan and deuterostome embryos. Our data from *L. anatina* shows that this is also the case in spiralians. Due to dorsal–ventral axis inversion, this is

reversed in chordates. However, the core molecular patterning mechanisms remain unchanged, especially for those of modulators and antagonists. In all cases, the neural domain develops on the side of the *chordin/sog* expression domain, since BMP signalling inhibits neural (neuro) and central nervous system (CNS) formation in most bilaterians, including the spiralian *L. anatina*.

patterning, the role of BMP signalling is modified as part of this lineage-specific variation.

Variation of early development within spiralians may be an extreme example of DSD[57,42,52,143]. Our functional transcriptomic analyses, which revealed that (i) some, but not all, key BMP pathway genes are affected by BMP signalling in the same way in *L. anatina* and *Xenopus* and (ii) the *L. anatina* BMP pathway upregulates the expression of evolutionarily young genes, are consistent with DSD. Differences between spiralians, such as the direction of effect on neural induction, may therefore arise from lineage-specific divergence in the BMP pathway's downstream responses while the end result (i.e. formation of a ventral nervous system domain) is unchanged. This is supported by sensitivity analysis from both *Nematostella* and *Xenopus* that showed a core, strongly constrained network centred around the BMP ligands and chordin but much weaker constraints on peripheral components such as Bambi[29]. This is reinforced by data from distantly-related annelids where *O. fusiformis* and *C. teleta* employ divergent downstream gene sets to execute dorsal–ventral patterning[57]. The results of our spiralian-wide characterisation of BMP signalling pathway components lend further weight to this, showing strong conservation of *chordin* and BMP ligands but more variation elsewhere in the pathway. There may therefore be a hierarchy of constraint, where chordin is the most constrained component, BMP ligands are semi-constrained, and peripheral signalling modulators show significantly more flexibility.

Overall, our study supports a model where the core functions of the BMP pathway are ancient and deeply conserved, with the ancestor of spiralians having BMP-mediated dorsal–ventral axis patterning—likely with a BMP seesaw—and repression of neural domain formation in the same manner as arthropods and vertebrates. Several spiralian groups have subsequently modified this system in a lineage-specific manner through DSD or selection, and incorporated newly emerged genes, possibly as a result of broader changes in early development and nervous system organisation.

## Methods

### Biological materials and artificial fertilisation
Between 2013 and 2016, gravid adults of the brachiopod *L. anatina* (approximately 150–200) were collected each August during low tide from Kasari Bay, Amami Island, Japan (28.440583 N, 129.667608 E). Around 50 individuals were housed in a 10 L aerated seawater tank and fed daily with 25 mL/tank of *Chaetoceros calcitrans* ($5 \times 10^7$ cells/mL). Oocyte maturation was induced by injecting each gonad with 30 μL of 40 mM dibutyryl-cAMP dissolved in phosphate-buffered saline (PBS)[59,63]. To control fertilisation timing, post-injection individuals were isolated in separate Petri dishes. Artificial spawning was facilitated by adjusting the temperature from 25 to 29 °C over 2 h and then rapidly returning it to 25 °C for a cold shock[144]. Prior to inducing fertilisation, sperm activity was inspected under a stereomicroscope. Fertilisation efficacy decreased sharply 4 h after spawning. Depending on seasonal factors, spawning rates varied from 20 to 70%.

### Genomic sample acquisition and DNA extraction
In August 2021, fresh adults of *L. anatina* were collected with the objective of obtaining high-quality genomic DNA suitable for both long-read and Hi-C sequencing methodologies. The dissected tissues, including the lophophore, mantle and adductor muscle, were immediately snap-frozen in liquid nitrogen. For genome resequencing, approximately 2 g of lophophore and mantle tissue and 1.5 g of adductor muscle tissue were collected from a single adult specimen. These tissues were then divided into two equal portions. One portion was used for genomic DNA extraction while the other was reserved for Hi-C sequencing. The cetrimonium bromide (CTAB) method was utilised to extract high molecular weight genomic DNA. After extraction,

the DNA was purified using the blood and cell culture DNA midi kit (Qiagen 13343). DNA purity was assessed with a NanoDrop spectrophotometer, and DNA concentration was determined using the Qubit 4.0 Fluorometer. Genomic DNA quality was assessed by gel electrophoresis. Samples meeting the quality criteria showed a main band centred around 100 kb with no detectable signal below 40 kb. DNA purity was evaluated using NanoDrop (A260/280 = 1.8–2.0; A260/230 = 2.0–2.2), and DNA concentration was measured using both the NanoDrop spectrophotometer and the Qubit 4.0 Fluorometer.

### RNA sequencing of adult tissues
Total RNA was extracted from the mantle gonad, lophophore, adductor muscle and pedicle using TRIzol. The RNA was reconstituted in 50 μL RNase-free water and its concentration was determined with a Nanodrop spectrophotometer. RNA quality was verified using the 5400 fragment analyser system (Agilent Technologies). Total RNA from each tissue sample was used directly for mRNA enrichment and library preparation using the TruSeq RNA Library Prep Kit v3, following the manufacturer's protocol. Briefly mRNA was purified using oligo(dT)-attached magnetic beads, followed by fragmentation and first-strand cDNA synthesis with random hexamer primers. Second-strand synthesis was then performed and the resulting cDNA underwent end repair, A-tailing, adaptor ligation, size selection, amplification and purification to complete library construction. Libraries were sequenced as 150 bp paired-end reads on the Illumina NovaSeq 6000. Quality control was conducted using FastQC[145]. The reads were trimmed with Trimmomatic (v0.39)[146] and then aligned to the genome using STAR (v2.7.10b)[147] for gene prediction. For gene expression analysis, transcript abundances were quantified using kallisto (v0.43.0)[148] (Supplementary Data 34). The same method was used to process data from a published developmental dataset of *L. anatina*[59] (Supplementary Data 35).

### Genome sequencing and assembly
For PacBio HiFi circular consensus sequencing, SMRTbell libraries were constructed following PacBio's standard protocol, utilising the 15 kb preparation solutions. In brief, high-molecular-weight DNA was extracted from the lophophore sample and was used for DNA library preparations. The genomic DNA was sheared to the desired fragment size using g-TUBEs (Covaris). After removal of single-strand overhangs, damage repair, end-repair and A-tailing, DNA fragments were ligated with the hairpin adaptor. Post-ligation, the library was treated with nuclease, cleaned using the SMRTbell Enzyme Cleanup Kit, and purified with AMPure PB Beads (Beckman Coulter). The desired fragments were subsequently isolated using BluePippin (Sage Science). An Agilent 2100 Bioanalyzer was used to determine the size distribution of the library fragments. The final sequencing of the SMRTbell library was conducted on the PacBio Sequel II platform utilising Sequencing Primer V2 and the Sequel II Binding 2.0 Kit at Grandomics Biosciences. HiFi reads with a mean length of 13 kb were assembled using Hifiasm (v0.16.0)[149], and redundant haplotype sequences were removed with Purge_dups (v1.2.5)[150].

The Hi-C library was constructed following a previously described method[151]. In brief, mantle, lophophore and adductor muscle tissues were cut into 2 cm segments and cross-linked using a nuclei isolation buffer with 2% formaldehyde. Post cross-linking tissues were ground to produce a nuclei suspension. The isolated nuclei were digested with 100 units of MboI and labelled with biotin-14-dATP. Unligated fragments had their biotin removed using T4 DNA polymerase. The DNA was sheared to 300–600 bp fragments, underwent blunt-end repair and A-tailing, and was isolated with streptavidin beads. After quality checks with a Qubit Fluorometer and an Agilent 2100 Bioanalyzer, the libraries were sequenced as 150 bp paired-end reads on the Illumina NovaSeq 6000 platform.

Hi-C data was used for assembly scaffolding[152] using the following pipeline: trimmomatic (quality filtering)[146]; FastQC (quality control)[145]; bwa_mem2 (sequence alignment)[153]; Juicer (data filtering)[154]; 3D-DNA (Hi-C-assisted assembly)[155]; and JuiceBox (visualisation and error correction)[156]. Having obtained a preliminary assembly, blasts, read coverage and GC content were used to check for sequences belonging to contaminants and symbionts. The assembly was first subjected to Diamond[157] blastx against the nr database of NCBI with 'sensitive mode' parameter and an *E*-value of 1e-25. Subsequently Hi-C reads were mapped to the original genome using HISAT2[158] with default parameters and in single-end mode to calculate the coverage of each fragment. These three data types were visualised using blob plots created with BlobToolKit[159]. This methodology revealed a small cluster of short sequences with an unusually high GC content (greater than 45%) which blasted to non-brachiopod taxa. All such fragments were removed from the genome. A second blob plot produced from the remaining fragments, identified only sequences blasting to Brachiopoda. Sequences present after this filtering process represent the final *L. anatina* assembly.

## Gene prediction and annotation

Repetitive elements were annotated de novo using RepeatModeler (v2.0.4)[160]. RepeatModeler employs RepeatScout[161] and RECON[162] to identify transposable elements and also uses the long terminal repeat (LTR)-specific tools LTRharvest[163] and LTR_retriever[164]. The BRAKER pipeline (v3.0.3)[165–173] was then used for gene prediction and annotation. Genomes were first soft-masked with RepeatMasker (v4.1.5; sensitive mode)[174], and then BRAKER was run using hints from mapped RNA sequencing data. RNA-seq reads (Supplementary Data 36) were downloaded from the NCBI Sequence Read Archive, trimmed with trimmomatic (v0.39) and aligned with STAR (v2.7.10b) before input to BRAKER in BAM format. The RNA-seq datasets generated in this study were also used as hints for BRAKER. Gene annotation quality was assessed using BUSCO (v5.4.7)[175]. InterProScan[176], KofamScan[177] and EggNOG-mapper[178,179] were used for functional annotation. For KofamScan, output is limited to hits where threshold > score (adjudged to be a significant hit). Orthologues of *L. anatina* genes in human and mollusc (*Patella vulgata*) genomes were identified with OrthoFinder (v2.5.4)[180]. RepeatLandscape was used to create Kimura substitution level plots for repeats in lophophorate genomes. Ribosomal RNA (rRNA) genes were annotated with barrnap (v0.9). Gene density and repeat density plots were made with RIdeogram (v0.2.2)[181].

## Phylogenomic analysis

Proteomes were downloaded from NCBI or produced by the gene prediction method outlined above (Supplementary Data 12). Orthology assignment was performed with OrthoFinder. OrthoSNAP (v0.01)[182] was run with default parameters to recover additional orthologues for phylogenetics. The resultant dataset contained 2036 OrthoSNAP orthogroups. To determine the strength of the phylogenetic signal possessed by each orthogroup we first aligned sequences with MAFFT (v7.520)[183,184] and trimmed alignments with ClipKIT (v1.4.1)[185]. Individual gene trees were then constructed with IQ-TREE (v2.2.2.3)[186] using ModelFinder automated model selection[187] and UFBoot2 ultra-fast bootstrapping[188]. Orthogroups with an average bootstrap score over 85% ($n = 109$) were selected for species tree building and alignments concatenated with PhyKIT (v1.11.7)[189]. The final tree was built using IQ-TREE as above (model LG + F + R5).

## Comparative genomics

We used OrthoFinder to identify putative homologues of BMP pathway components in lophotrochozoan genomes. Genes were identified by orthology to sequences from human and *Drosophila,* which are well-annotated and verified. All cases of putative gene losses and

duplications were then manually verified using gene tree construction with IQ-TREE (v2.2.2.3) plus reciprocal blast searches and microsynteny comparisons where necessary. Gene trees for each superfamily were built using a pipeline of MAFFT (v7.520) alignment, ClipKIT (v1.4.1) trimming and IQ-TREE (v2.2.2.3) tree-building using 1000 ultrafast bootstraps and automated substitution model optimisation with ModelFinder. Attempts to assess the functionality of genes based solely on genomic sequence can lead to the erroneous designation of functional genes as pseudogenes. Thus, for the purpose of this analysis, we consider genes as only the number of copies of a gene and do not speculate on functionality. Human, *Xenopus* and *Drosophila* BMP pathway genes are well-characterised and gene counts are not re-assessed here.

Gene family evolution was modelled using CAFE (v5.0.0)[190]. CAFE implements a stochastic birth and death model to estimate the number of gene gains and losses occurring at each node in a tree. The species set used for this analysis was identical to that used for phylogeny reconstruction with one exception: the nemertean *Notospermus geniculatus* was removed because the published genome has high levels of redundancy, which would result in inaccurate estimates. To run CAFE, an ultrametric version of the above species tree was calculated using pyr8s from iTaxoTools[191] to implement r8s[192]. OrthoFinder was used as above for the orthology assignment. Principal component analysis was performed on the orthogroup-species matrix using R.

## Macrosynteny analysis

Macrosynteny was compared between *L. anatina* and species representing four other phyla with relatively conserved genomic organisations: *B. floridae* (Florida lancelet, Chordata), *P. maximus* (scallop, Mollusca), *L. longissimus* (bootlace worm, Nemertea) and *O. fusiformis* (Annelida). Proteomes for these species were obtained from NCBI. SyntenyFinder (v1.0)[85] was then used to run OrthoFinder to identify single-copy orthologues across the five genomes. To place the macrosynteny results in the evolutionary context of chromosome evolution across bilaterians, genes were assigned to the 24 bilaterian ancestral linkage groups (ALG)[81] by homology. SyntenyFinder then runs the R package RIdeogram (v0.2.2) to visualise the outcome with idiogram plots and Oxford dot plots.

The genomic locations of BMP pathway genes, Wnt ligands, and Hox genes in each of the five species were identified with BLAST (v2.14.1+). We then determined the ALGs present on the chromosome hosting each gene using the above macrosynteny analysis. Conserved associations with ALGs were inferred from the genes' chromosomal locations: if a gene was located on a chromosome containing the same ALG in all five species then it was considered to have a conserved ALG association. To determine whether the rate of conservation of ALG associations is elevated in BMP pathway genes, Wnt ligands, and Hox genes compared to the background rate of all genes in the genome, the rate of conserved associations of developmental genes with a specific ALG was compared to that of a random sample of 100 single-copy orthologues using a chi-square test (Supplementary Data 37). The random sample was created by running OrthoFinder to identify single-copy orthologues across the five species and then using the shuf command in bash with option -n 100 to choose 100 random rows of the output single-copy orthogroups file. The presence or absence of conserved ALG associations for these genes was then determined by the same method as above.

## Manipulation of BMP signals

BMP signalling manipulation experiments were deployed to reveal the function of the BMP pathway during *L. anatina* development. For the BMP(−) condition, two small molecule inhibitors, LDN-193189 (Stemgent 04-0074)[86] and K02288 (Tocris 4986)[87], were used to block the BMP pathway by inhibiting type I

receptors for BMP ligand proteins. Two independent inhibitors were used to confirm that the observed phenotypes resulted from genuine BMP signal perturbation rather than being specific to a particular inhibitor. Though both molecules are highly specific to BMP receptors as opposed to TGF-β, and K02 has been reported to have no effect on TGF-β-induced Smad2 phosphorylation, they may have a weak off-target effect on Activin signalling[87,193,194]. To check the potential for off-target effects on TGF-β and particularly activin signalling in our system, we annotated the two *activin/inhibin-like* genes and other TGF-β-like genes in *L. anatina* (Supplementary Fig. 30) and quantified their expression during embryonic development. All TGF-β-like genes including the two *activin/inhibin*-like genes are either not expressed or expressed at negligible levels at the time of experimentation (Supplementary Fig. 31), suggesting that this pathway is not active at this stage of development and therefore that the effects of LDN-193189 and K02288 are not attributable to disturbed activin signalling.

Exogenous recombinant mouse BMP4 (mBMP4; R&D Systems 5020-BP) was used to overactivate the BMP pathway[88]. This condition is referred to as BMP(+). Manipulations were applied for two different durations, from the early blastula stage (5 h post fertilisation, hpf) to either the late gastrula (10 hpf) or the one-pair-cirri larval stage (24–27 hpf). Two doses (high and low) of each manipulator were applied to the late gastrula experiments: LDN-193189 (4 and 8 μM), K02288 (250 and 500 nM) and mBMP4 (100 and 200 ng/mL). One dose was applied to the one-pair-cirri larva experiments: LDN-193189 (2 μM), K02288 (250 nM) and mBMP4 (200 ng/mL). Controls were run for each manipulator with the vehicle only (bovine serum albumin [BSA] or dimethyl sulfoxide).

## Functional transcriptomics

RNA sequencing was used to explore the impacts of BMP signal manipulation experiments on the transcriptome. RNA was extracted for three biological replicates of each condition (45 samples, Supplementary Data 21) using TRIzol and sequenced using the Illumina HiSeq 4000 platform (total 855 million read pairs). After performing quality control with FastQC (v0.11.5) and trimming with trimmomatic (v0.36), transcript abundances were quantified using kallisto (v0.43.0). Differential expression analysis was conducted using a bundled script in Trinity (v2.3.2)[195]. For datasets without biological replicates, edgeR was used with a fixed dispersion value of 0.1 to enable conservative inference[196]. For newly generated datasets with biological replicates, the voom method was applied to model the mean–variance relationship prior to linear modelling[197]. Transcripts were considered statistically differentially expressed with a false discovery rate (FDR) of less than 0.05. Pearson's correlation coefficient was used to assess the similarity of each condition's transcriptome. Putative BMP signalling downstream genes were identified based on their expression changes (at least fold-change > 2; $P < 0.01$). Genes that were upregulated in the BMP(+) condition and downregulated in the BMP(−) condition were classified as BMP-upregulated. Conversely, genes that were downregulated in the BMP(+) condition and upregulated in the BMP(−) condition were classified as BMP-downregulated. The transcriptomes of samples receiving high versus low doses of each manipulator were indistinguishable using hierarchical clustering (Fig. 4a) so are combined and treated as a single condition in subsequent analyses. Gene ontology analysis was performed on upregulated and downregulated gene sets using reciprocal best hits from BLAST searches against the Swiss-Prot database from UniProt[198]. To focus on the most robustly differentially expressed genes, we limited the dataset to genes that were differentially expressed in both the late gastrula and one-pair-cirri-larva experiments. Differentially expressed gene sets were searched for genes known to be involved in dorsal–ventral patterning in *Xenopus*.

## Transcriptome age index analysis

The phylostratigraphic age of each gene in the *L. anatina* genome was first estimated using GenEra (v1.2.0)[199,200]. GenEra reduces biases in age assignment by using DIAMOND (v2.1.8) to search the entire NR database (Supplementary Data 38). To improve the resolution of age assignment in the case of *L. anatina* we also added one genome assembly for each animal phylum that does not have a RefSeq annotated genome (Supplementary Data 39). This is especially important within the Spiralia where key phyla like Bryozoa and Phoronida are unrepresented. Using the estimated gene ages, the R package myTAI (v1.0.1.9000)[201] was used to calculate the transcriptomic age index[122] for several *L. anatina* transcriptomic datasets. These were (i) a developmental time course from fertilised egg to two pair cirri larva, (ii) *L. anatina* adult tissues (adductor muscle, lophophore, ovary, mantle and pedicle) and (iii) *L. anatina* late gastrula and larval stages with BMP signalling manipulation.

## Bacterial-cloning-free riboprobe preparation

To prepare DNA templates for RNA probe synthesis, a bacterial-cloning-free protocol was developed to maintain both rapid preparation and target specificity (Supplementary Fig. 32). This protocol initially uses gene-specific primers (F1 and R1, designed from transcriptomes) to amplify target sequences in the first PCR. The amplified sequences are then ligated into the pGEM-T Easy Vector to attach an RNA polymerase promoter site. Inserts in the reverse direction (antisense) to the T7 promoter site were further amplified using T7 and gene-specific forward nested primers (F2) in a second PCR. This nested PCR ensures both the specificity of the PCR products and the correct transcriptional direction of the inserts. The second PCR products were used for sequencing validation and in vitro transcription. No cloning-based screening is required to select target clones, allowing the entire procedure to be completed within one day.

Total RNA from various embryonic stages was extracted using TRIzol and cleaned up with the RNeasy Micro Kit (Qiagen 74004). cDNA synthesis was performed using the SuperScript III First-Strand Synthesis System (Thermo Fisher Scientific 18080051). Target sequences were amplified in a first PCR using gene-specific primers (F1 and R1) and EmeraldAmp GT PCR Master Mix (TaKaRa RR320A). The PCR conditions were: 94 °C for 2 min, followed by 30 cycles of 94 °C for 15 s, 55 °C for 15 s, and 72 °C for 1 min 30 s, with a final extension at 72 °C for 2 min. PCR products were analysed on a 1% agarose TAE gel stained with SYBR Safe DNA Gel Stain (Thermo Fisher Scientific S33102), and remaining products were purified using the QIAquick PCR Purification Kit (Qiagen 28104).

Purified PCR products were ligated into the pGEM-T Easy Vector (Promega A1360) using 5 μl of 2X Rapid Ligation Buffer, 0.5 μl of pGEM-T Easy Vector (25 ng), 3.5 μl of PCR product, and 1 μl of T4 DNA Ligase in a 10 μl reaction, incubated for 1 h at room temperature. A second PCR was performed using T7 and gene-specific forward nested primers (F2). The PCR conditions were: 94 °C for 2 min, followed by 35 cycles of 94 °C for 15 s, 55 °C for 15 s and 72 °C for 1 min 30 s, with a final extension at 72 °C for 2 min. PCR products were analysed on a 1% agarose TAE gel, and target bands were purified using the Wizard SV Gel and PCR Clean-Up System (Promega A9282). Primer sequences for probe synthesis are provided in Supplementary Data 40.

In vitro transcription was performed using T7 RNA polymerase (Promega RP2075) and DIG RNA Labelling Mix (Roche 11277073910). The reaction was incubated for 2–4 h at 37 °C, followed by DNase I treatment for 15 min at 37 °C. RNA probes were precipitated with 50 μl of nuclease-free water, 30 μl of 7.5 M LiCl, and 300 μl of 100% ethanol, chilled at −20 °C for at least 30 min, and centrifuged at 16,000 × g for 20 min at 4 °C. The RNA pellet was washed with 300 μl of cold 80% ethanol, air-dried, and resuspended in 25 μl of nuclease-free water. RNA probes were then diluted with 25 μl of 100% formamide and stored at −20 or −80 °C.

## In situ hybridisation for localising gene expression

Embryos were fixed overnight at 4 °C with 4% paraformaldehyde (PFA; Electron Microscopy Sciences 15714) in filtered seawater. Post-fixation embryos were washed with filtered seawater, dehydrated in 100% methanol, and stored at −20 °C. For rehydration, embryos were transferred from methanol into baskets with 40-µm nylon mesh, immersed in PBST (0.1% Tween 20 in PBS) in a 24-well plate, and incubated for 10 min with 1 ml per well. Permeabilization was performed in PBSN (1% NP-40 and 1% SDS in PBS) for 10 min, followed by PBSTX (0.2% Triton X-100 in PBS) for 10 min. Optional bleaching was done in 2% $H_2O_2$ in PBST at room temperature for 30–60 min under direct light. Embryos were washed in PBST for 5 min repeated three times, and rinsed with wash buffer (50% formamide, 5X SSC, 1% SDS, 5 mM EDTA, and 0.1% Tween 20). Prehybridization was conducted in hybridisation buffer (50% formamide, 5% dextran sulfate, 5X SSC, 1% SDS, 1X denhardt's, 100 µg/ml torula RNA, 5 mM EDTA, 0.1% tween 20, and 50 µg/ml heparin) at 60 °C for at least 1 h with slight rocking, with plates sealed in a plastic bag or covered with plastic wrap to prevent evaporation.

For hybridisation, the hybridisation buffer with probe (1:100–1:250, >100 ng/ml) was preheated at 70 °C for 5 min, then applied to the embryos and incubated at 60 °C overnight with slight rocking. Post-hybridisation washes were done at 60 °C for 15 min each with three washes in wash buffer, followed by washes with wash buffer + 2X SSCT (1:1), 2X SSCT, 0.2X SSCT, and 0.1X SSCT. Embryos were then rinsed with wash buffer at room temperature for 15 min per wash. Blocking was performed in MAB blocking buffer (100 mM maleic acid, 150 mM NaCl, 2% blocking reagent [Roche 1096176], 10% sheep serum, 0.1% Tween 20, and 0.2% Triton X-100) at room temperature for at least 1 h. Primary antibody incubation was carried out overnight at 4 °C using anti-DIG-AP solution (Roche 11093274910) in MAB blocking buffer. Embryos were washed in MABTX (100 mM maleic acid, 150 mM NaCl with 0.1% Tween 20 and 0.2% Triton X-100) five times for 20 min each, followed by two washes in TMN buffer (100 mM NaCl, 50 mM MgCl_2, 100 mM Tris-Cl, 0.05% Tween 20) for 5 min each.

For the chromogenic reaction, embryos were transferred to clean wells to avoid staining from impurities. The reaction was carried out using BM Purple (Roche 1442074) in the dark without rocking, ranging from 20 min to several days depending on the probe. The reaction was stopped by washing in PBST, fixing in 4% PFA/PBST for 20–30 min, washing in 100% ethanol for 5–10 min, and rinsing in PBST. Finally, embryos were mounted in 70% glycerol in PBS with 0.1% NaN_3 and stored overnight for full immersion at 4 °C, protected from light. The embryos were imaged using a Zeiss Axio Imager Z1 or Axio Imager A1 upright microscope with DIC optics.

## Immunostaining and imaging

Immunostaining was performed to identify the presence of phospho-Smad1/5 (pSmad1/5; BMP signalling readout) and phospho-Histone H3 (pHistone H3, indicator of mitotic cell at metaphase) during *L. anatina* development under control and BMP signal manipulation conditions. Embryos from early cleavage to larval stages were fixed in 4% PFA in filtered seawater, followed by dehydration in chilled methanol and storage at −20 °C. For immunostaining, embryos were first rehydrated with PBST for 10 min and then subjected to permeabilization using PBSTX for 30 min. To block non-specific antigens, embryos were treated with 3% BSA in PBST for at least 1 h. They were subsequently incubated with either rabbit anti-phospho-Smad1/5 (1:200; Cell Signalling 9511S) or rabbit anti-phospho-Histone H3 (Ser10) (1:100; Millipore 06-570) antibody in 3% BSA in PBST overnight at 4 °C. Alexa Fluor goat anti-rabbit secondary antibody (1:400; Invitrogen A-11037) was used for signal visualisation of the primary antibodies. For chitin detection, a fluorescein-conjugated chitin-binding probe (1:200; NEB P5211S) was applied. Nuclei were stained with Hoechst 33342 (1:1000

dilution from a 10 mg/mL solution; Invitrogen H1399), and cytoplasmic membranes were labelled using CellMask Deep Red (1:2000; Invitrogen C10046). Imaging was performed on a Zeiss LSM 780 or LSM 710 confocal microscope.

## Statistics and reproducibility

Two-sided *t*-tests were used for statistical comparisons as described in the figure legends. $P < 0.05$ was considered statistically significant. Exact *P*-values are provided in the figure legends or in the Supplementary data. Whole-mount in situ hybridisation and immunostaining experiments were performed using embryos from at least three independent fertilisation batches with similar results. For each in situ hybridisation or immunostaining condition, at least 30–50 embryos were examined and representative images are shown. For transcriptomic analyses, three biologically independent samples per condition were used. No statistical method was used to predetermine sample size. Embryos from the same fertilisation batch were randomly allocated into experimental groups. Blinding was not applicable, as BMP signalling manipulation produced visible phenotypic effects on embryonic development.

## Reporting summary

Further information on research design is available in the Nature Portfolio Reporting Summary linked to this article.

## Data availability

Genome sequencing and RNA sequencing datasets are accessible through NCBI BioProject under accession number PRJNA1068743. The *L. anatina genome* is made available on NCBI with the accession GCA_051362555.1 [https://www.ncbi.nlm.nih.gov/datasets/genome/GCA_051362555.1/]. A *L. anatina* genome browser with a BLAST function is available at https://marinegenomics.oist.jp/lan_3_0/viewer/info?project_id=124. The genome sequence, gene models, and functional annotations from UniProt, Pfam, InterProScan, EggNOG and KOfam are also available at this location. Genomic, transcriptomic and imaging data from this study, including the genome assembly, gene annotations, repeat annotations, gene expression matrices and original Zeiss LSM imaging files, are available on Zenodo (https://doi.org/10.5281/zenodo.16916709). Source data are provided with this paper.

## Code availability

Code for analyses in this study is available on GitHub (https://github.com/symgenoevolab/lingula_genome) and Zenodo (https://doi.org/10.5281/zenodo.18591213).

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

## Acknowledgements
We acknowledge the permission from Amami City, Kagoshima, Japan, for collecting *L. anatina* specimens. We thank Ryo Koyanagi in the OIST DNA Sequencing Section for his support of RNA sequencing and Asuka Sentoku for support with the embryonic experiment at the University of Ryukyu. We also thank Keisuke Nakashima for a gift of chitin-binding probes and Jr-Kai Yu for support with microscopy. We are grateful to Konstantin Khalturin for facilitating the shipment of fixed embryos from Japan to the USA and to D. Marcela Bolaños for handling the shipment from the USA to Taiwan. This work was supported in part by multiple funding sources over the past decade, including a Japan Society for the Promotion of Science Grant-in-Aid for JSPS Research Fellows (15J01101), a Royal Society Newton International Fellowship (NIF\R1\201315), an Academia Sinica Career Development Award (AS-CDA-112-L06), an Academia Sinica Grand Challenge Program Seed Grant (AS-GCS-114-L08), and National Science and Technology Council Research Project Grants (113-2311-B-001-026 and 114-2311-B-001-024-MY3) to Y.-J.L., T.S.

was supported by a JSPS Overseas Research Fellowship. Y.H.W. was supported by the Innovation Team Project of Universities in Guangdong Province (2023KCXTD028) and the National Natural Science Foundation of China General Program (42276104). We thank the Symbiosis Genomics and Evolution Lab members for their assistance and support, and three anonymous reviewers for their insightful comments.

## Author contributions
T.D.L., Y.-J.L. conceived the project. K.S., K.E., N.S. and Y.-J.L. collected specimens. Y.H.W. sequenced the chromosome-scale genome and tissue transcriptomes. T.D.L. and M.-E.C. annotated the genomes. Y.-J.L. conducted embryonic experiments and functional transcriptomics. T.S., K.S., L.-J.K., Y.-L.C. and Y.-J.L. performed whole mount in situ hybridisation. T.D.L., I.J.-Y.L. and Y.-J.L. prepared code for GitHub. K.H. prepared the genome browser. T.D.L. and Y.-J.L. analysed data and wrote the manuscript. K.E., P.W.H.H. and Y.H.W. discussed the results and edited the manuscript. All authors contributed to the revision of the manuscript.

## Competing interests
The authors declare no competing interests.

## Additional information
**Supplementary information** The online version contains Supplementary material available at https://doi.org/10.1038/s41467-026-70403-5.

