## [Peer Review File · Nature Communications]

Brachiopod genome unveils the evolution of BMP signalling in bilaterian body patterning

Corresponding Author: Dr Yi-Jyun Luo

Version 0:

Reviewer comments:

Reviewer #1

(Remarks to the Author)

Lewin et al. present a chromosome-scale genome of the brachiopod *Lingula anatina* and show that 4 out of 10 *Lingula* chromosomes are results of fusion-with-mixing. By comparing the genomes of a mollusc, an early-branching annelid and *Lingula* they find evidence for lineage-specific fusions. Lewin et al. then focus on BMP signaling components, since the role of BMP signaling in the DV patterning of *Spiralia* is insufficiently well-understood. They show that BMP2/4 and Chordin are expressed on the opposing sides of the DV axis (dorsally and ventrally, respectively), and demonstrate a pSMAD1/5 gradient with a dorsal maximum, similar to the situation in the mollusc *Lyanassa* and non-spiralian bilaterians. Given the diversity of the ways spiralian pattern their DV axis, this manuscript provides important support for the hypothesis that Chordin-BMP-dependent DV patterning was an ancestral feature of spiralian development shared not only by *Spiralia* but by Bilateria as a whole. It is an important paper, and I strongly recommend publication, however, there are several points, which need to be resolved first.

Major points

1. Both, in the Introduction and in the Discussion, the Authors give a misleading, oversimplified view of the BMP signaling pathway, which is reflected in their Fig. 2b and Fig. 2f, and which causes some obvious misinterpretation of the results. The Authors seem to think that BMP2/4 (=Dpp) is the only BMP responsible for the DV patterning. This is not the case.

First, BMPs function as dimers, and usually the DV-patterning-relevant ligands are the BMP2/4/BMP5-8 heterodimers. So, BMP5-8 is not an "alternative BMP paralog" relevant only for the leech. It is involved in DV patterning across Bilateria and in the directive axis patterning in anthozoan Cnidaria. All the models Lewin et al. mention in the Introduction use BMP5-8 together with BMP2/4. In *Xenopus*, BMP7 (belongs to the BMP5-8 group) is expressed ventrally together with BMP4; in *Drosophila*, Screw (=BMP5-8) is expressed ubiquitously, while Dpp is dorsal; in leech, BMP5-8 is dorsal and BMP2/4 is ubiquitous etc.

Second, the Authors repeatedly call ADMP a BMP antagonist, but it is not. It is a BMP ligand (Reversade and De Robertis, 2005). And this misunderstanding probably made Lewin et al. overlook their own most striking result – they showed a "BMP seesaw" in a spiralian without noticing it. In anthozoan Cnidaria as well as in ambulacrarian and chordate deuterostomes, the second body axis is patterned by a "BMP seesaw". Both ends of the second body axis express BMPs, but BMP signaling takes place on the end of the axis, which is opposite to Chordin. For example, in *Nematostella*, one end of the directive axis expresses BMP2/4, BMP5-8 (BMPs) and Chordin, and the other end expresses GDF5-like (BMP). BMP signaling is on the GDF5-like side. In sea urchin, BMP2/4, ADMP1 (BMPs) and Chordin are expressed ventrally, and ADMP2 (BMP) - dorsally. BMP signaling is dorsal. In the frog, BMP4 and BMP7 (BMPs) are ventral, BMP2, ADMP (BMPs) and Chordin are dorsal. BMP signaling is ventral. Lewin et al. see exactly the same in a spiralian, with dorsally expressed BMP2/4 and BMP5-8 (BMPs), and ventrally expressed ADMP, BMP3 (BMPs) and Chordin and pSMAD1/5 opposite to Chordin. This is absolutely fantastic and has to become the focus of the revised manuscript. In the revision, the Authors must show pSMAD1/5 stainings and in situ of Chordin and all BMPs expressed at gastrula (BMP2/4, BMP5-8, BMP3, ADMP) in controls and upon up- and downregulation of BMP signaling on one main text figure – not scattered between the main text and extended figures – and make a really big evolutionary point about the "BMP seesaw" being potentially the ancestral DV patterning mechanism in the Discussion (stressing the similarity to the mollusc data doi:10.1093/molbev/msab322 and to non-spiralian data). In contrast, *Bambi* and *CV2* in situ are, of course, good to have – as examples of conserved BMP modulators expressed in the "high BMP signaling" region across species, but they can easily go into the supplement. However, in situ quality needs to be improved, especially for BMP2/4, BMP5-8 and *CV2*, where signal-to-noise ratio is

unacceptable. The Authors should maybe consider clearing their embryos with one of the recently published protocols e.g. <https://www.science.org/doi/10.1126/sciadv.aba0365> to improve the quality both of the in situs and of the morphological description shown in Extended Data Fig. 7.

2. If the Authors publish a chromosome-scale genome, they should also make it easy to use for the community. I understand that this is a lot of work, but a publicly accessible *Lingula* genome browser with a built-in BLAST option rather than a link to a bioproject with raw reads would be great.

Other comments

1. Line 81-88. The "typical spiralian" argument is not ideal. Among high-level spiralian taxa, spiral cleavage exists in mollusks and annelids, however, as far as I am aware, no lophophorates (bryozoans, phoronids etc) exhibit spiral cleavage. Flatworms or nemertines also do not have spiral cleavage. Most gnathiferans (chaetognathes, rotifers etc.) also do not have spiral cleavage, but gnathostomulids have it. Given that, I do not think we can state with certainty that spiral cleavage was lost multiple times or gained multiple times in Spiralia (as the Authors do by saying "possibly reflecting a reversion to the ancestral state"). On the other hand, some spirally cleaving molluscs use BMP for DV patterning, and spirally cleaving annelids maybe use it as well - for example *Owenia* has Chordin and BMP2/4 and likely uses it for the DV patterning. In general, I suggest that instead of explaining why other models are not ideal, the Authors should argue that with the increased sampling across the spiralian phylogeny, we get a better understanding of the ancestral mode of BMP signaling in Spiralia.

2. Fig. 1c. How did the Authors get the list of BMP signaling component orthologs for Fig. 1c? Was it BLAST or reciprocal BLAST or something else? How certain are the Authors that lophotrochozoan BMP3 is an ortholog of chordate BMP3? I may have easily missed something, but if I am not mistaken, the only published non-chordate putative BMP3 comes from the ctenophore *Mnemiopsis*, however, its phylogenetic position as a BMP is not robust (compare Fig. 2 in doi: 10.1371/journal.pone.0024152 and Supplementary Fig. 2 in doi:10.1242/dev.141507). If BMP3 is indeed always present in Lophotrochozoa, it is an important piece of information. A phylogenetic tree with BMP ligand distribution across Bilateria (including ADMP, GDF5-6 etc.) would be a very useful supplementary figure. It would also be very useful to add *Xenopus* to Fig. 1c either in addition or instead of human, since this is the animal authors constantly refer to in the text.

3. Fig. 1d. I assume the Authors did not do this for space reasons, but they should add *Owenia* and *Linneus* data on Fig. 1d. Having them separately on Extended Fig. 6a-b is very confusing, especially since the order of the chromosomes is always different for each species on each figure. If having amphioxus, *Pecten*, *Lingula*, *Owenia* and *Linneus* on one panel is impossible (although I do not see why it should not be!), and the Authors want to keep their *Pecten* + *Lingula* + "1 more species in the middle" comparisons, they should fix the order of *Pecten* and *Lingula* chromosomes, adjust the third species to this fixed order and put all these figures as panels on the main text figure.

4. For ALG association of BMP genes, the result is quite striking, but I may have overlooked a careful explanation of how random genes were selected. It is important and must be clearly explained in the main text. Also, the Authors should comment on whether the chromosomal linkages of BMP signaling components are preserved in highly scrambled genomes (*Drosophila*, *C. elegans*, clitellate annelids, tunicates etc.) or genomes which underwent more translocations than, say amphioxus or *Pecten* - e.g. human genome, *Streblospio benedicti*, *Carcinoscorpius rotundicauda*.

5. Lines 239-240 and Fig 2a. The Authors write that "The co-option of BMP-Chordin signalling pathway into dorsal-ventral axis patterning is a cornerstone of the bilaterian body plan". While I fully agree that BMPs and Chordin are important, I am not sure about the co-option. Co-opted from doing what? Co-option means that BMP and Chordin were doing something else in the bilaterian ancestors and then were re-deployed in DV patterning. If their function in DV was re-deployment, what was the ancestral function? Also in the Fig. 2a, the Authors suggest that "The BMP-Chordin axis predates the origin of bilaterians and, therefore, the dorsal-ventral (DV) axis." I am not sure how the Authors make this conclusion. What do they call a BMP-Chordin axis? A body axis patterned by BMP and Chordin? Their tree on Fig. 2a suggests that it evolved prior to the split of Cnidaria and Bilateria. In that case, cnidarian second body axis (directive axis of anthozoans) is homologous to the DV axis of Bilateria, and it should be therefore called DV axis as well. However, it is not entirely certain whether directive and DV are homologous or evolved convergently. If it is convergence, then Lewin et al. cannot say that "BMP-Chordin axis predates the origin of Bilateria" - in this case, these are different body axes, which just happened to use BMP signaling for patterning. I suggest removing the term "co-option" and carefully rephrasing the legend.

6. Fig. 2b. I understand that the Authors tried to keep things simple, however, not only do BMPs signal as dimers (see major point 1) but also the transduction of the signal is performed not by a type I BMP receptor but by a heterotetrameric complex containing two type I BMP receptors and two type II BMP receptors. Additionally, BMPRI is located within the cell in the schematic, however the receptor complex should be positioned on the cell surface instead.

7. Fig 2c. What is the Author's definition of the "main BMP signalling ligands, mediators, and modulators". I would recommend either including the full heatmap of *L. anatina* BMP pathway components shown in extended data Fig. 8a or restricting the listed components in Fig 2c to only the BMP ligands (including ADMP) and Chordin, as these are the focus of the paper.

8. Fig. 2d-e. It is not entirely clear why these have to be main text figures. It is nice to show that *Lingula* SMAD1/5 is really a SMAD1/5 (Fig. 2d), and it is excellent that the Authors demonstrate that the epitope recognized by the anti-pSMAD1/5 antibody used in the paper is present in *Lingula* SMAD1/5 (Fig. 2e), however, these are "technical details", which should go into the supplement. Also, the Authors start using the term R-Smad without explaining what it is. They must write that Smad1/5 is the R-Smad in the BMP signaling (and Smad2/3 is the R-Smad in the Nodal/TGF β signaling) - otherwise unclear.

9. Fig. 2f. I find this very confusing. First, the signal is normally transduced not by a Smad1 homotrimer as the Authors depict on the image but by a heterotrimer containing two Smad1/5 molecules and one Smad4 (Co-Smad) because such heterotrimer is energetically more stable than a homotrimer. Second, I do not understand why they point out the L3 loop - they never discuss this in the text, and this is not the site where their antibody binds. L3 should either be removed from the image and legend or elaborated upon in the text. If no additional discussion of the Fig. 2f content is added, I suggest its removal from the paper as it does not contribute significantly to the main narrative.

10. Fig. 3a. Why were two treatment durations used? This is not clear and must be explained in the main text. In addition, it must be made clear in the main text or figure legend of each experiment throughout the entire manuscript which treatment regime was used.
11. Fig 3, Fig 5, extended data figure 9. Sample size and phenotype penetrance should be provided for all immunostainings and RNA in situ hybridisations (i.e. X embryos out of a sample of Y develop the phenotype shown on the representative image).
12. Fig 3c-f. It is mentioned briefly in the main text that manipulation of BMP signalling delayed gastrulation. Do the authors have data regarding the extent to which gastrulation is delayed? Is this the case for all individuals? Do the animals eventually complete gastrulation?
13. Fig. 3g-j. Why are 2 different doses of inhibitor/recombinant BMP used? Which dosage were the animals shown in panels g-j exposed to? It must also be made clear in the figure legend or main text whether the animals used in experiments shown in fig. 3c-f and 3k-n were treated with high or low doses. This information must also be made clear for all experiments throughout the manuscript that made use of the BMP inhibitors/recombinant BMP.
14. Fig. 3k-n. This needs better (larger and better annotated) pictures. Also, the Authors had two types of treatment - do they see the chitin phenotype in both or only in the treatment spanning longer? They have to comment on whether there are any differences between the treatments.
15. Lines 376-377. GO terms are useful but also very imprecise. I think that calling these genes "neuronal" is a BIG stretch. I do not know which alk the Authors mean - for me *Alk2*, *Alk3/6*, *Alk4/5/7* are different type I BMP or TGF β receptors, which cannot be called "neuronal". *FoxB* is primarily an endomesodermal marker. *Lhx1/5* has functions in the neurons but also in the development of the lymphoid tissue. *Netrin* is an axon guidance molecule but has other functions as well. *Nkx2.4* is expressed in the brain, but predominantly in the testes. Other genes the Authors mention here can probably be called neural without too many additional disclaimers. I do not doubt that in some context all these genes are expressed in neurons, but I think that this bit needs some very careful re-phrasing. E.g. "dominated by genes demonstrating either exclusively neuronal expression or known to be expressed in neurons in specific developmental contexts in other model organisms". I understand that in situ validation of neuronal expression of all these genes in *Lingula* maybe too much for this paper (although it would be great to have it), but without it the argument has to be tuned down and carefully phrased.
16. Fig 4g. *BMPR* should be positioned on the plasma membrane, not inside the cell.
17. Fig. 5a. The Authors should replace *BMP5* with *BMP7* in the *Xenopus* ventral centre, add *BMP2*, and re-color ADMP in the dorsal centre (ADMP is not a BMP antagonist! It is a BMP!).
18. Lines 480-490. The section on TAI analysis is confusing. It is unclear how the results of this analysis indicate that BMP signals are crucial for gastrulation (Line 488). Is it because of the GO terms associated with the gastrulation process? The delayed gastrulation observed in Fig. 3 is more convincing evidence that BMP signalling is important for gastrulation. In addition, the description of the role of BMP signalling as "crucial" for gastrulation is too strong, as it is previously stated on line 319 that BMP manipulations only delay gastrulation, and not that BMP inhibition prevents it.
19. Supplementary fig 5a. In the text, late blastula and mid gastrula stages are highlighted as stages with the highest TAI score but the Authors refer to a figure highlighting a different stage – the late gastrula, which has the lowest TAI.
20. Supplementary figure 5b. It should be made clear what "BMP related" genes are. Are these genes that are members of the BMP family, or are they components of the BMP pathway?
21. Lines 491-499. This needs to be re-written once my Major point 1 has been addressed. Also more data needs to be presented to confirm that *BMP2/4* and *BMP5-8* expression on the dorsal side is suppressed by BMP signaling. It is counterintuitive, and if true, very interesting. However, it is not sufficiently well demonstrated. High quality in situ and QPCR are necessary upon up- and downregulation of BMP signaling.
22. Lines 512-513. Hemichordates and sea urchins are not "basal deuterostomes". Together, they are Ambulacraria and a sister group to Chordata. If the Authors wish to take sea urchins and hemichordates separately, then Chordata would be an earlier branch of Deuterostomia than either Hemichordata or Echinodermata.
23. Fig.6. The BMP gradient depicted (pink) should be much narrower. The width of the pink shaded region better reflects the region of *Dpp* expression than the nuclear *pMad* gradient (Raftery and Sutherland, 2003).
24. Fig 6. In the legend, the blue region on each embryo cartoon is described as the "anti-BMP domain". This is inaccurate/vague since this area expresses its own BMP ligands and it should just be referred to as the chordin/sog expression domain.

Reviewer #2

(Remarks to the Author)

BMP signaling is involved with establishing the dorsal-ventral axis and a region of neuroectoderm in several taxa in deuterostomes and ecdysozoans. However, experiments testing the function of BMP signaling in these processes across spiralian taxa have found differing levels of involvement. To understand whether BMP signaling was ancestrally involved in forming the dorsal-ventral axis including the neuroectodermal domain in bilaterians, the authors examined BMP function in the spiralian *Lingula anatina*, a brachiopod that develops by radial rather than the ancestral spiral cleavage. Unlike some other spiralian taxa, *L. anatina* appears to have some of the same regulatory logic of the BMP signaling gradient in chordates, including an anti-neural function for BMP. The authors also generated a chromosome-level assembly for *L. anatina* and reconstructed the complement of BMP pathway components for multiple spiralian taxa. The paper is well-written and appropriate for a broad audience, the data are of high quality, and the experiments and findings are important for understanding how the dorsal-ventral axis including the neuroectoderm evolved in bilaterians.

Main comments/suggestions

The authors argue that the radial cleavage present in *L. anatina* may be a 'reversion to the ancestral state'. However, this

still raises the question of the evolutionary history of BMP signaling in spiralian. Since spiral cleavage was likely ancestral for the clade containing brachiopods, annelids, and mollusks, then BMP signaling was first used in the context of spiral cleavage. It would be interesting for the authors to comment on how they think BMP signaling was used to form the dorsal-ventral axis in spirally-cleaving embryos and was then re-evolved to form the dorsal-ventral axis in a radially-cleaving embryo. Why do you think there is such variability in the involvement of BMP signaling and dorsal-ventral axis formation across different spiralian taxa? I think the discussion points on lines 563–570 are very interesting, and begin to get at this question, but I would like the authors to comment on this in the context of gain and loss of spiral cleavage. In *L. anatina*, BMP signaling does not appear to be active (pSMAD1/5 levels) during cleavage stages, unlike other taxa with spiral cleavage, but instead is active in the blastula. To really demonstrate a lack of BMP signaling during cleavage, I think it's important to test function of BMP during this time window (see additional comments for Fig. 2 below).

The regulation of *bambi*, *cv2*, *admp*, and *chordin* by BMP signaling in *L. anatina* (Fig. 5) is interesting. The authors suggest that this is indicative of a BMP signaling gradient controlling fates along the dorsal-ventral axis as in chordates, and that this system may have been present in the last common ancestor of the two groups (e.g., lines 450–451). An alternate interpretation of the functional transcriptomic data is that a gradient of BMP signaling was ancestral and that regulatory logic is maintained in *L. anatina*, irrespective of what developmental program (e.g., fates along the dorsal-ventral axis) this signaling system controlled. I think that ventralization or dorsalization (or loss/gain) of multiple fates after drug treatments or BMP protein treatment in *L. anatina* needs to be demonstrated more clearly to support the hypothesis of an ancestral function in dorsal-ventral axis formation. The change in expression domains of *bambi*, *cv2*, *admp*, and *chordin* after perturbing BMP signaling does not necessarily indicate a shift in fates along the forming dorsal-ventral axis, especially since many of the other Spemann-Mangold organizer genes were not found to be regulated by BMP signaling in the same direction as in *Xenopus* (e.g., lines 476–478).

In *L. anatina*, does the neuroectoderm form on one side of the trunk ectoderm during gastrulation, and are the differentially-regulated genes expressed in the neuroectoderm? It looks like the references for the sentence on lines 375–377 are for other animals. For Fig. 4, I would like to see spatial expression by in situ hybridization of some of the differentially expressed neural genes identified (e.g., *elav*, *pax3/6*, *soxB1*) in controls and before and after manipulation of BMP signaling.

Additional comments and questions

Intro

line 52 Capitalize “ADMP” since it's an abbreviation for “anti-dorsalizing morphogenetic protein”.

lines 55–56 “Basal deuterostomes” would be the nodes before the terminal/extant taxa. I think you mean “early-branching deuterostomes”.

Results and Figures

Fig. 2

In panels m, n, and o, there are some cells where the pSMAD1/5 labeling appears to only be on one side of the nucleus. Is this common for pSMAD1/5 labeling and if so, what does it indicate? I'm asking about the cells with a circular domain of labeling, not the cells with punctate spots of pSMAD1/5.

The view in panel k is not labeled (av?).

Is vegetal is down in panel l?

For panel o, I am not certain what I am looking at. Why is there a stripe of pSMAD on the ventral side of the larva? Is this a lateral view similar to Fig. 3k? A diagram of the tissues including the forming mantle and pSMAD in an early larva would be helpful.

Why do you think there is no pSMAD1/5 at cleavage stages even though there is maternal *bmp5-8* and *smad1/5* transcript and ubiquitous *bmp* receptor expression? Did you try blocking BMP signaling with the two drugs during cleavage stages (i.e., a drug pulse during early cleavage and assessment of resulting larval phenotypes)? Similarly, if you add BMP protein during cleavage stages, do you see pSMAD during these stages? Is it possible that there is a lower level of BMP signaling during cleavage that is not detected by the cross-reactive antibody? Is *bmp5-8* transcript spatially restricted to a subset of cells during cleavage?

Fig. 3

For the BMP manipulation experiments, the long time-window went from 5 hpf to ~24-27 hpf. What was the evidence that the BMP protein or the BMP inhibitors were active for that whole time in *L. anatina*? Was the BMP protein or BMP signaling inhibitors only applied once at 5 hpf? It would be useful to show pSMAD1/5 at early larval and 1 PCL stages for the long treatment times.

For the two BMP protein doses (low and high) and the two drug doses, did you test whether these doses resulted in different levels of pSMAD1/5 in *L. anatina*? Are the data shown in Fig. 3d–n after high dose treatment for the protein and the drugs? I don't see data comparing the different doses except effect on pH3, and the change in number of pH3+ cells between the low and high treatments does not look very dramatic compared to the shift from no treatment to the low treatments.

How was the overall morphology including the dorsal-ventral axis affected by up or downregulating BMP signaling? You

state that gastrulation was delayed and that mantle lobe folding was disrupted, but a description of additional phenotypes would be useful for understanding how manipulating BMP signaling affected development and formation of fates along the dorsal-ventral axis.

What is the view in Fig 3n?

It would be helpful to include a sentence explaining where in the BMP signaling pathway LDN-193189 (LDN) and K02288 (K02) act. Since both drugs inhibit multiple ALK receptors, what is the evidence that the dose you are using is selective for BMP signaling and not Activin signaling in *L. anatina*? Did you test the effects on pSMAD1/2? Similarly, for the BMP protein treatments, did you test whether you were cross-activating Activin signaling (SMAD1/2)? It's important to demonstrate the selectivity for perturbation of only BMP signaling since Activin signaling has been shown to be an organizer signal in at least one other spiralian.

Fig. 4

For the functional transcriptomic experiments, did you find differences in which genes were up and down-regulated between the 100 and 200 ng/mL BMP treatments or between the low and high drug concentrations? Do you think these low and high concentrations are both maximally upregulating/blocking BMP signaling, or did you find evidence for BMP acting as a morphogen in *L. anatina*?

Fig. 5

In panel e, the z-scores for some genes (e.g., *shh*, *twsg1*, *bmp5*, *bmp4*, *fst*...) seem to vary across replicates within a treatment. Is this due to low expression levels for these genes?

In Extended Fig. 9, the up-regulation of *bmp2/4* expression in the ectoderm after LDN treatment is not very clear to me in the animal shown.

Discussion

Be careful of inferring gene function in *L. anatina* based on what a gene homolog does in chordates. For example, in lines 515–516, you discuss “ventralising factors, particularly chordin”. However, the function of chordin in *L. anatina* has not been tested to my knowledge.

Methods

line 589 It looks like multiple tissues were collected for genome sequencing, but only mantle, lophophore, and adductor muscle were used. Was the DNA in the other tissues degraded?

line 596 For genome sequencing, did you assay DNA integrity using BioAnalyzer in addition to assessing purity with a NanoDrop?

line 601 & 738 Did you do any column purification after RNA extraction by TRIzol?

line 742 Why was edgeR used versus DESeq2, and what were the criteria for using a dispersion parameter of 0.1?

Reviewer #3

(Remarks to the Author)

Reviewer #4

(Remarks to the Author)

The role of BMP signaling in patterning the dorsoventral body axis has been described in a diversity of species, including deuterostomes, ecdysozoans and spiralian, and is widely considered to be the ancestral bilaterian state. While initial studies in a handful of model organisms, primarily in vertebrates and insects, suggested BMPs are required to inhibit neural induction in early embryogenesis, recent work in emerging model systems, particularly those found in the Spiralia, suggest a more complex story where neural induction is independent of BMP signaling in multiple species (e.g. Lambert et al., 2016; Webster et al., 2021; Webster & Meyer, 2024).

In the manuscript “Brachiopod genome unveils the evolution of the BMP-Chordin network in bilaterian body patterning” Lewin et al., present a chromosomal-scale genome for the brachiopod *Lingula anatina*. This genome improves on a previous scaffold-level assembly for the same species and extends observations from this previous publication from a number of these authors that BMP patterning may be involved in shell formation in brachiopods. While the chromosome-level genome will prove to be a key resource for future work in this clade, the study largely focuses on the role of BMP signaling in this species. The authors employ the genome to describe the complement of BMP signaling pathway members in this species as well as other spiralian. The experiments presented, including pharmacological approaches and RNA seq, confirm a role for BMP in patterning the dorsoventral body axis, which has been suggested in the previous genome description and was previously described in other spiralian, including two other brachiopod species, *T. transversa* and *N. anomala* (Martin-

Duran et al., 2016).

Based on differential expression in RNAseq datasets generated from pharmacologically up and down regulating BMP signaling, the authors suggest that BMP also inhibits neural development in *L. anatina*. They find dozens of genes to be differentially regulated upon manipulation of BMP signaling, including a number of neural-associated genes. Based on this finding, the authors conclude that BMP is required to inhibit neural development, a role well described in vertebrates and arthropods. However, the experimental evidence for this assertion is not sufficient, and indeed, expression patterns for several neural genes contradicts this hypothesis. For example, *elav* is unchanged or even slightly downregulated in BMP(-), while *soxB1* is upregulated in both BMP(-) and BMP(+) embryos. Furthermore, the (differential) expression of neural genes should also be confirmed using in situ hybridization assays to visualize where these genes are expressed in relation to the BMP signal and potential changes in expression domain after treatment.

Throughout the manuscript, the authors discuss the roles of BMP in DV-axis patterning and in neural development as one common feature, with one being dependent on the other. However, recent work in other spiralian indicates that these can be two distinct roles that can be present independently of each other. The authors imply that because they see an effect on dorsoventral patterning, a role in neural development must be the ancestral state and vice versa. This discussion would benefit from clear delineation and test of these different roles, and would greatly benefit from an attempt to differentiate these functions experimentally. Indeed, recent work in spiralian suggests a more complex story for the role of BMPs in neural development, with functional studies in both annelids and molluscs demonstrating neural development independent of BMP signaling. This needs to be discussed in more detail, taking into account different scenarios like convergent evolution or loss of traits.

Some additional concerns and comments:

1. The role of Chordin: The title and abstract of the manuscript refer to the role of the BMP-Chordin network. While the manipulation of BMP signaling seems to affect expression levels of Chordin (whether directly through transcriptional regulation or indirectly through changes in size of the ventral domain is also not addressed), the actual role of Chordin in body axis patterning and neural development is not directly investigated.

2. Expression of BMPs and pSMAD signal

Using antibody staining, the authors detect phosphorylated SMAD1/5, a read-out for BMP signaling activity, at the dorsal side of the embryo (Figure 2). In Extended Data Figure 9, they also show the expression of *bmp2/4*, *bmp3* and *bmp5-8*, using ISH. This expression data should be included in Figure 2. The expression data for *bmp2/4* and *bmp5-8* expression raises concerns. While there is a dorsal signal that likely overlaps with pSMAD1/5 signal, *bmp2/4* and *bmp5-8* also seem to be expressed more globally throughout the embryo. Can the authors provide an explanation for this, or ISH images that are easier to interpret for non-brachiopod specialists?

3. Role of BMP in cell proliferation

In Figure 3, the authors describe an effect of BMP manipulation on cell proliferation. This observation should be discussed in more detail in the manuscript. What cell types are affected by this role? How does this compare to known roles of BMP signaling in other systems? How could this be related to other observed phenotypes?

4. Role of BMP in shell formation

In Figure 3, the authors describe a role of BMP signaling in establishing the embryonic shell fields. They conclude that BMP is required to divide the shell field into two valves, as the fields seem to be fused in BMP(-) embryos. However, in BMP(+) embryos the shell field seems to be expanded and experience over-folding, rather than lost as might be expected if BMP indeed inhibits shell field formation. How can these results be explained?

5. RNA sequencing

The RNA sequencing data require further confirmation and discussion: 1) How much overlap is there between DEGs in BMP(+) and BMP(-) embryos? Are the same pathways/genes affected? 2) Is there evidence in the RNA sequencing data for the described role in cell proliferation and shell formation/folding? What are affected signaling pathways? 3) How reliable is GO term analysis in this species? 4) As mentioned above, the changes in neural gene expression require further confirmation/investigation (see above). 5) The authors describe two different incubation windows used for BMP manipulation. It is not clear which one of these was used for RNA sequencing.

6. The Lophophorata hypothesis: The authors present a phylogenetic analysis in Fig1b that suggests that brachiopods are in a clade with phoronids and bryozoans to form the "Lophophorata". While this is a clade proposed in historic analyses, recent phylogenomic work suggests that bryozoans may be sister to entoprocts or cyclophorids, which the authors suggest may be the result of including orthologs with lower phylogenetic signal. However, these results are difficult to interpret as the authors did not include entoprocts or cyclophorids in their analysis. Could the authors search for their gene set in the deposited data for these species to provide support for their hypothesis? Or, could they leverage their new chromosome-level assembly could be to search for shared derived syntentic relationships between brachiopods and bryozoans that might support their hypothesis?

7. Terminology

The authors make references to "basal deuterostomes" throughout the manuscript. This should be corrected to "basally-branching deuterostomes" or referred to as the phylogenetic group specifically (e.g. ambulacrarians).

8. Does the proposed karyotype comport with expectations in this group?

Other minor comments:

Main Figures

Figure 1, Panel D: Why is the Hox cluster pointed out specifically? It is hardly mentioned in the text. The lower half of this panel should be discussed and explained in the figure legend.

Figure 1, Panel E: This panel requires more explanation.

Figure 2, Panel D-F: These could go into the supplement.

Figure 2, Panel G-O: Labeling of body axis and potentially an embryo schematic would be helpful here. Expression of BMPs should be added to this Figure.

Figure 3, Panel K-N: Overlaying the chitin staining with the pSmad staining would be helpful to confirm that BMP signaling is active in the Chitin-negative regions.

Figure 3, legend c-f: What do the empty arrowheads indicate?

Figure 3, legend k-n: What does the arrowhead indicate?

Figure 4, Panel A-H: Panels A and G-H require additional explanation or should be presented in a more intuitive way. Panels B-D do not provide crucial information and could be moved to Supplement.

Figure 4, Panel A: Pearson correlation between ctrl and BMP(-) conditions show surprisingly little difference. How can this be explained?

Figure 4, Panel A: What are the different samples that each column and row represents?

Figure 4, Panel C-D: This could go into supplementary figures.

Figure 5, Panel B: Body axes should be labeled for easier understanding. For *admp* and *bambi* the difference between control and BMP(-) embryos is not very clear. A quantification or at least a better representative image should be provided.

Extended Data

Figure 7: This figure is not mentioned in the text.

Figure 9, Panel A: ISH images are difficult to interpret. While a *bmp2/4* signal is present in the dorsal part of the embryo where pSmad was detected in Figure 1, it also seems to be expressed in other parts of the embryo?

Figure 9, Panel B: Where is *bmp2/4* marked in the schematic?

Figure 9, Panel C: This requires further investigation. Can this be confirmed with the RNAseq data? Is the effect on *chordin* expression direct or indirect because the ventral domain is lost?

Figure 9, Panel D: What experimental evidence is the distinction between ectoderm and endomesoderm based on?

Figure 9: Please provide more explanation for the arrows and pathways in the Figure legend.

Supplementary Data

Figure 4: Why was the tree only produced for the Smad family? At the very least this should be produced for all major proteins discussed in this study (i.e. BMP family, *Chordin*, selected neural genes, dorsoventral markers). Also: add description of species abbreviations.

Text

Page 1, Line 30: "We uncover a BMP signaling gradient..." - The presence of a signaling gradient was never actually discussed/shown. For BMP only mRNA expression was shown, and not the distribution of protein. We appreciate the challenges of working with emerging model systems, but this should be clarified in the text. The pSmad antibody could be used as a read-out for a signaling gradient, but this would need to be discussed in more detail.

Page 2, Line 52: "...In the fruit fly *Drosophila*, the BMP gradients is completely inverted..." - This is true for invertebrates in general.

Page 9, Line 244: "...Extended Data Fig. 7..." - This should probably refer to Extended Data Fig. 8?

Page 11, Line 306: "...underscoring a deeply conserved BMP gradient..." - See above. The use of the word gradient should be justified.

Page 11, Line 310: "...we applied [...] inhibitors and [...] proteins for two durations..." - The reasoning for selection of these stages/durations should be explained.

Page 11, Line 314: "...LDN-193189 (LDN) and K02288 (K02)..." - What is the difference between these two inhibitors? Why were two used?

Page 11, Line 316: "...led to a loss of dorsal pSmad1/5..." – From the images presented, it looks like the dorsal signal persists, and additional signal is gained throughout the rest of the embryo.

Page 11, Line 319: "...altered gastrula morphology..." - The authors should describe how gastrula morphology is altered and what can be concluded for the role of BMP signaling in the process.

Page 13, Line 371: "Gene ontology (GO) analysis [...] highlighted the 'nervous system development' ..." - How reliable is GO term analysis in this species? Phylogenetic trees should be presented, at least for the main protein families of interest to confirm this.

Page 17, line 477: "...organiser genes *cer1*, *noggin* and *shh* are BMP-upregulated when they might be expected to be downregulated..." - The authors should provide confirmation of this with in situ staining. This also merits further discussion.

Page 17, Line 491: "Finally, we investigated how BMP ligand expression related to the BMP signaling gradient..." - This should be shown in Figure 1 or 2 and not the extended data.

Version 1:

Reviewer comments:

Reviewer #1

(Remarks to the Author)

Lewin et al. have significantly improved the manuscript, and I am happy the changes. There are some textual things, which, in my opinion, should be addressed before the acceptance, but I do not think another revision round is necessary. The data are excellent, and once the authors fix the text a little, the paper should be accepted.

Here are some comments in the order of appearance in the text.

1. Throughout the manuscript, the authors state that both in vertebrates and invertebrates, the neurogenic domain forms on the "low BMP signaling" side of the DV axis and mention the ancestrality of the centralized nervous system in Bilateria as a proven fact. "CNS in low BMP zone" also marked on the root of their tree on Fig. 6. This is a misleading oversimplification. Insects with their ventral CNS do not represent the variety of NS types in ecdysozoans, and not every spiralian has a neat ventral nerve cord like *Platynereis*. Also ambulacrarian deuterostomes do not have a CNS. There are many invertebrate groups with much less centralized – sometimes even diffuse – nervous systems, but a dorsal BMP signaling maximum is clearly an ancestral feature of Bilateria. This automatically means that some neural tissue in such "non-CNS" animals must be forming in the "high BMP" area. See <https://www.sciencedirect.com/science/article/pii/S001216061730475X> and <https://www.biorxiv.org/content/10.1101/2025.06.08.658475v1.full> It is very important that the authors discuss the negative regulation of the potentially pro-neural markers in *Lingula*, however, they also must put this into context. Lewin et al. also need to elaborate on what is known about the embryonic origin of neurons in *Lingula* and the extent of centralization of its nervous system. Does the SoxB2-expressing area of the *Lingula* gastrula really represent neurectoderm? Or do Lewin et al. think that it is neurectoderm because it expresses SoxB1, SoxB2 and Pax6? How centralized is the NS of the *Lingula* larva and adult?

2. Line 89-90: The authors talk about the "variability of ligand localisation" – please replace with "variability of ligand expression".

3. In the new version of the manuscript, the authors replaced "basal deuterostomes" with "early-branching deuterostomes". Ambulacrarians are as early-branching as the chordates. I would just stick to "ambulacrarian deuterostomes".

4. Line 463: Lewin et al. write about "neurulation in spiralian". Spiralian do not neurulate. Neurulation is a chordate-specific term.

5. In their discussion of the organiser (Lines 622 and further), the authors raise a question of whether the organiser may have been present outside chordates. The blastopore lip organiser is clearly much older than chordates – it exists in cnidarians.

See <https://www.nature.com/articles/ncomms11694> However, molecularly, cnidarian organiser resembles the Nieuwkoop center more than the Spemann-Mangold organiser. If *Lingula* has a blastopore lip organiser, it would be important to know whether only Chordin-expressing side is capable of axis induction.

(Remarks on code availability)

Reviewer #2

(Remarks to the Author)

I appreciate the efforts the authors made to revise the manuscript; my concerns and questions have been adequately addressed. The revised focus on BMP signaling---description of a BMP 'seesaw', separation of d-v axis formation and neural induction as potentially separate events, and a more nuanced description of BMP signaling and putative evolutionary scenarios throughout---has greatly improved the manuscript. I also think the addition of new gene expression data for 'neural' genes strengthens the hypothesis that BMP signaling affects neural fate in *L. anatina*. I also appreciate the clarifications throughout the manuscript in terms of methodology, reporting of results, and interpretation of those results. I think this manuscript makes important contributions to our understanding of the function of BMP signaling during development in Spiralia and evolution of BMP signaling across Bilateria and should be published.

I have a few minor questions, but I don't need a response to these before publication.

Is there a chordin-like homolog in addition to chordin *L. anatina*?

Are there distinct *elav1* and *elav2* homologs in *L. anatina*?

I am still not convinced that the neural tissue is increased/decreased after perturbation of BMP signaling in *L. anatina*. The expression of *elav* after manipulating BMP signaling is not very strong.

Fig. 2e – How do you know which side is 'dorsal' from a lateral view (is the future dorsal side morphologically distinct from the ventral side)?

(Remarks on code availability)

Point-by-point response to the reviewers' comments

We would like to begin by thanking all four reviewers for their detailed and thoughtful comments on the initial version of our manuscript. We have addressed nearly all of the points raised and have substantially revised many parts of the work as a result. In particular, key improvements include:

1. creation of a *Lingula* genome browser with integrated BLAST search capabilities;
2. inclusion of a more complete and detailed description of BMP signalling in the Introduction;
3. reworking of the narrative to focus on the identification of a BMP seesaw in a spiralian;
4. clarification of the distinct roles of BMP signalling in dorsal–ventral patterning versus neural induction;
5. repetition of *in situ* hybridisation experiments using an optimised protocol, yielding clearer results;
6. addition of new *in situ* experiments revealing neural gene responses to BMP manipulation, supporting transcriptomic data;
7. construction of gene trees for all gene families characterised in this study.

Below, we provide a point-by-point response to each comment detailing the changes made.

Reviewer #1

Lewin et al. present a chromosome-scale genome of the brachiopod *Lingula anatina* and show that 4 out of 10 *Lingula* chromosomes are results of fusion-with-mixing. By comparing the genomes of a mollusc, an early-branching annelid and *Lingula* they find evidence for lineage-specific fusions. Lewin et al. then focus on BMP signaling components, since the role of BMP signaling in the DV patterning of Spiralia is insufficiently well-understood. They show that BMP2/4 and Chordin are expressed on the opposing sides of the DV axis (dorsally and ventrally, respectively), and demonstrate a pSMAD1/5 gradient with a dorsal maximum, similar to the situation in the mollusc *Ilyanassa* and non-spiralian bilaterians. Given the diversity of the ways spiralian pattern their DV axis, this manuscript provides important support for the hypothesis that Chordin-BMP-dependent DV patterning was an ancestral feature of spiralian development shared not only by Spiralia but by Bilateria as a whole. It is an important paper, and I strongly recommend publication, however, there are several points, which need to be resolved first.

We thank the reviewer for recognising the value of our study and for their detailed, constructive comments throughout the length of our manuscript.

Major points

1. Both, in the Introduction and in the Discussion, the Authors give a misleading, oversimplified view of the BMP signaling pathway, which is reflected in their Fig. 2b and Fig. 2f, and which causes some obvious misinterpretation of the results. The Authors seem to think that BMP2/4 (=Dpp) is the only BMP responsible for the DV patterning. This is not the case. First, BMPs function as dimers, and usually the DV-patterning-relevant ligands are the BMP2/4/BMP5-8 heterodimers. So, BMP5-8 is not an "alternative BMP paralog" relevant only for the leech. It is involved in DV patterning across Bilateria and in the directive axis patterning in anthozoan Cnidaria. All the models Lewin et al. mention in the Introduction use BMP5-8 together with BMP2/4. In *Xenopus*, BMP7 (belongs to the BMP5-8 group) is expressed ventrally together with BMP4; in *Drosophila*, Screw (=BMP5-8) is expressed ubiquitously, while Dpp is dorsal; in leech, BMP5-8 is dorsal and BMP2/4 is ubiquitous etc.

First, we would like to state our appreciation for the reviewer raising these issues. It is clear that our attempts to provide a simple, concise background to our work resulted in oversimplification and in some cases incorrect statements and interpretations. To address these concerns, we have carefully revised the Introduction to provide a more accurate overview of BMP paralogues and their roles in DV patterning. With this new context, we have reworked the Introduction and added an entirely new paragraph to address the raised issues. In particular, we now describe more accurately, but still concisely, the ways that different BMP paralogues contribute to DV patterning, and highlight that they function most potently as heterodimers.

For instance, the second paragraph of the introduction now begins:

"The action of several BMP ligands, including Bmp2/4, Bmp5–8 and Admp (anti-dorsalizing morphogenetic protein), combines to regulate the process of dorsal–ventral patterning. These proteins function as homodimers but more potently as heterodimers^{18–22}, binding to type I and type II serine/threonine kinase receptors and resulting in the phosphorylation of the receptor-regulated Smad1/5^{23–26}."

Second, the Authors repeatedly call ADMP a BMP antagonist, but it is not. It is a BMP ligand (Reversade and De Robertis, 2005). And this misunderstanding probably made Lewin et al. overlook their own most striking result – they showed a "BMP seesaw" in a spiralian without noticing it. In anthozoan Cnidaria as well as in ambulacrarian and chordate deuterostomes, the second body axis is patterned by a "BMP seesaw". Both ends of the second body axis express BMPs, but BMP signaling takes place on the end of the axis, which is opposite to Chordin. For example, in *Nematostella*, one end of the directive axis expresses BMP2/4, BMP5-8 (BMPs) and Chordin, and the other end expresses GDF5-like (BMP). BMP signaling is on the GDF5-like side. In sea urchin, BMP2/4, ADMP1 (BMPs) and Chordin are expressed ventrally, and ADMP2 (BMP) - dorsally. BMP signaling is dorsal. In the frog, BMP4 and BMP7 (BMPs) are ventral, BMP2, ADMP (BMPs) and Chordin are dorsal. BMP signaling is ventral. Lewin et al. see exactly the same in a spiralian, with dorsally expressed BMP2/4 and BMP5-8 (BMPs), and ventrally expressed ADMP, BMP3 (BMPs) and Chordin and pSMAD1/5 opposite to Chordin. This is absolutely fantastic and has to become the focus of the revised manuscript.

We are grateful to the reviewer for highlighting the significance of the BMP seesaw identified by our initial *in situ* hybridisation experiments. Indeed, we now recognise the importance of

this result and have modified the manuscript significantly to emphasise its significance. First, we now include in the Introduction a detailed exploration of the current knowledge of BMP seesaw systems in *Nematostella*, sea urchin and *Xenopus*. Second, we have edited the Results section to highlight the significance of this finding. Finally, we added a paragraph in the Discussion to further explore the implications emerging from this result.

We apologise for the mistake regarding ADMP, and have now corrected this and modified our interpretation of the results accordingly.

In the revision, the Authors must show pSMAD1/5 stainings and in situs of Chordin and all BMPs expressed at gastrula (BMP2/4, BMP5-8, BMP3, ADMP) in controls and upon up- and downregulation of BMP signaling on one main text figure – not scattered between the main text and extended figures – and make a really big evolutionary point about the “BMP seesaw” being potentially the ancestral DV patterning mechanism in the Discussion (stressing the similarity to the mollusc data doi:10.1093/molbev/msab322 and to non-spiralian data). In contrast, Bambi and CV2 in situs are, of course, good to have – as examples of conserved BMP modulators expressed in the “high BMP signaling” region across species, but they can easily go into the supplement.

In addition to exploring the current state of knowledge regarding the BMP seesaw in the Introduction, we now also draw attention to this topic in our Discussion. We have dedicated an entire new paragraph to it, including suggesting that it may be the ancestral DV patterning mechanism of bilaterians.

In situs of Chordin and all BMPs are also now provided together in Figure 2 alongside the pSmad1/5 staining, showing the expression locations of BMP ligands and Chordin plus the location of BMP signal activation.

However, *in situ* quality needs to be improved, especially for BMP2/4, BMP5-8 and CV2, where signal-to-noise ratio is unacceptable. The Authors should maybe consider clearing their embryos with one of the recently published protocols e.g. <https://www.science.org/doi/10.1126/sciadv.aba0365> to improve the quality both of the in situs and of the morphological description shown in Extended Data Fig. 7.

We completely agree with the reviewer that the quality of several of the previous in situs was sub-standard, and thank them for their recommendation of clearing protocols. We have now completely re-done the poor quality *in situ* experiments, in addition to adding new ones, and achieved far cleaner results.

2. If the Authors publish a chromosome-scale genome, they should also make it easy to use for the community. I understand that this is a lot of work, but a publicly accessible *Lingula* genome browser with a built-in BLAST option rather than a link to a bioproject with raw reads would be great.

We agree with the reviewer’s comment that our *Lingula anatina* genome should be easily accessible to the community. For this reason, we have built a genome browser for the chromosome-level genome hosted on the Okinawa Institute of Science and Technology (OIST) website: https://marinegenomics.oist.jp/lan_3_0/viewer/info?project_id=124. As

suggested, the browser has a built in BLAST option, and we also included gene model tracks with protein annotations. Additionally, all relevant files including the genome sequence itself, gene models in various formats, and gene model annotations are made available for download from this website. We anticipate that this will facilitate use of our genome by the wider community.

The genome is also now available on NCBI with the accession GCA_051362555.1. The Data Availability section has been updated to reflect these improvements.

Other comments

1. Line 81-88. The "typical spiralian" argument is not ideal. Among high-level spiralian taxa, spiral cleavage exists in mollusks and annelids, however, as far as I am aware, no lophophorates (bryozoans, phoronids etc) exhibit spiral cleavage. Flatworms or nemertines also do not have spiral cleavage. Most gnathiferans (chaetognathes, rotifers etc.) also do not have spiral cleavage, but gnathostomulids have it. Given that, I do not think we can state with certainty that spiral cleavage was lost multiple times or gained multiple times in Spiralia (as the Authors do by saying "possibly reflecting a reversion to the ancestral state"). On the other hand, some spirally cleaving molluscs use BMP for DV patterning, and spirally cleaving annelids maybe use it as well - for example *Owenia* has Chordin and BMP2/4 and likely uses it for the DV patterning. In general, I suggest that instead of explaining why other models are not ideal, the Authors should argue that with the increased sampling across the spiralian phylogeny, we get a better understanding of the ancestral mode of BMP signaling in Spiralia.

We thank the reviewer for highlighting the sub-optimal nature of this line of reasoning. Indeed, we agree completely that we cannot state with certainty whether spiral cleavage was lost or gained multiple times within Spiralia. To resolve this, we have made the changes that the reviewer suggested, moving away from explicit statements about "reversion to the ancestral state" and instead arguing that brachiopods are valuable models to increase sampling across spiralian phylogeny and get a better understanding of the ancestral mode of BMP signalling in Spiralia. We therefore modified both the Introduction and the Discussion to reflect this. As an example, the modified paragraph of the Introduction is as follows:

"In this context, the focus of most spiralian studies on only two phyla, molluscs and annelids, limits our understanding. Broader sampling across spiralian phyla is essential to clarify the ancestral role of BMP signalling within this group and beyond. To this end, we aimed to study dorsal–ventral patterning in brachiopods, a phylum of marine spiralian phyla with calcified dorsal–ventral shells that is significantly understudied compared to other spiralian phyla."

2. Fig. 1c. How did the Authors get the list of BMP signaling component orthologs for Fig. 1c? Was it BLAST or reciprocal BLAST or something else? How certain are the Authors that lophotrochozoan BMP3 is an ortholog of chordate BMP3? I may have easily missed something, but if I am not mistaken, the only published non-chordate putative BMP3 comes from the ctenophore *Mnemiopsis*, however, its phylogenetic position as a BMP is not robust (compare Fig. 2 in doi: 10.1371/journal.pone.0024152 and Supplementary Fig. 2 in doi:10.1242/dev.141507). If BMP3 is indeed always present in Lophotrochozoa, it is an important piece of information. A phylogenetic tree with BMP ligand distribution across

Bilateria (including ADMP, GDF5-6 etc.) would be a very useful supplementary figure. It would also be very useful to add *Xenopus* to Fig. 1c either in addition or instead of human, since this is the animal authors constantly refer to in the text.

We thank the reviewer for bringing to our attention the lack of clarity in this section of our manuscript and for suggesting that we double check our BMP3 annotations. To resolve these several queries, we performed the following additional analyses and modified the text and supplementary figures accordingly:

- 1) We have included data for *Xenopus* in Fig. 1c as suggested.
- 2) We have built the suggested tree of BMP ligand distribution across bilaterians. This is now presented as Supplementary Fig. 7. It includes all TGF-beta ligands from both human and *Drosophila* as a baseline, plus all ligands for all species identified in this work. With regards to BMP3, our tree shows a strongly supported clade of genes containing vertebrate BMP3 and GDF10 (also known as BMP3b) genes alongside our annotated BMP3 orthologues in spiralian. This is consistent with the results of a previous study (Kenny et al. 2014 *Int. J. Dev. Biol.*) which also reported BMP3 genes in spiralian (and, like us, failed to find one in ecdysozoans). In addition, BMP3s have also been reported in species-specific studies in many spiralian (e.g. annelid *Capitella*, Lanza and Seaver 2020 *Development*; oyster *Pinctada*, Fan et al. 2019 *J. Appl. Anim. Res.*; clam *Ruditapes*, Wang et al. 2021 *Genomics*; scallop *Mizuhopecten*, Zhang et al. 2024 *BMC Genomics*). Given the single BMP3/GDF10 loci present in spiralian and in non-vertebrate chordates like amphioxus (Sun et al. 2010 *Development, Growth and Differentiation*), it appears likely that the duplication of an ancestral BMP3/GDF10 occurred in vertebrates to form these two genes. To be consistent with previous studies, the majority of which use the nomenclature 'BMP3', we refer to the orthologues we identify as BMP3 rather than GDF10 or BMP3/GDF10.
- 3) We have re-worded this section of the Methods to explain more clearly how genes were identified: "We used OrthoFinder (v2.5.4) to identify putative homologues of BMP pathway components in lophotrochozoan genomes. Genes were identified by orthology to sequences from human and *Drosophila*, which are well-annotated and verified. All cases of putative gene losses and duplications were then manually verified using gene tree construction with IQ-TREE (v2.2.2.3) plus reciprocal blast searches and microsynteny comparisons where necessary. Gene trees for each superfamily were built using a pipeline of MAFFT (v7.520) alignment, ClipKIT (v1.4.1) trimming, and IQ-TREE tree-building (v2.2.2.3) using 1000 ultra-fast bootstraps and automated substitution model optimisation with ModelFinder."

3. Fig. 1d. I assume the Authors did not do this for space reasons, but they should add *Owenia* and *Linneus* data on Fig. 1d. Having them separately on Extended Fig. 6a-b is very confusing, especially since the order of the chromosomes is always different for each species on each figure. If having amphioxus, *Pecten*, *Lingula*, *Owenia* and *Linneus* on one panel is impossible (although I do not see why it should not be!), and the Authors want to keep their *Pecten* + *Lingula* + "1 more species in the middle" comparisons, they should fix the order of *Pecten* and *Lingula* chromosomes, adjust the third species to this fixed order and put all these figures as panels on the main text figure.

The reviewer is correct that we initially omitted these comparisons from the main figure due to space constraints. We agree that it would be better, however, if all were shown on a single figure, and have now done the necessary reformatting to achieve this. Figure 1d has been modified as suggested, adding *Owenia* and *Lineus* to the main figure with the other three species. We are happy that this makes it easier for readers to make 5-way comparisons between the species without flicking between pages.

In addition, we have updated the corresponding dot plots in Supplementary Fig. 18 to ensure all comparisons are represented and that the chromosomes are in the same order in all figures.

4. For ALG association of BMP genes, the result is quite striking, but I may have overlooked a careful explanation of how random genes were selected. It is important and must be clearly explained in the main text. Also, the Authors should comment on whether the chromosomal linkages of BMP signaling components are preserved in highly scrambled genomes (*Drosophila*, *C. elegans*, clitellate annelids, tunicates etc.) or genomes which underwent more translocations than, say amphioxus or *Pecten* - e.g. human genome, *Streblospio benedicti*, *Carcinoscorpius rotundicauda*.

We thank the reviewer for highlighting the lack of explanation of the methods. To rectify this, we have created a separate paragraph in the Methods section to explain the process by which random genes were selected:

“The genomic locations of BMP pathway-related genes, Wnt ligands, and Hox genes in each of the five species were identified with blast (v2.14.1). We then determined the ALGs present on the chromosome hosting each gene using the above macrosynteny analysis. Conserved associations with ALGs were inferred from the genes’ chromosomal locations: if a gene was located on a chromosome containing the same ALG in all five species, then it was considered to have a conserved ALG association. To determine whether the rate of conservation of ALG associations is elevated in BMP pathway-related genes, Wnt ligands, and Hox genes compared to the background rate of all genes in the genome, the rate of conserved associations of developmental genes with a specific ALG was compared to that of a random sample of 100 single-copy orthologues using a chi-square test (Supplementary Table 35). The random sample was created by running OrthoFinder (v2.5.4)¹²⁰ to identify single copy orthologues across the five species and then using the shuf command in bash with option -n 100 to choose 100 random rows of the output single copy orthogroups file. The presence or absence of conserved ALG associations for these genes was then determined by the same method as above.”

Whether incrementally more rearranged genomes preserve these linkages of BMP components despite their higher rearrangement levels is an interesting question and we are grateful to the reviewer for raising it. In *Owenia*, which has an intermediate level of chromosome rearrangement, higher than *Pecten* or amphioxus and similar to that of *Streblospio* (Lewin et al. 2024 *Mol Biol Evol*) we also found 100% conservation of BMP gene linkages. However, in response to the reviewer’s comment, we wondered whether this was true in even more highly rearranged species. To test this, we used the bryozoan *Membranipora membranacea*. We chose this species because it is (1) a spiralian, (2) already in our dataset of BMP characterisation species and (3) a member of Bryozoa, a

phylum that has undergone extensive rearrangements, with all 24 ancestral linkage groups fused or fissioned at least once, but, importantly, still present and recognisable in the genome (Lewin et al. 2025 *Genome Res*). We found that of the 17 BMP pathway gene families present in this species, 16/17 maintained conserved associations with the expected ancestral linkage group. One gene, *smad4*, is associated with ALG C2 in all other species but not in *M. membranacea*, suggesting a small-scale translocation of this gene. We interpret this to mean that BMP-related gene linkages are highly stable but not completely fixed, even in lineages where extensive interchromosomal rearrangements have occurred. We have expanded Supplementary Table 17 and added several sentences into our manuscript main text to reflect this result:

“Even in species whose genomes have undergone extensive interchromosomal rearrangements, like that of the bryozoan *M. membranacea*^{61,73}, 16 of 17 BMP pathway gene families present have conserved ALG associations. This suggests that BMP-related gene linkages are highly stable but not completely fixed, even in lineages where extensive interchromosomal rearrangements have occurred.”

Finally, we note that in very highly scrambled genomes like those of clitellate annelids and *Drosophila*, the original bilaterian ancestral linkage groups have been completely lost, with genes from each linkage group spread around the genome and showing no statistical association with any chromosome (Vargas-Chávez et al. 2025 *Nat Ecol Evol.*). Since the linkage groups themselves are no longer present, it is unfortunately not possible or meaningful to test for a conserved association of BMP genes with that linkage group in this way.

5. Lines 239-240 and Fig 2a. The Authors write that “The co-option of BMP–Chordin signalling pathway into dorsal–ventral axis patterning is a cornerstone of the bilaterian body plan”. While I fully agree that BMPs and Chordin are important, I am not sure about the co-option. Co-opted from doing what? Co-option means that BMP and Chordin were doing something else in the bilaterian ancestors and then were re-deployed in DV patterning. If their function in DV was re-deployment, what was the ancestral function?

We thank the reviewer for raising this and agree that the word co-option was misleading. We have therefore rephrased the sentence as follows to remove it:

“The use of the BMP–Chordin signalling pathway for dorsal–ventral axis patterning is a cornerstone of the bilaterian body plan (Fig. 2a).”

Figure 6 has also been edited to remove this phrase.

Also in the Fig. 2a, the Authors suggest that “The BMP-Chordin axis predates the origin of bilaterians and, therefore, the dorsal–ventral (DV) axis.” I am not sure how the Authors make this conclusion. What do they call a BMP-Chordin axis? A body axis patterned by BMP and Chordin? Their tree on Fig. 2a suggests that it evolved prior to the split of Cnidaria and Bilateria. In that case, cnidarian second body axis (directive axis of anthozoans) is homologous to the DV axis of Bilateria, and it should be therefore called DV axis as well. However, it is not entirely certain whether directive and DV are homologous or evolved convergently. If it is convergence, then Lewin et al. cannot say that “BMP-Chordin axis

predates the origin of Bilateria" - in this case, these are different body axes, which just happened to use BMP signaling for patterning. I suggest removing the term "co-option" and carefully rephrasing the legend.

We agree with the reviewer's statement that "it is not entirely certain whether directive and DV are homologous or evolved convergently". Though some authors have proposed a common origin (e.g. Finnerty et al. 20024 *Science*, Matus et al. 2006 *PNAS*), evidence is conflicting and inconclusive (Genikhovich et al. 2015 *Cell Reports*), and it seems unlikely in particular that Chordin-mediated demarcation of a CNS is ancestral to cnidarians and bilaterians (Saina et al. 2009 *PNAS*). Indeed, a recent review again highlighted the possibility of directive and DV axis homology, but still could not rule out convergence (Mörsdorf et al. 2024 *Development Genes and Evolution*). Since our manuscript focuses only on bilaterians, and in particular spiralian, a pre or post-cnidarian origin for this axis is not directly relevant to our results or their implications. We have therefore elected to remove discussion of this and avoid potentially controversial statements on this subject. As a result, we have (1) removed our statement that the "BMP-Chordin axis predates the origin of Bilateria" and carefully rephrased the legend to avoid implying as such; (2) revised the figure by removing the label 'BMP-Chordin axis' at the base of Eumetazoa and adding a new label, 'BMP-mediated secondary axis', to reflect the observation that both cnidarians and bilaterians possess a secondary axis patterned by BMP signalling; and (3) eliminated our use of co-option with respect to the origin of the BMP-Chordin axis throughout the manuscript. We are grateful to the reviewer for highlighting this potential source of controversy and allowing us to address it before publication.

6. Fig. 2b. I understand that the Authors tried to keep things simple, however, not only do BMPs signal as dimers (see major point 1) but also the transduction of the signal is performed not by a type I BMP receptor but by a heterotetrameric complex containing two type I BMP receptors and two type II BMP receptors. Additionally, BMPRI is located within the cell in the schematic, however the receptor complex should be positioned on the cell surface instead.

Thank you for the suggestion. While we prefer to keep the schematic itself simple, we have adjusted it for greater accuracy by placing the label for BMP receptors on top of the plasma membrane. We have also added text into the legend to clarify their location and that they function as a heterotetrameric complex.

7. Fig 2c. What is the Authors' definition of the "main BMP signalling ligands, mediators, and modulators". I would recommend either including the full heatmap of *L. anatina* BMP pathway components shown in extended data Fig. 8a or restricting the listed components in Fig 2c to only the BMP ligands (including ADMP) and Chordin, as these are the focus of the paper.

We have now included the full heatmap in Fig. 2c as requested.

8. Fig. 2d-e. It is not entirely clear why these have to be main text figures. It is nice to show that *Lingula* SMAD1/5 is really a SMAD1/5 (Fig. 2d), and it is excellent that the Authors demonstrate that the epitope recognized by the anti-pSMAD1/5 antibody used in the paper is present in *Lingula* SMAD1/5 (Fig. 2e), however, these are "technical details", which should go

into the supplement. Also, the Authors start using the term R-Smad without explaining what it is. They must write that Smad1/5 is the R-Smad in the BMP signaling (and Smad2/3 is the R-Smad in the Nodal/TGF β signaling) - otherwise unclear.

Other reviewers also raised this and we agree that these sub-figures represent technical details that are non-essential for an understanding of the main points of the manuscript. We have thus moved them to the supplementary material as suggested.

In addition we have completely removed any uses of the term R-Smad. This was only used in the figure legend and is not relevant for the remainder of the manuscript.

9. Fig. 2f. I find this very confusing. First, the signal is normally transduced not by a Smad1 homotrimer as the Authors depict on the image but by a heterotrimer containing two Smad1/5 molecules and one Smad4 (Co-Smad) because such heterotrimer is energetically more stable than a homotrimer. Second, I do not understand why they point out the L3 loop - they never discuss this in the text, and this is not the site where their antibody binds. L3 should either be removed from the image and legend or elaborated upon in the text. If no additional discussion of the Fig. 2f content is added, I suggest its removal from the paper as it does not contribute significantly to the main narrative.

We agree that this figure was both confusing and unnecessary for understanding the paper and have therefore completely removed it.

10. Fig. 3a. Why were two treatment durations used? This is not clear and must be explained in the main text. In addition, it must be made clear in the main text or figure legend of each experiment throughout the entire manuscript which treatment regime was used.

We applied two treatment durations to manipulate BMP signalling after its activation, as BMP signal readout begins at the early blastula stage. Since our focus was on two key developmental stages representing major milestones in development—late gastrula and one-pair-cirri larva—it was logical to block or enhance BMP signalling during these specific periods. We have revised the main text to clarify this rationale. The updated text now reads:

“To investigate the role of asymmetric BMP activation, we perturbed BMP signalling with receptor inhibitors and recombinant BMP proteins to block or enhance its activity, respectively. Treatments were applied either from early blastula to late gastrula, when dorsal–ventral molecular features first appear, or extended to the one-pair-cirri larva, after axis patterning is established (Fig. 3a, Supplementary Table 21).”

11. Fig 3, Fig 5, extended data figure 9. Sample size and phenotype penetrance should be provided for all immunostainings and RNA *in situ* hybridisations (i.e. X embryos out of a sample of Y develop the phenotype shown on the representative image).

We used approximately 30–50 embryos for all experiments. Phenotype penetrance was 100% for all immunostainings. For the *in situ* hybridisation data, penetrance was typically greater than 90%. As some of these experiments were conducted nearly a decade ago, the original slides are no longer available for precise quantification. For newly generated *in situ* hybridisations, penetrance is indicated in the bottom right corner of each representative

image (for example, 50/50 denotes that all 50 embryos analysed exhibited the same phenotype).

12. Fig 3c-f. It is mentioned briefly in the main text that manipulation of BMP signalling delayed gastrulation. Do the authors have data regarding the extent to which gastrulation is delayed? Is this the case for all individuals? Do the animals eventually complete gastrulation?

Gastrulation delay was assessed based on archenteron length. In the control, the archenteron nearly reached the animal ectoderm by the late gastrula stage (Fig. 3c). However, treatment with BMP inhibitors such as LDN193189 (Fig. 3d) and K02288 (Fig. 3e) disrupted this process. K02288 caused a more severe phenotype, significantly delaying gastrulation and preventing archenteron formation at the same stage as the control. Despite this delay, the embryos eventually completed gastrulation, as evidenced by the presence of larval structures at later stages (Fig. 3k–n). We have updated the figure legends to clarify this gastrulation defect.

“Note that gastrulation was disrupted in embryos treated with BMP inhibitors such as LDN193189 and K02288, with K02288 causing a more severe phenotype. This treatment significantly delayed gastrulation and prevented archenteron formation at the same stage as the control.”

13. Fig. 3g-j. Why are 2 different doses of inhibitor/recombinant BMP used? Which dosage were the animals shown in panels g-j exposed to? It must also be made clear in the figure legend or main text whether the animals used in experiments shown in fig. 3c-f and 3k-n were treated with high or low doses. This information must also be made clear for all experiments throughout the manuscript that made use of the BMP inhibitors/recombinant BMP.

We initially applied two doses to assess dose dependency but found them to be indistinguishable. Thus, no clear dose-dependent effect was observed under our experimental conditions. Instead, phenotypic differences were more pronounced when comparing different BMP inhibitors, with K02288 consistently inducing a more severe phenotype than LDN193189, even at nanomolar concentrations, suggesting that K02288 is a more potent inhibitor. This was particularly evident when quantifying the number of pHistone H3-positive cells. Although a slight increase or decrease was observed under BMP inhibitors and recombinant BMP, respectively, the differences were not statistically significant (Fig. 3o).

For all experiments, immunostaining and *in situ* hybridisation were performed for control, low-dose, and high-dose treatments. However, for clarity, only control and high-dose treatment images are shown in the main figures. The figure legends have been updated accordingly.

“Since no clear dose-dependent effect was observed, all images for BMP manipulation are from the high-dose treatment unless otherwise noted.”

14. Fig. 3k-n. This needs better (larger and better annotated) pictures. Also, the Authors had two types of treatment - do they see the chitin phenotype in both or only in the treatment spanning longer? They have to comment on whether there are any differences between the treatments.

As noted above, no dose-dependent effect was observed on morphological changes. Since these images are confocal z-stacks, we have provided the original Zeiss LSM files, preserving all z-slices at high resolution. These files are now available on Zenodo (<https://doi.org/10.5281/zenodo.16916709>) under the following file names.

- Lan_Larval_Chitin_DMSO_control_01.lsm
- Lan_Larval_Chitin_LDN193189_03.lsm
- Lan_Larval_Chitin_K02288_01.lsm
- Lan_Larval_Chitin_mBMP4_01.lsm

15. Lines 376-377. GO terms are useful but also very imprecise. I think that calling these genes “neuronal” is a BIG stretch. I do not know which alk the Authors mean - for me Alk2, Alk3/6, Alk4/5/7 are different type I BMP or TGFb receptors, which cannot be called “neuronal”. FoxB is primarily an endomesodermal marker. Lhx1/5 has functions in the neurons but also in the development of the lymphoid tissue. Netrin is an axon guidance molecule but has other functions as well. Nkx2.4 is expressed in the brain, but predominantly in the testes. Other genes the Authors mention here can probably be called neural without too many additional disclaimers. I do not doubt that in some context all these genes are expressed in neurons, but I think that this bit needs some very careful re-phrasing. E.g. “dominated by genes demonstrating either exclusively neuronal expression or known to be expressed in neurons in specific developmental contexts in other model organisms”. I understand that in situ validation of neuronal expression of all these genes in *Lingula* maybe too much for this paper (although it would be great to have it), but without it the argument has to be tuned down and carefully phrased.

We agree with the reviewer that our previous statement of these genes simply as ‘neural’ was insufficiently clear. Indeed, we have now used their suggested phrasing, and agree that this makes the argument more accurate and specific. The full sentence now reads:

“In this dataset, the BMP-downregulated targets were dominated by genes demonstrating either exclusively neural expression or known to be expressed in neurons in specific developmental contexts in other model organisms, including *elav*, which may be a universal neural marker⁹², *foxb1*, *lhx1/5*, *netrin*, *nkx2.4*, *otx2*, *pax6*, *six3/6*, *sim1/2*, *soxb1* and *soxb2*⁹³⁻¹⁰⁰ (Fig. 4h, Supplementary Figs. 27, 28).”

We have also applied this same logic throughout the length of the manuscript, being more specific at each previous mention of neural genes. For instance, we have replaced the phrase “neural genes” with “genes typically associated with neural expression” at the previous L33.

In addition, we have removed *alk* from this list as we agree that it is not sufficiently specific as a neural marker.

Moreover, we also selected four key neural genes (*soxb1*, *soxb2*, *pax6* and *elav*) and performed *in situ* hybridisation experiments under both control and BMP signal manipulation conditions. All four genes are expressed at the ventral side of the embryo directly opposite the pSmad1/5 maximum. What is more, for *soxb1*, *soxb2* and *pax6* genes, the results of BMP signal manipulation experiments strongly support our contention that neural gene expression is repressed by BMP signals, while the effects on *elav* are weaker but still detectable. The Results section describing this now reads:

“To validate the transcriptomic data, we examined expression of four key neural markers—*soxb1*, *soxb2*, *pax6* and *elav*—by *in situ* hybridisation in control, BMP(+) and BMP(-) embryos (Fig. 4d, Supplementary Table 29). In control embryos, all four genes were expressed ventrally, directly opposite the dorsal BMP maximum (Fig. 4d, top row). For *soxb1*, *soxb2*, and *pax6*, inhibition of BMP signalling caused a dorsal expansion of expression domains, with stronger effects under K02 compared to LDN treatment (Fig. 4d, second and third rows). In contrast, BMP(+) treatment abolished detectable expression of these three genes (Fig. 4d, bottom row). *Elav* expression was reduced under BMP(+), and showed a slight, though less clear, upregulation with K02, consistent with the RNA-seq data, where the effect on *elav* is less pronounced than other neural genes (Fig. 4c). Overall, these data show that key neural genes are expressed in the same location as *chordin*, diametrically opposed to the high point of BMP signal activation. Moreover, their expression domains expand when BMP signalling is inhibited and are eliminated when it is overactivated, confirming that BMP signals restrict the expression of neural genes.”

16. Fig 4g. BMPR should be positioned on the plasma membrane, not inside the cell.

We have removed the cellular context lines in the revised Fig. 4b, c and retained only the gene–gene interactions to simplify the presentation.

17. Fig. 5a. The Authors should replace BMP5 with BMP7 in the *Xenopus* ventral centre, add BMP2, and re-color ADMP in the dorsal centre (ADMP is not a BMP antagonist! It is a BMP!).

We have replaced BMP5 with BMP7 as suggested. The colour in this case is simply blue for dorsal and red for ventral, so will not change based on the fact that ADMP is a BMP. We have also modified Fig. 5d to reflect the fact that ADMP is indeed a BMP signalling ligand.

18. Lines 480-490. The section on TAI analysis is confusing. It is unclear how the results of this analysis indicate that BMP signals are crucial for gastrulation (Line 488). Is it because of the GO terms associated with the gastrulation process? The delayed gastrulation observed in Fig. 3 is more convincing evidence that BMP signalling is important for gastrulation. In addition, the description of the role of BMP signalling as “crucial” for gastrulation is too strong, as it is previously stated on line 319 that BMP manipulations only delay gastrulation, and not that BMP inhibition prevents it.

We agree with the reviewer that the TAI analysis does not suggest that BMP signals are “crucial” for gastrulation in *Lingula*, and that the word crucial is too strong. We have now modified the offending sentence to remove the word crucial and to incorporate the evidence

from BMP manipulation experiments for involvement in gastrulation. The new sentence reads:

“When combined with our BMP signal inhibition experiments, it suggests that BMP signals are involved in a gastrulation process that is marked by increased expression of genes that emerged relatively recently in evolution.”

19. Supplementary fig 5a. In the text, late blastula and mid gastrula stages are highlighted as stages with the highest TAI score but the Authors refer to a figure highlighting a different stage – the late gastrula, which has the lowest TAI.

We thank the reviewer for drawing our attention to this inconsistency. The figure has now been corrected to highlight the stage with the highest TAI, as is stated in the text.

20. Supplementary figure 5b. It should be made clear what “BMP related” genes are. Are these genes that are members of the BMP family, or are they components of the BMP pathway?

We have now clarified this in the figure legend: “The TAI of BMP pathway genes (dataset from Main Text Fig. 1c)...”

We have also changed the three mentions of BMP-related genes in the main text to “BMP pathway” genes.

21. Lines 491-499. This needs to be re-written once my Major point 1 has been addressed. Also more data needs to be presented to confirm that BMP2/4 and BMP5-8 expression on the dorsal side is suppressed by BMP signaling. It is counterintuitive, and if true, very interesting. However, it is not sufficiently well demonstrated. High quality *in situ* and QPCR are necessary upon up- and downregulation of BMP signaling.

We have now removed from the manuscript attempts to assess the effect of BMP signalling on the expression of most BMP ligands in general as this was not directly related to our main question and results of *in situ* hybridisation experiments were inconsistent. It does, however, appear from the *in situ* that the BMP ligand ADMP is indeed downregulated by BMP signalling but we have chosen not to emphasise this in the text given the lack of corroborating evidence.

22. Lines 512-513. Hemichordates and sea urchins are not “basal deuterostomes”. Together, they are Ambulacraria and a sister group to Chordata. If the Authors wish to take sea urchins and hemichordates separately, then Chordata would be an earlier branch of Deuterostomia than either Hemichordata or Echinodermata.

We have now changed all mentions of “basal deuterostomes” to “early-branching deuterostomes” as suggested by Reviewer 1. We have also specifically defined this term at its first use as: “echinoderms and hemichordates, together forming Ambulacraria”

23. Fig.6. The BMP gradient depicted (pink) should be much narrower. The width of the pink shaded region better reflects the region of Dpp expression than the nuclear pMad gradient (Raftery and Sutherland, 2003).

We thank the reviewer for the suggestion, but we would like to retain our original schematic. For broader cross-species comparisons, it is important to maintain consistency by presenting the BMP gradient using the BMP signalling readout—in this case, the pSmad staining results. This approach is particularly relevant for species such as sea urchins, where BMP ligands are co-expressed with *chordin* in the same domain. Therefore, we chose to focus on the BMP gradient and *chordin* as the primary components of dorsal–ventral axis patterning, rather than on the expression patterns of *bmp/dpp* and *chordin*.

24. Fig 6. In the legend, the blue region on each embryo cartoon is described as the “anti-BMP domain”. This is inaccurate/vague since this area expresses its own BMP ligands and it should just be referred to as the *chordin/sog* expression domain.

We now refer to a *chordin/sog* expression domain rather than anti-BMP domain as suggested.

Reviewer #2

BMP signaling is involved with establishing the dorsal-ventral axis and a region of neuroectoderm in several taxa in deuterostomes and ecdysozoans. However, experiments testing the function of BMP signaling in these processes across spiralian taxa have found differing levels of involvement. To understand whether BMP signaling was ancestrally involved in forming the dorsal-ventral axis including the neuroectodermal domain in bilaterians, the authors examined BMP function in the spiralian *Lingula anatina*, a brachiopod that develops by radial rather than the ancestral spiral cleavage. Unlike some other spiralian taxa, *L. anatina* appears to have some of the same regulatory logic of the BMP signaling gradient in chordates, including an anti-neural function for BMP. The authors also generated a chromosome-level assembly for *L. anatina* and reconstructed the complement of BMP pathway components for multiple spiralian taxa. The paper is well-written and appropriate for a broad audience, the data are of high quality, and the experiments and findings are important for understanding how the dorsal-ventral axis including the neuroectoderm evolved in bilaterians.

We are grateful to the reviewer for recognising the quality of our data and importance of the findings we report. We detail below the changes made in response to each of their comments.

Main comments/suggestions

The authors argue that the radial cleavage present in *L. anatina* may be a ‘reversion to the ancestral state’. However, this still raises the question of the evolutionary history of BMP signaling in spiralian. Since spiral cleavage was likely ancestral for the clade containing brachiopods, annelids, and mollusks, then BMP signaling was first used in the context of spiral cleavage. It would be interesting for the authors to comment on how they think BMP

signaling was used to form the dorsal-ventral axis in spirally-cleaving embryos and was then re-evolved to form the dorsal-ventral axis in a radially-cleaving embryo. Why do you think there is such variability in the involvement of BMP signaling and dorsal-ventral axis formation across different spiralian taxa? I think the discussion points on lines 563–570 are very interesting, and begin to get at this question, but I would like the authors to comment on this in the context of gain and loss of spiral cleavage. In *L. anatina*, BMP signaling does not appear to be active (pSMAD1/5 levels) during cleavage stages, unlike other taxa with spiral cleavage, but instead is active in the blastula. To really demonstrate a lack of BMP signaling during cleavage, I think it's important to test function of BMP during this time window (see additional comments for Fig. 2 below).

This comment from the reviewer raises several key points which are each worthy of significant attention. In order to accommodate this, we have added two additional paragraphs into the Discussion of our manuscript. These sections discuss possible reasons why there is such variability in the involvement of BMP signalling on dorsal-ventral axis formation across spiralian within the context of how BMP signalling is used to form this axis in spirally cleaving embryos. We hope that we now have a more complete exploration of the relationship between BMP signalling pathway evolution in spiralian and the gain and loss of spiral cleavage. For instance, one of the new paragraphs reads:

“Irrespective of the direction of change, it is clear that the role of BMP signals in neural induction and dorsal–ventral patterning shows extensive diversity across the Spiralia^{42,47,48,52,55,56,128}. Why has a mechanism highly conserved from arthropods to vertebrates been repeatedly modified in this specific group? The answer likely lies in a combination of two features of spiralian development. First, spiralian development typically consists not only of spiral cleavage itself but also stereotyped specification of certain blastomeres (i.e., highly regular cell division and conserved cell lineages)^{129,130}. This is intricately interlinked with axis specification and dorsal–ventral patterning. For example, the spiralian-specific D-quadrant organiser is a key contributor to dorsal–ventral axis specification^{131–133}, and the organising activity of the D-quadrant of the molluscs *Ilyanassa obsoleta* and *Lottia peitaihoensis* relies on *Bmp2/4*^{55,56} (though this appears not to be the case in annelids^{47,134}). Second, spiralian development is highly variable and it is clear that both spiral cleavage and stereotyped cell lineage specification have been modified and lost repeatedly^{130,135,136}. Lineages exhibiting equal versus unequal cleavage, for instance, polarise their secondary axis at different times via different mechanisms^{131,137} and brachiopods themselves, though phylogenetically unquestionably spiralian, do not show spiral cleavage. It follows that, since spiralian development is highly variable and intertwined with dorsal–ventral patterning, the role of BMP signalling is modified as part of this lineage-specific variation.”

We also note that at the request of Reviewer 1, we have now moved away from commenting on the potential ancestral state of cleavage within spiralian, including removing the claim that there may be a ‘reversion to the ancestral state’ and brachiopods.

Finally, we agree with the reviewer that it would be interesting to test the function of BMP during cleavage and that this is important to confirm a lack of BMP signalling at this time. However, due to a recent move of the lab since these experiments were performed, repeating them is now unfortunately highly impractical and, given the need to collect animals

from a remote island in Japan at a specific time of year (because we are working with a non-model organism with no complete life cycle in the lab), would likely delay publication by several years. Given the fact that, while interesting, the lack of BMP function at this stage is not central to the paper, we do not think that this question justifies such a delay. This is especially true since in this second version we have, at the request of other reviewers, moved away from the narrative of arguing that *L. anatina* is a superior model because of its lack of spiral cleavage. Despite our inability to directly address this question with experimentation, we recognise its importance. To ensure our manuscript has a clear discussion of this issue and makes only claims that are directly supported by our data, we have now ensured that, while highlighting the lack of Smad1/5 during the cleavage stages, the manuscript does not claim that BMP signalling is inactive at this time point. We also inserted a sentence to remind the reader that we did not actively test the role of BMP signalling during cleavage stages and another to reiterate the importance of performing these experiments in the future. This section of the Results has been carefully rewritten and now reads:

“From the 1-cell to 12864-cell stages, we detected no pSmad1/5 nuclear signal (Fig. 2d, f–n, Supplementary Fig. 23). Expression of the main BMP ligand *bmp2/4* is very low before the early blastula stage (Fig. 2d), suggesting that BMP signalling is not activated until after the cleavage stages. However, we did not directly test the effects of BMP signal manipulation at this stage, and further experimentation is required to confirm this.”

The regulation of *bambi*, *cv2*, *admp*, and *chordin* by BMP signaling in *L. anatina* (Fig. 5) is interesting. The authors suggest that this is indicative of a BMP signaling gradient controlling fates along the dorsal-ventral axis as in chordates, and that this system may have been present in the last common ancestor of the two groups (e.g., lines 450–451). An alternate interpretation of the functional transcriptomic data is that a gradient of BMP signaling was ancestral and that regulatory logic is maintained in *L. anatina*, irrespective of what developmental program (e.g., fates along the dorsal-ventral axis) this signaling system controlled. I think that ventralization or dorsalization (or loss/gain) of multiple fates after drug treatments or BMP protein treatment in *L. anatina* needs to be demonstrated more clearly to support the hypothesis of an ancestral function in dorsal-ventral axis formation. The change in expression domains of *bambi*, *cv2*, *admp*, and *chordin* after perturbing BMP signaling does not necessarily indicate a shift in fates along the forming dorsal-ventral axis, especially since many of the other Spemann-Mangold organizer genes were not found to be regulated by BMP signaling in the same direction as in *Xenopus* (e.g., lines 476–478).

We agree with the reviewer that the observed similarities in gene regulatory interactions of the BMP signalling networks of *Lingula* and *Xenopus* do not necessarily indicate a conserved role in controlling cell fates along the dorsal-ventral axis. Similarly, we agree that claims of dorsalisation and ventralisation are insufficiently supported with the current data. To rectify this, we have re-focused the relevant sections to discuss only gene expression and gene regulation and not make claims about dorsal-ventral axis specification and subsequent changes to this like ventralisation. For instance, the sentence on previous lines 450-451 has been changed from discussing ‘dorsal–ventral patterning system architecture’ to ‘BMP signalling network architecture’.

In *L. anatina*, does the neuroectoderm form on one side of the trunk ectoderm during gastrulation, and are the differentially-regulated genes expressed in the neuroectoderm? It looks like the references for the sentence on lines 375–377 are for other animals. For Fig. 4, I would like to see spatial expression by *in situ* hybridization of some of the differentially expressed neural genes identified (e.g., *elav*, *pax3/6*, *soxB1*) in controls and before and after manipulation of BMP signaling.

Yes, the neuroectoderm forms at the ventral side of the ectoderm during gastrulation. In this context, we agree with the reviewer that *in situ* hybridisation data is required to independently verify our claims about the effects of BMP signalling on differentially expressed genes and to determine their expression location in control embryos. We therefore selected four key neural genes (*soxb1*, *soxb2*, *pax6* and *elav*) and performed *in situ* experiments under both control and BMP signal manipulation conditions. All four genes are expressed at the ventral side of the embryo at the site of neuroectoderm formation, directly opposite the pSmad1/5 maximum. Moreover, all four genes show suppressed expression under the BMP(+) condition and expanded expression under the BMP(-) condition, although the effect is less pronounced for *elav* than for the other three genes, consistent with the hierarchical clustering results in Fig. 4c. These results strongly support the conclusion drawn from RNA-seq data that neural gene expression is repressed by BMP signals in *L. anatina*. The text describing this now reads:

“To validate the transcriptomic data, we examined expression of four key neural markers—*soxb1*, *soxb2*, *pax6* and *elav*—by *in situ* hybridisation in control, BMP(+) and BMP(-) embryos (Fig. 4d, Supplementary Table 29). In control embryos, all four genes were expressed ventrally, directly opposite the dorsal BMP maximum (Fig. 4d, top row). For *soxb1*, *soxb2*, and *pax6*, inhibition of BMP signalling caused a dorsal expansion of expression domains, with stronger effects under K02 compared to LDN treatment (Fig. 4d, second and third rows). In contrast, BMP(+) treatment abolished detectable expression of these three genes (Fig. 4d, bottom row). *Elav* expression was reduced under BMP(+), and showed a slight, though less clear, upregulation with K02, consistent with the RNA-seq data, where the effect on *elav* is less pronounced than other neural genes (Fig. 4c). Overall, these data show that key neural genes are expressed in the same location as *chordin*, diametrically opposed to the high point of BMP signal activation. Moreover, their expression domains expand when BMP signalling is inhibited and are eliminated when it is overactivated, confirming that BMP signals restrict the expression of neural genes.”

Additional comments and questions

Intro

line 52 Capitalize “ADMP” since it’s an abbreviation for “anti-dorsalizing morphogenetic protein”.

We thank the reviewer for the suggestion. However, we have opted not to capitalise “Admp,” as the use of all-uppercase protein or gene names is a convention specific to humans. Instead, we follow a naming convention increasingly adopted in invertebrate research, where

protein names begin with a capital letter and gene names are written in all lowercase italics (Gaviño et al. 2011; Kogure et al. 2022, among others). This usage is also becoming more common in some vertebrate studies (e.g., Yan et al. 2019). However, we have now included the full spelling of Admp as anti-dorsalizing morphogenetic protein at its first use.

References:

- Gaviño MA, Reddien PW. A Bmp/Admp regulatory circuit controls maintenance and regeneration of dorsal-ventral polarity in planarians. *Curr Biol*. 2011 Feb 22;21(4):294-9. <https://doi.org/10.1016/j.cub.2011.01.017>
- Kogure YS, Muraoka H, Koizumi WC, Gelin-Alessi R, Godard B, Oka K, Heisenberg CP, Hotta K. Admp regulates tail bending by controlling ventral epidermal cell polarity via phosphorylated myosin localization in *Ciona*. *Development*. 2022 Nov 1;149(21):dev200215. <https://doi.org/10.1242/dev.200215>
- Yan Y, Ning G, Li L, Liu J, Yang S, Cao Y, Wang Q. The BMP ligand Pinhead together with Admp supports the robustness of embryonic patterning. *Sci Adv*. 2019 Dec 18;5(12):eaau6455. <https://doi.org/10.1126/sciadv.aau6455>

lines 55–56 “Basal deuterostomes” would be the nodes before the terminal/extant taxa. I think you mean “early-branching deuterostomes”.

We thank the reviewer for pointing out our error, and have now changed all mentions of “basal deuterostomes” to “early-branching deuterostomes” as suggested. We have also specifically defined this term at its first use as: “echinoderms and hemichordates, together forming Ambulacraria”

Results and Figures

Fig. 2

In panels m, n, and o, there are some cells where the pSMAD1/5 labeling appears to only be on one side of the nucleus. Is this common for pSMAD1/5 labeling and if so, what does it indicate? I’m asking about the cells with a circular domain of labeling, not the cells with punctate spots of pSMAD1/5.

Thank you for highlighting this interesting observation. We also noticed this staining pattern and speculate that it may be related to the level of BMP signalling, with cells receiving lower BMP signals showing an uneven distribution of pSmad1/5. In our study, we did not quantify this pattern as it was outside the scope of our work, but a similar gradient of pSmad5 has been reported in zebrafish (Greenfeld et al., 2021 *Plos Biol*).

Reference:

- Greenfeld H, Lin J, Mullins MC. The BMP signaling gradient is interpreted through concentration thresholds in dorsal-ventral axial patterning. *PLoS Biol*. 2021 Jan 22;19(1):e3001059. <https://doi.org/10.1371/journal.pbio.3001059>

The view in panel k is not labeled (av?).

We have labelled it as lateral view (lv) (now Fig. 2i).

Is vegetal is down in panel l?

It is difficult at this stage to precisely locate the vegetal pole. Nevertheless, we have rotated the embryo to better align it with panel k (now Fig. 2j).

For panel o, I am not certain what I am looking at. Why is there a stripe of pSMAD on the ventral side of the larva? Is this a lateral view similar to Fig. 3k? A diagram of the tissues including the forming mantle and pSMAD in an early larva would be helpful.

Due to extensive tissue development and movement, the larval structure of *Lingula* is difficult to interpret in direct relation to the gastrula stage. We have now therefore provided the following schematics to help readers understand the staining pattern (Supplementary Fig. 21 and replicated as **Reviewer Fig. 1**). The opposite side of the larval vegetal view is shown in Fig. 2m, and new *in situ* data of BMP ligands and antagonists in the same orientation are presented in Fig. 2d.

Reviewer Fig. 1 | Schematics of *L. anatina* embryos at the early larval stage. a, Early larva in lateral view. **b**, One-pair-cirri larva in vegetal view. **c**, One-pair-cirri larva in rotated vegetal view.

Why do you think there is no pSMAD1/5 at cleavage stages even though there is maternal *bmp5-8* and *smad1/5* transcript and ubiquitous *bmp* receptor expression? Did you try blocking BMP signaling with the two drugs during cleavage stages (i.e., a drug pulse during early cleavage and assessment of resulting larval phenotypes)? Similarly, if you add BMP protein during cleavage stages, do you see pSMAD during these stages? Is it possible that there is a lower level of BMP signaling during cleavage that is not detected by the cross-reactive antibody? Is *bmp5-8* transcript spatially restricted to a subset of cells during cleavage?

On the first question, we reanalysed the confocal images and confirmed that no detectable signal was observed using the same staining protocol and imaging conditions. A pSmad gradient is detectable only at the blastula stage, but not at the early blastula stage (**Reviewer Fig. 2**). However, we cannot rule out the possibility that extremely low levels of pSmad are present but fall below the detection limit of the commercial antibody. Sequence alignment shows that the epitope is highly conserved—and in fact identical to the *L. anatina* sequence—and this antibody has been shown to cross-react with multiple species, including amphioxus, sea urchins, hemichordates, and sea anemones (Supplementary Fig. 22).

Reviewer Fig. 2 | Immunostaining with pSmad antibody and signal quantification. a, Immunostaining of *L. anatina* embryos using a pSmad1/5 antibody, with nuclear counterstaining by Hoechst 33342. Regions used for signal profiling are indicated with dashed boxes. **b,** Signal intensity profiling of pSmad staining in blastula-stage embryos, with quantification shown in b' using ImageJ. pSmad signals were absent at both the 32-cell and early blastula stages.

As for why pSmad signals are undetectable during the cleavage stage, we suspect this is due to the absence of *bmp2/4* expression. In molluscs, *Bmp2/4* has been shown to be the primary driver of Smad1/5 phosphorylation (Tan et al. 2022, *Mol Biol Evol*). We also note that *chordin*, which plays a central role in establishing the BMP signalling gradient (Genikhovich et al. 2015, *Cell Rep*), is not expressed at this stage, further suggesting that a BMP signalling gradient is unlikely to be present.

On the idea of re-doing the experiments with BMP manipulation at cleavage stages, we agree that this would be very interesting but for the reasons mentioned above, mainly the move of the lab and the limited spawning season and lack of a lab culture for *L. anatina*, it would significantly delay publication. Equally, since our experiments focussed later than the cleavage stage on the time point where pSmad1/5 signals were present, we do not have the data as it stands to say whether the *bmp5-8* transcript is spatially-restricted at this earlier stage. We do however, agree that these experiments would hold merit and, though outside the scope of the current manuscript, should certainly be prioritised as a next step.

Fig. 3

For the BMP manipulation experiments, the long time-window went from 5 hpf to ~24-27 hpf. What was the evidence that the BMP protein or the BMP inhibitors were active for that whole time in *L. anatina*? Was the BMP protein or BMP signaling inhibitors only applied once at 5 hpf? It would be useful to show pSMAD1/5 at early larval and 1 PCL stages for the long treatment times.

BMP proteins and their inhibitors are generally stable in seawater for several days, and most manipulation experiments in marine invertebrates are conducted within a 24 to 48-hour window. For example, in amphioxus, recombinant proteins were applied from 2.5 to 36 hours post-fertilisation (Yu et al. 2007 *Nature*); in sea urchins and hemichordates, treatments spanned 24 to 48 hours post-fertilisation (Su et al. 2019 *PNAS*); and in the mollusc *Lottia goshimai*, treatments were applied from 0 to 6 hours post-fertilisation (Tan et al. 2022 *Mol Biol Evol*). It is worth noting that the shorter treatment duration in molluscs reflects their

faster embryonic development, rather than limited activity of the applied proteins or small molecules.

In our experiments, treatments were initiated at the early blastula stage (5 hours post-fertilisation), before any visible pSmad signals were detectable (see **Reviewer Fig. 2**). For both short (5 to 10 hours post-fertilisation) and long (5 to 24 hours post-fertilisation) treatments, we assessed the efficacy of the applied recombinant proteins or inhibitors at the late gastrula stage using pSmad staining, confirming that BMP signalling was either completely suppressed or ectopically activated (Fig. 3c–f). At the larval stage, pSmad staining further confirmed that both the recombinant proteins and inhibitors remained active (**Reviewer Fig. 3**). We have now also included these additional data as Supplementary Fig. 24.

Reviewer Fig. 3 | BMP signalling readout in the larval stage detected by pSmad immunostaining. a, Schematic of the *L. anatina* developmental timeline (hours post-fertilisation, hpf) and experimental manipulation of BMP signalling from the early blastula (EB) to one-pair-cirri larval (1PCL) stage. B, blastula; EG, early gastrula; MG, mid-gastrula; LG, late gastrula; EL, early larva. **b**, Vegetal view of a 1PCL larva. **c**, Immunostaining of *L. anatina* embryos treated with small-molecule inhibitors or recombinant proteins, using a pSmad1/5 antibody and Hoechst 33342 nuclear counterstain.

References:

- Yu JK, Satou Y, Holland ND, Shin-I T, Kohara Y, Satoh N, Bronner-Fraser M, Holland LZ. Axial patterning in cephalochordates and the evolution of the organizer. *Nature*. 2007 Feb 8;445(7128):613-7. <https://doi.org/10.1038/nature05472>
- Su YH, Chen YC, Ting HC, Fan TP, Lin CY, Wang KT, Yu JK. BMP controls dorsoventral and neural patterning in indirect-developing hemichordates providing insight into a possible origin of chordates. *Proc Natl Acad Sci U S A*. 2019 Jun 25;116(26):12925-12932. <https://doi.org/10.1073/pnas.1901919116>

Tan S, Huan P, Liu B. Molluscan dorsal-ventral patterning relying on BMP2/4 and chordin provides insights into spiralian development and evolution. *Mol Biol Evol.* 2022 Jan 7;39(1):msab322. <https://doi.org/10.1093/molbev/msab322>

For the two BMP protein doses (low and high) and the two drug doses, did you test whether these doses resulted in different levels of pSMAD1/5 in *L. anatina*? Are the data shown in Fig. 3d–n after high dose treatment for the protein and the drugs? I don't see data comparing the different doses except effect on pH3, and the change in number of pH3+ cells between the low and high treatments does not look very dramatic compared to the shift from no treatment to the low treatments.

For the pSmad staining, treatment with either inhibitor resulted in complete loss of signal (Fig. 3d,e), while recombinant protein treatment produced widespread ectopic pSmad activation (Fig. 3f). As the phenotype was binary (presence or absence of signal), we did not perform quantification of pSmad-positive cells, in contrast to the quantitative analysis we carried out for pHistone H3 staining. Signal intensity was also not quantified in recombinant protein-treated embryos. Our interpretation is that even the low-dose treatment is sufficient to saturate the response, such that no clear difference can be observed between low and high doses.

How was the overall morphology including the dorsal-ventral axis affected by up or downregulating BMP signaling? You state that gastrulation was delayed and that mantle lobe folding was disrupted, but a description of additional phenotypes would be useful for understanding how manipulating BMP signaling affected development and formation of fates along the dorsal-ventral axis.

We have expanded the description in the main text as follows:

“ BMP(-) treatment results in a shortened archenteron (Fig. 3d) or a complete delay in invagination when using a stronger BMP inhibitor (Fig. 3e). In contrast, BMP(+) treatment delays invagination but promotes ingression (Fig. 3f).”

and

“This suggests that the depressions that split the mantle fold into two lobes⁹¹ during normal development have failed to form properly. Meanwhile, BMP(+) embryos exhibited extended shell domains, with the chitinous shell extending ventrally, resulting in the tentacle and mouth being encased inside the shell (Fig. 3n). The lack of a clear opposite effect on shell formation in BMP(-) and BMP(+) embryos may be because the effects of BMP signalling on mantle lobe and shell formation are a product of interference with embryonic morphogenetic processes (e.g., the formation of the furrows that split the mantle fold) rather than a direct promotion or inhibition of shell formation. The disturbance of the distribution of proliferative cells may also contribute to this process.”

What is the view in Fig 3n?

This is a lateral view similar to Fig. 3k–m, but slightly rotated to bring the ventral side partially into view. We have added this clarification to the figure legend.

It would be helpful to include a sentence explaining where in the BMP signaling pathway LDN-193189 (LDN) and K02288 (K02) act. Since both drugs inhibit multiple ALK receptors, what is the evidence that the dose you are using is selective for BMP signaling and not Activin signaling in *L. anatina*? Did you test the effects on pSMAD1/2? Similarly, for the BMP protein treatments, did you test whether you were cross-activating Activin signaling (SMAD1/2)? It's important to demonstrate the selectivity for perturbation of only BMP signaling since Activin signaling has been shown to be an organizer signal in at least one other spiralian.

We agree with the reviewer that it would be helpful to include a sentence explaining where in the BMP signalling pathway our two inhibitors act. We have now added the following sentence to the methods section:

“For the BMP(-) condition, two small molecule inhibitors, LDN193189 (LDN; Stemgent 04-0074)⁸⁵ and K02288 (K02; Tocris 4986)⁸⁶, were used to block the BMP pathway by inhibiting type I receptors for BMP ligand proteins.”

We are also grateful to the reviewer for highlighting the potential off-target effects on activin signalling of the two inhibitors used. Indeed, while the literature shows that they are both highly specific to BMP signalling over TGF β (with 200-300 fold difference for LDN [Yu et al. 2008 *Nat Med*, Yu et al. 2008 *Chem Biol*] and K02288 (K02) showing no effect at all [Sanvitale et al. 2013 *PLOS One*]), there is a weak effect on activin signalling (Sanvitale et al. 2013 *PLOS One*).

However, though we did not test the effects of the treatment on Smad2/3 phosphorylation, our RNA sequencing data shows that neither of the *L. anatina* activin/inhibin genes are expressed at this stage of development (**Reviewer Fig. 4**). Furthermore, the two other TGF β -like ligands we identified, Gdf15 and Myostatin also show negligible gene expression at the time of our experiments. The lack of activin gene expression and high specificity of the inhibitors for BMP signals over TGF β mean that we are confident that the observed effect is caused by perturbations to BMP signalling and not these other signalling pathways.

This is corroborated by the fact that our BMP(-) and BMP(+) treatments have corresponding opposite effects and the BMP(+) treatment is BMP-specific given it involves solely the addition of mBMP4. Moreover, we used two independent inhibitors as a further control to verify that the phenotypes observed are not the consequence of some unique effect of a specific inhibitor, and in every case both inhibitor molecules produce highly similar results.

Reviewer Fig. 4 | Expression of TGF β -like ligands including activin/inhibin A and B during *L. anatina* development. All four TGF β -like genes identified in this work are expressed only to negligible levels at the time of experimentation.

Despite this, we agree with the reviewer that this was unclear in our previous submission of the manuscript and have now explicitly stated this logic in the text. In addition, we have added a further two supplementary figures to show firstly a tree of activin/inhibin ligands (Supplementary Fig. 30) and secondly the activin/inhibin gene expression is at negligible levels during the time of our experiments (Supplementary Fig. 31). The new text reads:

“Two independent inhibitors were used as controls to confirm that the observed phenotypes resulted from BMP signal perturbation rather than being specific to a particular inhibitor. Though both molecules are highly specific to BMP receptors as opposed to TGF- β , and K02 has been reported to have no effect on TGF- β -induced Smad2 phosphorylation, they may have a weak off-target effect on Activin signalling^{86,190,191}. To check the potential for off-target effects on TGF- β and particularly activin signalling in our system, we annotated the two *activin/inhibin-like* genes and other TGF- β -like genes in *L. anatina* (Supplementary Fig. 30) and quantified their expression during embryonic development. All TGF- β -like genes including the two *activin/inhibin-like* genes are either not expressed or expressed at negligible levels at the time of experimentation (Supplementary Fig. 31), suggesting that this pathway is not active at this stage of development and therefore that the effects of LDN and K02 are not attributable to disturbed activin signalling.”

Fig. 4

For the functional transcriptomic experiments, did you find differences in which genes were up and down-regulated between the 100 and 200 ng/mL BMP treatments or between the low and high drug concentrations? Do you think these low and high concentrations are both maximally upregulating/blocking BMP signaling, or did you find evidence for BMP acting as a morphogen in *L. anatina*?

We thank the reviewer for raising this because it highlights a lack of proper explanation in the text. The transcriptomic response to the high and low treatments were very similar and indistinguishable using hierarchical clustering (Fig. 4a). For this reason, we believe as the reviewer suggested that both treatments are maximally upregulating/blocking BMP signaling. As a result, for the transcriptomic analyses we therefore integrate both the high and low treatments as a single dataset to increase the sample size.

To ensure that this is better explained and easily understood, we have now added the following sentence into the methods section of the manuscript: “The transcriptomes of samples receiving high versus low doses of each manipulator were indistinguishable using hierarchical clustering (Fig. 4a) so are combined and treated as a single condition in subsequent analyses.”

We also changed the sample names in Figure 4a to make it clearer which samples correspond to which treatment.

Fig. 5

In panel e, the z-scores for some genes (e.g., *shh*, *twsg1*, *bmp5*, *bmp4*, *fst*...) seem to vary across replicates within a treatment. Is this due to low expression levels for these genes?

We agree with the reviewer that it would be useful for readers to know the expression levels of these genes. We have therefore added an additional part of the figure showing the mean expression level in the experiment for each gene. The majority of genes are highly expressed, including *twsg1* (now renamed *twsg*), *bmp5* (now renamed *bmp5-8*) and *bmp4* (now renamed *bmp2/4*). A few genes, especially those in the bottom section including *fst* (now renamed *folliculin*), do show low expression levels, which may contribute to their varied Z-scores. We have now focussed our discussion of this in the text only on those genes that are highly expressed.

In Extended Fig. 9, the up-regulation of *bmp2/4* expression in the ectoderm after LDN treatment is not very clear to me in the animal shown.

In situs for BMP ligands following treatment conditions have now been removed from the manuscript as they were not central to our major biological questions.

Discussion

Be careful of inferring gene function in *L. anatina* based on what a gene homolog does in chordates. For example, in lines 515–516, you discuss “ventralising factors, particularly chordin”. However, the function of chordin in *L. anatina* has not been tested to my knowledge.

We agree with the reviewer that our previous draft of the manuscript was on several occasions insufficiently precise when discussing *L. anatina* homologues of genes investigated in other species but not directly in *L. anatina*. To rectify this specific example, we have inserted the phrase “associated with neural expression in several model organisms”, to highlight the fact that this function is based on work in different species.

In addition, we have aimed to clear the paper of other examples of this by carefully rephrasing several other sentences. In particular, as raised by Reviewer 1, there were similar problems when discussing neural genes, and these have now all been corrected. We hope that the new manuscript draft is now more precise when discussing *L. anatina* homologous genes known from other species.

Methods

line 589 It looks like multiple tissues were collected for genome sequencing, but only mantle, lophophore, and adductor muscle were used. Was the DNA in the other tissues degraded?

We initially extracted genomic DNA from multiple tissues, including the lophophore, mantle, adductor muscle, gonad and pedicle. However, only the lophophore, mantle and adductor muscle yielded high-quality, high-molecular-weight DNA that met our quality control criteria (intact band around 100 kb, no signal below 40 kb, NanoDrop A260/280 = 1.8–2.0 and A260/230 = 2.0–2.2, and a Qubit-to-NanoDrop concentration ratio between 0.8 and 1.2). DNA from the gonad and pedicle was degraded or failed to meet one or more of these criteria and was therefore excluded. To meet the 15 µg input requirement for library construction, we pooled 5 µg of DNA from each of the three qualified tissues. We have revised the Methods section accordingly and removed unnecessary details about excluded tissues to improve clarity.

line 596 For genome sequencing, did you assay DNA integrity using BioAnalyzer in addition to assessing purity with a NanoDrop?

While we did not assess DNA integrity using a BioAnalyzer, we used conventional gel electrophoresis, which allowed us to evaluate fragment size and detect potential degradation. We have clarified this point in the Methods section with the following addition:

"Genomic DNA quality was assessed by gel electrophoresis. Samples meeting the quality criteria showed a main band centred around 100 kb with no detectable signal below 40 kb. DNA purity was evaluated using NanoDrop (A260/280 = 1.8–2.0; A260/230 = 2.0–2.2), and DNA concentration was measured using both NanoDrop spectrophotometer and the Qubit 4.0 Fluorometer."

line 601 & 738 Did you do any column purification after RNA extraction by TRIzol?

We did not perform column purification after RNA extraction with TRIzol. Relevant details have been provided in the Methods section as follows.

"Total RNA from each tissue sample was used directly for mRNA enrichment and library preparation using the TruSeq RNA Library Prep Kit v3, following the manufacturer's protocol. Briefly, mRNA was purified using oligo(dT)-attached magnetic beads, followed by fragmentation and first-strand cDNA synthesis with random hexamer primers. Second-strand synthesis was then performed, and the resulting cDNA underwent end repair, A-tailing, adapter ligation, size selection, amplification and purification to complete library construction."

line 742 Why was edgeR used versus DESeq2, and what were the criteria for using a dispersion parameter of 0.1?

We thank the reviewer for this technical comment. For datasets with only a single replicate per condition (due to historical constraints), we used edgeR with a fixed dispersion value of 0.1 to enable conservative differential expression analysis. For the newly sequenced datasets with biological replicates, we used the voom method, which estimates the mean–variance relationship directly from the data (Law et al. 2014). These choices were based on sample size and follow standard practice for RNA-seq analysis. We have clarified this point in the Methods section.

The revised Methods section now reads:

“For datasets without biological replicates, edgeR was used with a fixed dispersion value of 0.1 to enable conservative inference. For newly generated datasets with biological replicates, the voom method was applied to model the mean–variance relationship prior to linear modelling.”

Reference:

Law CW, Chen Y, Shi W, Smyth GK. voom: Precision weights unlock linear model analysis tools for RNA-seq read counts. *Genome Biol.* 2014 Feb 3;15(2):R29.
<https://doi.org/10.1186/gb-2014-15-2-r29>

Reviewer #3

We thank the Early Career Researcher for co-reviewing our manuscript and support this initiative to provide training opportunities.

Reviewer #4

The role of BMP signaling in patterning the dorsoventral body axis has been described in a diversity of species, including deuterostomes, ecdysozoans and spiralian, and is widely considered to be the ancestral bilaterian state. While initial studies in a handful of model organisms, primarily in vertebrates and insects, suggested BMPs are required to inhibit neural induction in early embryogenesis, recent work in emerging model systems, particularly those found in the Spiralia, suggest a more complex story where neural induction is independent of BMP signaling in multiple species (e.g. Lambert et al., 2016; Webster et al., 2021; Webster & Meyer, 2024).

In the manuscript “Brachiopod genome unveils the evolution of the BMP-Chordin network in bilaterian body patterning” Lewin et al., present a chromosomal-scale genome for the brachiopod *Lingula anatina*. This genome improves on a previous scaffold-level assembly for the same species and extends observations from this previous publication from a number of these authors that BMP patterning may be involved in shell formation in brachiopods. While the chromosome-level genome will prove to be a key resource for future work in this clade, the study largely focuses on the role of BMP signaling in this species. The authors employ the genome to describe the complement of BMP signaling pathway members in this species as well as other spiralian. The experiments presented, including pharmacological approaches and RNA seq, confirm a role for BMP in patterning the dorsoventral body axis, which has been suggested in the previous genome description and was previously described in other spiralian, including two other brachiopod species, *T. transversa* and *N. anomala* (Martin-Duran et al., 2016).

We thank the reviewers for summarising the key studies in the field, recognising the importance of our study, and providing thorough and constructive feedback on our manuscript. Below, we address each of their comments in detail:

Based on differential expression in RNAseq datasets generated from pharmacologically up and down regulating BMP signaling, the authors suggest that BMP also inhibits neural development in *L. anatina*. They find dozens of genes to be differentially regulated upon manipulation of BMP signaling, including a number of neural-associated genes. Based on this finding, the authors conclude that BMP is required to inhibit neural development, a role well described in vertebrates and arthropods. However, the experimental evidence for this assertion is not sufficient, and indeed, expression patterns for several neural genes contradicts this hypothesis. For example, *elav* is unchanged or even slightly downregulated in BMP(-), while *soxB1* is upregulated in both BMP(-) and BMP(+) embryos. Furthermore, the (differential) expression of neural genes should also be confirmed using *in situ* hybridization assays to visualize where these genes are expressed in relation to the BMP signal and potential changes in expression domain after treatment.

We are grateful to the reviewers for highlighting the need for more robust experimental evidence to support our conclusion that BMP inhibits neural development in *L. anatina*. We agree that this is an important test of the manuscript’s main findings, and therefore conducted *in situ* hybridisation to visualise the expression patterns of four neural genes (*soxB1*, *soxB2*, *pax6* and *elav*) to verify the effects of BMP signalling in neurulation. In control embryos, all four genes are expressed at the ventral side of the embryo directly opposite the pSmad1/5 maximum, supporting the contention that they are restricted by BMP signals. To directly test this, we also conducted *in situ* hybridisation in embryos treated by our BMP signal manipulation procedures. These clearly show that expression of all four genes is eliminated in the BMP(+) condition and expanded in the BMP(-) condition (though we note that the effect is less pronounced for *elav* than the other three genes). This provides strong, independent evidence supporting our claim that BMP signals repress the expression of neural genes. This is now described in the following passage in the Results:

“To validate the transcriptomic data, we examined expression of four key neural markers—*soxB1*, *soxB2*, *pax6* and *elav*—by *in situ* hybridisation in control, BMP(+) and BMP(-) embryos (Fig. 4d, Supplementary Table 29). In control embryos, all four genes were

expressed ventrally, directly opposite the dorsal BMP maximum (Fig. 4d, top row). For *soxb1*, *soxb2*, and *pax6*, inhibition of BMP signalling caused a dorsal expansion of expression domains, with stronger effects under K02 compared to LDN treatment (Fig. 4d, second and third rows). In contrast, BMP(+) treatment abolished detectable expression of these three genes (Fig. 4d, bottom row). *Elav* expression was reduced under BMP(+), and showed a slight, though less clear, upregulation with K02, consistent with the RNA-seq data, where the effect on *elav* is less pronounced than other neural genes (Fig. 4c). Overall, these data show that key neural genes are expressed in the same location as *chordin*, diametrically opposed to the high point of BMP signal activation. Moreover, their expression domains expand when BMP signalling is inhibited and are eliminated when it is overactivated, confirming that BMP signals restrict the expression of neural genes.”

Throughout the manuscript, the authors discuss the roles of BMP in DV-axis patterning and in neural development as one common feature, with one being dependent on the other. However, recent work in other spiralian indicates that these can be two distinct roles that can be present independently of each other. The authors imply that because they see an effect on dorsoventral patterning, a role in neural development must be the ancestral state and vice versa. This discussion would benefit from clear delineation and test of these different roles, and would greatly benefit from an attempt to differentiate these functions experimentally. Indeed, recent work in spiralian suggests a more complex story for the role of BMPs in neural development, with functional studies in both annelids and molluscs demonstrating neural development independent of BMP signaling. This needs to be discussed in more detail, taking into account different scenarios like convergent evolution or loss of traits.

We thank the reviewer for raising this issue. It was not our intention to conflate these two ideas and, indeed, the reason we highlight our findings with respect to BMP downregulation of neural genes is precisely because we see this as potentially independent from dorsal-ventral patterning (though of course, in many cases they are tightly intertwined). For instance, our intention was not to imply that because we see an effect on dorsoventral patterning, a role in neural development must be the ancestral state. Instead, this inference comes from our RNA-seq analysis (now backed up with *in situ* hybridisations) which shows BMP-mediated suppression of neural genes.

We recognise the importance of rectifying this, and have now sought to clarify it throughout the manuscript. The place in the previous version of the manuscript that was most problematic and unclear on this topic was the L549 - 557 paragraph in the Discussion. We have now completely re-written this to highlight the distinction. In this paragraph, we also now mention the possibilities of convergent evolution and secondary modification/losses. The new paragraph reads:

“A role of BMP signals that is closely linked to but potentially independent from dorsal-ventral patterning is neural induction. Across both deuterostomes and ecdysozoans, BMP signals repress the formation of neural tissue⁴⁶. One significant observation casting doubt on whether the ancestor of spiralian utilised BMP signals during early development in the same way as ecdysozoans and deuterostomes⁵¹ was the discovery that, in some molluscs and annelids, BMP signals have no effect or even a positive effect on neural development^{48,55-57}. Through functional transcriptomics of embryos with manipulated levels

of BMP signalling, we found strong inhibition of genes associated with neural expression in several model organisms by the *L. anatina* BMP pathway during development. *In situ* hybridisation in BMP-manipulated embryos confirmed that the expression domains of these genes are restricted by BMP signals *in vivo*. These results suggest that the function of BMP signalling in restricting neural induction is similar in some spiralian to that in arthropods and deuterostomes. One possibility is that this function was ancestral to all three major bilaterian clades, but several spiralian lineages modified or replaced it (annelids and molluscs, for instance) while it was conserved in brachiopods. It is, however, impossible to rule out from these data that BMP-mediated suppression of neural genes evolved convergently in brachiopods from a divergent state in the ancestor of spiralian.

Furthermore, we added a sentence into the Introduction to highlight the potential independence of these two features to make sure that this is in the reader's mind from the outset of the manuscript. "This suggests potential independent effects of BMP signals on dorsal–ventral fate and neural induction."

Some additional concerns and comments:

1. The role of Chordin: The title and abstract of the manuscript refer to the role of the BMP-Chordin network. While the manipulation of BMP signaling seems to affect expression levels of Chordin (whether directly through transcriptional regulation or indirectly through changes in size of the ventral domain is also not addressed), the actual role of Chordin in body axis patterning and neural development is not directly investigated.

We thank the reviewers for pointing out this discrepancy. In response, we have changed instances of the phrase "BMP-Chordin" to "BMP signalling" in the title and abstract. The new title reads:

"Brachiopod genome unveils the evolution of BMP signalling in bilaterian body patterning"

We have also carefully modified the rest of the text to avoid this problem.

2. Expression of BMPs and pSMAD signal

Using antibody staining, the authors detect phosphorylated SMAD1/5, a read-out for BMP signaling activity, at the dorsal side of the embryo (Figure 2). In Extended Data Figure 9, they also show the expression of *bmp2/4*, *bmp3* and *bmp5-8*, using ISH. This expression data should be included in Figure 2. The expression data for *bmp2/4* and *bmp5-8* expression raises concerns. While there is a dorsal signal that likely overlaps with pSMAD1/5 signal, *bmp2/4* and *bmp5-8* also seem to be expressed more globally throughout the embryo. Can the authors provide an explanation for this, or ISH images that are easier to interpret for non-brachiopod specialists?

We apologise to the reviewers for two problems with the previous version of the manuscript: (1) poor *in situ* quality, and (2) insufficient explanation of this result. To rectify these issues we have redone the *in situs*, moved all the results to Figure 2 as suggested, and expanded the Introduction and Discussion sections to better explore this finding.

In the new *in situs*, we see expression of BMP ligands at both ends of the gradient (Figure 2d). *Bmp3* is clearly dorsal and *admp* clearly ventral while *bmp5-8* is more widely expressed and *bmp2/4* is less clear, but appears to be dorsal at the gastrula stage.

The key to explaining this, which was missing from our previous version of the manuscript, is that it is known that it is the position of *chordin* expression, not that of the BMP ligands themselves, that is the key determinant of the BMP signalling gradient (Genikhovich et al. 2015 *Cell Rep.*). This is because Chordin acts as a shuttle, transporting BMP molecules away from the location at which it is expressed (Mizutani et al. 2005 *Dev Cell.*). Across bilaterians and cnidarians, the expression patterns of BMP signals vary widely and are even sometimes coexpressed with chordin, but the BMP signalling maximum always forms at the non-Chordin end. We have now provided a much more comprehensive description of this in the Introduction of our revised manuscript, including the concept of a BMP seesaw suggested by Reviewer 1, and then explore it further in the Discussion section.

3. Role of BMP in cell proliferation

In Figure 3, the authors describe an effect of BMP manipulation on cell proliferation. This observation should be discussed in more detail in the manuscript. What cell types are affected by this role? How does this compare to known roles of BMP signaling in other systems? How could this be related to other observed phenotypes?

We agree that discussion of this result was lacking and have now rectified this. First, in the section describing Figure 3, we have added the following text:

“This is consistent with dose-dependent effects of BMP signals reported from vertebrate development, where low doses promote proliferation while high doses inhibit proliferation and induce terminal differentiation^{78–80}, and suggests that a similar effect may be exerted during *L. anatina* development.”

We also found in our RNA-seq analysis that cell cycle genes were widely downregulated by BMP signalling. This is consistent with the Histone H3 staining reported in Figure 3 but in our previous version this was insufficiently described. We have now completely re-written the paragraph discussing this aspect of the RNA-seq results to integrate these two aspects of the manuscript:

“In addition, at the gastrula stage, each of the 12 most statistically significant GO terms for BMP-downregulated genes relates to processes involved in cell cycle regulation (Supplementary Table 30), including ‘DNA replication’, ‘chromosome organisation’ and ‘cell cycle’ (Fig. 4e). BMP-downregulated genes include *atr*, *cdc16*, *ctf18*, *pcna* and *pole*, all of which encode essential components of the cellular DNA replication machinery^{102–107}. All six *mcm2-7* genes, encoding the core components of the eukaryotic replicative helicase complex¹⁰⁸, are also BMP-downregulated. Conversely, no GO terms associated with BMP-upregulated genes relate to DNA replication or the cell cycle (Supplementary Table 25). This is consistent with the above result that the presence of mitotic cells was reduced in BMP(+) embryos and increased in BMP(-) embryos, and supports the contention that BMP signals inhibit cellular proliferation at this time point. A role for BMPs in regulating proliferation is well-known but the extent and direction of effect is highly context-dependent¹⁰⁹.”

Additionally, in the modifications made in response to the next comment we also mention the possible interaction between the cellular proliferation and shell formation phenotypes.

4. Role of BMP in shell formation

In Figure 3, the authors describe a role of BMP signaling in establishing the embryonic shell fields. They conclude that BMP is required to divide the shell field into two valves, as the fields seem to be fused in BMP(-) embryos. However, in BMP(+) embryos the shell field seems to be expanded and experience over-folding, rather than lost as might be expected if BMP indeed inhibits shell field formation. How can these results be explained?

We agree with the reviewer that the BMP(+) phenotype is somewhat surprising given the failure to divide the shell field in BMP(-) embryos. Our interpretation of this is that BMP signals do not directly inhibit or promote shell formation but are rather key to the orchestration of developmental processes (e.g. the formation of furrows which split the mantle fold into two lobes) which, when interfered with, manifest as defects of shell formation. This may, for instance, be a product of more/less cellular proliferation in the BMP(-)/(+) treatments, respectively. We have now modified this section of the manuscript in order to raise this possibility but also highlight our uncertainty based on current data. Additionally, we (1) added more context from previous studies describing this part of *L. anatina* development and (2) concluded that the 'overfolding' terminology is potentially misleading and removed it. The new text reads:

“During *L. anatina* development, the embryo develops a mantle fold which is subsequently split into two mantle lobes that go on to form the two shells (Fig. 3b)⁸¹. We assessed the necessity of BMP signalling to this process by manipulating the BMP pathway and observing embryonic shell formation with a chitin-binding probe. In control embryos, the larval shell is marked by chitin staining across both mantle lobes, excluding the dorsal edge (Fig. 3k). In contrast, BMP(-) embryos showed fused shell fields and a single, unseparated shell, indicated by continuous chitin staining, including at the dorsal edge (Fig. 3l-m). This suggests that the depressions that split the mantle fold into two lobes⁸¹ during normal development have failed to form properly. Meanwhile, BMP(+) embryos exhibited extended shell domains, with the chitinous shell extending ventrally, resulting in the tentacle and mouth being encased inside the shell (Fig. 3n). The lack of a clear opposite effect on shell formation in BMP(-) and BMP(+) embryos may be because the effects of BMP signalling on mantle lobe and shell formation are a product of interference with embryonic morphogenetic processes (e.g. the formation of the furrows that split the mantle fold) rather than a direct promotion or inhibition of shell formation. The disturbance of the distribution of proliferative cells may also contribute to this process. Overall, these experiments indicate that dorsal BMP signalling is essential for splitting the developing larval shell into two distinct valves and, more broadly, maintaining embryonic structural integrity and proper morphogenesis.”

5. RNA sequencing

The RNA sequencing data require further confirmation and discussion: 1) How much overlap is there between DEGs in BMP(+) and BMP(-) embryos? Are the same pathways/genes affected? 2) Is there evidence in the RNA sequencing data for the described role in cell proliferation and shell formation/folding? What are affected signaling pathways? 3) How reliable is GO term analysis in this species? 4) As mentioned above, the changes in neural gene expression require further confirmation/investigation (see above). 5) The authors

describe two different incubation windows used for BMP manipulation. It is not clear which one of these was used for RNA sequencing.

1) We thank the reviewer for this comment because it highlights a lack of explanation on our part as to how this analysis was carried out. The upregulated and downregulated gene sets are produced as a combination of both the BMP(-) and BMP(+) treatments. For instance, genes downregulated in the BMP(-) treatment and upregulated in the BMP(+) treatment are considered BMP-upregulated, and vice versa. Therefore, by definition, all the considered DEGs are oppositely regulated in the (+) and (-) treatments. Though this was stated in the methods, it was definitely unclear in the Results section. To rectify this, we have now (1) rewritten the first part of this section of the Results and (2) added labels to Fig. 4a to highlight this gene set (**Reviewer Fig. 5**). The new text reads:

“Differential gene expression and hierarchical clustering analyses identified gene sets regulated by BMP signalling. Genes downregulated under BMP(-) treatment and upregulated under BMP(+) treatment were classified as BMP-upregulated, whereas genes showing the opposite pattern were categorised as BMP-downregulated.”

Reviewer Fig. 5 | Heat map of 881 differentially expressed genes at the late gastrula stage under the manipulation of BMP signals. New main text Fig. 4a highlighting the fact that the BMP-upregulated gene set consists of genes upregulated in the BMP(+) condition and downregulated in the BMP(-) condition and *vice versa*.

2) We thank the reviewer for this recommendation. We have mined our enriched GO term dataset for evidence that cell proliferation and shell formation/folding is affected by manipulations to the BMP signaling pathway. This was particularly fruitful with respect to cell proliferation, where each of the top 12 GO terms for genes downregulated by BMP signalling at the late gastrula stage is related to DNA replication or the cell cycle. This is consistent with the pHistone H3 staining, which increased significantly when BMP signalling was inhibited and vice versa. We have now created a small new paragraph in the Results section

to discuss this result, and inserted an additional supplementary table (Supplementary Table 30) containing this list of cell cycle-related GO terms. The new text reads:

“In addition, at the gastrula stage, each of the 12 most statistically significant GO terms for BMP-downregulated genes relates to processes involved in cell cycle regulation (Supplementary Table 30), including ‘DNA replication’, ‘chromosome organisation’ and ‘cell cycle’ (Fig. 4e). BMP-downregulated genes include *atr*, *cdc16*, *ctf18*, *pcna* and *pole*, all of which encode essential components of the cellular DNA replication machinery^{102–107}. All six *mcm2-7* genes, encoding the core components of the eukaryotic replicative helicase complex¹⁰⁸, are also BMP-downregulated. Conversely, no GO terms associated with BMP-upregulated genes relate to DNA replication or the cell cycle (Supplementary Table 25). This is consistent with the above result that the presence of mitotic cells was reduced in BMP(+) embryos and increased in BMP(-) embryos, and supports the contention that BMP signals inhibit cellular proliferation at this time point. A role for BMPs in regulating proliferation is well-known but the extent and direction of effect is highly context-dependent¹⁰⁹.”

With respect to signalling pathways, the clearest effect is that Wnt signalling is downregulated by BMP signalling when assayed at the larval stage of development.

3) The method by which we conducted the GO term analysis is highly reliable, using BLAST searches to identify human orthologues for *L. anatina* genes and retaining only reciprocal best hits. We elected to use this procedure because the human GO datasets are the best curated and most detailed, giving the most accurate results for universal animal processes (for instance, cell proliferation). However, the cost to this is that it is not able to identify enrichment of genes for *L. anatina*-specific processes like shell formation.

4) Neural gene *in situs* have now been performed as described in the above responses.

5) For the RNA-seq experiments, embryos were treated from the early blastula stage (5 hours post-fertilisation) to the late gastrula stage (10 hours post-fertilisation). This information has been added to the figure legend.

6. The Lophophorata hypothesis: The authors present a phylogenetic analysis in Fig1b that suggests that brachiopods are in a clade with phoronids and bryozoans to form the “Lophophorata”. While this is a clade proposed in historic analyses, recent phylogenomic work suggests that bryozoans may be sister to entoprocts or cycliophorids, which the authors suggest may be the result of including orthologs with lower phylogenetic signal. However, these results are difficult to interpret as the authors did not include entoprocts or cycliophorids in their analysis. Could the authors search for their gene set in the deposited data for these species to provide support for their hypothesis? Or, could they leverage their new chromosome-level assembly to search for shared derived syntenic relationships between brachiopods and bryozoans that might support their hypothesis?

We thank the reviewer for this comment as they are both very interesting proposals.

To address the first, we built a phylogeny incorporating transcriptome data from members of the phyla Entoprocta and Cycliophora, which is now presented as Supplementary Figure 6. This dataset also supports a monophyletic Lophophorata.

With regards to the second proposal, we actually had the same idea to use the chromosome-level assembly of *Lingula* to search for derived syntenic changes between bryozoans and brachiopods. Our paper describing this analysis was recently published in *Genome Research* (<https://doi.org/10.1101/gr.279636.124>). In this work, we found evidence for nine linkage group fusion events shared between bryozoans and brachiopods, which is consistent with a close phylogenetic relationship between these two groups. We interpret this data as evidence supporting the Lophophorata hypothesis. However, chromosome-level assemblies of members of Phoronida, Cycliophora and Entoprocta are needed to confirm this.

Finally, we have opted to scale back our claims regarding the Lophophorata hypothesis since that is not the main subject of this manuscript and may generate unnecessary distraction and controversy.

7. Terminology

The authors make references to “basal deuterostomes” throughout the manuscript. This should be corrected to “basally-branching deuterostomes” or referred to as the phylogenetic group specifically (e.g. ambulacrarians).

We have now changed all mentions of “basal deuterostomes” to “early-branching deuterostomes”. We have also specifically defined this term at its first use as: “echinoderms and hemichordates, together Ambulacraria”

8. Does the proposed karyotype comport with expectations in this group?

Karyotype observations in *L. anatina* reported $n = 10$, consistent with our assembly. In our manuscript, the text reads:

“The majority of the assembly consists of 10 chromosome-scale scaffolds accounting for 97.8% of the total length, consistent with previous karyotype observations in *L. anatina* ($n = 10$)⁴².”

The reference is: Nishizawa, A., Sarashina, I., Tsujimoto, Y., Iijima, M. & Endo, K. Artificial fertilization, early development and chromosome numbers in the brachiopod *Lingula anatina*. *Special Papers in Palaeontology* 84, 309–316 (2010).

Other minor comments:

Main Figures

Figure 1, Panel D: Why is the Hox cluster pointed out specifically? It is hardly mentioned in the text. The lower half of this panel should be discussed and explained in the figure legend.

We agree with the reviewer that the location of the Hox cluster was not specifically relevant and have now removed it from the figure. We have also added labels of ‘Lophotrochozoan shared fusions’ and ‘Bilaterian ancestral linkage groups’ to the two parts of the in-figure legend.

In addition, we have expanded the figure caption and separated it into 'upper' and 'lower' parts for clarity:

“Upper: Chromosome-scale gene linkage observed between amphioxus, scallops and brachiopods. Horizontal bars denote chromosomes. Dark grey chromosomes indicate ancestral spiralian fusion events, while light grey chromosomes represent lineage-specific fusion events. Vertical lines between chromosomes link the genomic positions of orthologous genes. Lines colour-coded based on bilaterian ancestral linkage groups (ALGs)⁴¹. Lower: summary of lophotrochozoan shared fusions (H⊗Q, J2⊗L, K⊗O2 and O1⊗R) and colour code for bilaterian ALGs.”

Figure 1, Panel E: This panel requires more explanation.

We have now expanded the explanation of this panel:

“BMP signalling pathway genes are consistently associated with the same ALG (coloured blocks) across the five species shown in (d), with the exception of noggin-like, which is always found on chromosomes associated with both ALGs H and Q.”

Figure 2, Panel D-F: These could go into the supplement.

We have moved panels D and E to the supplementary material as suggested. Further, in response to a comment from Reviewer 1, panel F has been completely removed.

Figure 2, Panel G-O: Labeling of body axis and potentially an embryo schematic would be helpful here. Expression of BMPs should be added to this Figure.

As suggested, we have added new *in situ* data showing BMP ligand expression to this figure (Fig. 2e), along with an embryo schematic (Fig. 2c) and body axis labelling (Fig. 2l). In addition, given the complexity of larval morphology, we have included a schematic of the larval stage as Supplementary Fig. 21.

Figure 3, Panel K-N: Overlaying the chitin staining with the pSmad staining would be helpful to confirm that BMP signaling is active in the Chitin-negative regions.

We thank the reviewer for the suggestion. Unfortunately, we do not have double staining data for this stage. Nevertheless, we have added new pSmad staining data for the larval stage, now included as Supplementary Fig. 24, showing staining in the mantle and in potential chitin-negative regions (see **Reviewer Fig. 3**).

Figure 3, legend c-f: What do the empty arrowheads indicate?

We have clarified in the legend that blank arrowheads indicate ingressed mesenchymal cells located in the blastocoel, beneath the blastodermal epithelium.

“Nuclearised pSmad1/5 signals are marked by arrowheads and empty arrowheads in blastodermal and ingressed mesenchymal cells, respectively.”

Figure 3, legend k-n: What does the arrowhead indicate?

We have clarified in the legend that the arrowhead indicates the dorsal edge of the larva, which lacks chitin staining.

“The arrowhead marks the dorsal edge of the larva, which lacks chitin staining in the control (k).”

Figure 4, Panel A-H: Panels A and G-H require additional explanation or should be presented in a more intuitive way. Panels B-D do not provide crucial information and could be moved to Supplement.

Having looked back at our previous text and figure legend we agree that these panels were insufficiently explained. To correct this, we have expanded the main text at the point where this figure is introduced and completely rewritten the figure legend:

Main text: “ To investigate this further, we wanted to use a highly robust, conservative dataset of genes for which we had very strong evidence for regulation by BMP signalling. To this end, we restricted the dataset to only genes that were BMP-regulated in both our late gastrula and early larval stage experiments, resulting in a total of 49 BMP-upregulated and 45 BMP-downregulated genes (Fig. 4b, c, Supplementary Fig. 26).”

Figure legend: “**b, c** Restricted dataset of genes with strong evidence for being regulated by BMP signalling in the *L. anatina* embryo. Venn diagrams show the overlap of BMP-upregulated (**b**) and BMP-downregulated (**c**) genes in the two experiments we conducted (late gastrula and early larva). Only genes identified in both the late gastrula and early larva experiments (49 genes for BMP-upregulated and 45 genes for BMP-downregulated) were used for further analysis and the heatmap below. Heatmaps show expression profiles of selected BMP-upregulated (**b**) and BMP-downregulated (**c**) genes at the late gastrula stage. The BMP-downregulated gene set contains many genes known for their neural expression in model organisms, such as *elav* and *soxb1*. Asterisks highlight *Xenopus* ventral centre and Spemann–Mangold organiser genes. ”

In addition, we have added a Venn diagram to the supplementary to show in greater detail how this gene set was arrived at (Supplementary Fig. 26).

Finally, panels B-D have been moved to the supplement as suggested.

Figure 4, Panel A: Pearson correlation between ctrl and BMP(-) conditions show surprisingly little difference. How can this be explained?

Figure 4, Panel A: What are the different samples that each column and row represents?

Though we agree with the reviewers that on first look it seems that there is surprisingly little difference, there are still stark, observable differences. Each of the treatments forms a separate cluster in the hierarchical clustering, which must reflect genuine differences

between treatments. In addition, while BMP-downregulated genes appear to change little between control and BMP(-) embryos, BMP-upregulated genes show a much greater effect. We also note that, as described in our earlier response to Reviewer 1, comment #13, we did not observe a clear dose-dependent effect in our treatments, likely due to saturation at the lower dose, particularly for the inhibitor treatments. This is reflected in the heatmap, where samples from different doses and inhibitors are intermixed without distinct clustering. We have now added more information about the samples and updated panel a, which has been moved to Supplementary Fig. 25 (Reviewer Fig. 6).

Reviewer Fig. 6 | Pearson's correlation analysis of transcriptomic response to BMP manipulation at the late gastrula stage. The analysis is based on 881 differentially expressed genes (fold-change > 4, $p < 0.001$). Each row and column represents one sample. Pearson's correlation coefficient (r) indicates transcriptomic similarity, with purple denoting the highest similarity and orange the lowest. LG, late gastrula; C, concentration; R1–R3, biological replications. Treatments and units: LDN193189 (LDN), μM ; K02288 (K02), nM; mouse BMP4 (mBMP), ng/mL.

Figure 4, Panel C-D: This could go into supplementary figures.

Panels C and D have been moved to the supplementary figures as suggested.

Figure 5, Panel B: Body axes should be labeled for easier understanding. For *admp* and *bambi* the difference between control and BMP(-) embryos is not very clear. A quantification or at least a better representative image should be provided.

We have added additional arrowheads to indicate the expression patterns, and included a new body axis schematic in Fig. 4e.

Extended Data

Figure 7: This figure is not mentioned in the text.

We thank the reviewers for bringing this oversight to our attention. We have ensured that this figure is now referred to in the text.

Figure 9, Panel A: ISH images are difficult to interpret. While a *bmp2/4* signal is present in the dorsal part of the embryo where pSmad was detected in Figure 1, it also seems to be expressed in other parts of the embryo?

*In situ*s for BMP ligands have now been repeated and are all presented together in Figure 2. As noted above, we do not necessarily expect BMP ligands to be expressed at the dorsal part of the embryo where the pSmad1/5 is detected since evidence from other systems shows that only the location of *chordin* expression is critical to this and BMP expression varies.

Figure 9, Panel B: Where is *bmp2/4* marked in the schematic?

We have elected to remove this subfigure as it was excessively complex and distracting from the main narrative of the manuscript.

Figure 9, Panel C: This requires further investigation. Can this be confirmed with the RNAseq data? Is the effect on *chordin* expression direct or indirect because the ventral domain is lost?

We have elected to remove this subfigure as it was excessively complex and distracting from the main narrative of the manuscript.

Figure 9, Panel D: What experimental evidence is the distinction between ectoderm and endomesoderm based on?

We have elected to remove this subfigure as it was excessively complex and distracting from the main narrative of the manuscript.

Figure 9: Please provide more explanation for the arrows and pathways in the Figure legend.

We have elected to remove this subfigure as it was excessively complex and distracting from the main narrative of the manuscript.

Supplementary Data

Figure 4: Why was the tree only produced for the Smad family? At the very least this should be produced for all major proteins discussed in this study (i.e. BMP family, Chordin, selected neural genes, dorsoventral markers). Also: add description of species abbreviations.

We have now produced trees for all BMP pathway gene families annotated in this study, including BMP ligands, receptors, smads, and signalling modulators like *chordin*, *noggin* and *follistatin*. These are presented as supplementary Figs. 7 to 17. All figure legends contain lists of species abbreviations.

Text

Page 1, Line 30: “We uncover a BMP signaling gradient...” - The presence of a signaling gradient was never actually discussed/shown. For BMP only mRNA expression was shown, and not the distribution of protein. We appreciate the challenges of working with emerging model systems, but this should be clarified in the text. The pSmad antibody could be used as a read-out for a signaling gradient, but this would need to be discussed in more detail.

We thank the reviewer for highlighting this inconsistency and we agree with their remark. We have now edited the text to read “We uncover asymmetrical activation of the BMP signalling pathway...”

We have also carefully reviewed our use of the word gradient throughout the manuscript and modified it in several other places, and also clarified that pSmad1/5 is used as a BMP signalling readout.

Page 2, Line 52: “...In the fruit fly *Drosophila*, the BMP gradients is completely inverted...” - This is true for invertebrates in general.

We have now modified the text to reflect this: “In the fruit fly *Drosophila* and other invertebrates...”

Page 9, Line 244: “...Extended Data Fig. 7...” - This should probably refer to Extended Data Fig. 8?

We apologise for this mistake and thank the reviewer for correcting it.

Page 11, Line 306: “...underscoring a deeply conserved BMP gradient...” - See above. The use of the word gradient should be justified.

The text has been modified to read: “underscoring a deeply conserved dorsal-ventral asymmetry of BMP pathway activation in spiralian...”

Page 11, Line 310: “...we applied [...] inhibitors and [...] proteins for two durations...” - The reasoning for selection of these stages/durations should be explained.

This was also raised by an earlier reviewer and we agree that our logic was insufficiently clear. We have now revised both the Results and the Methods to explain this in further detail.

Page 11, Line 314: “...LDN-193189 (LDN) and K02288 (K02)...” - What is the difference between these two inhibitors? Why were two used?

The two are just independent molecules which both inhibit the BMP signalling pathway by blocking type I BMP receptors. They have similar but distinct binding profiles in the ATP pocket, forming unique hydrogen bonds with different amino acid residues (Sanvitale et al. 2013 *PLoS ONE*). LDN is an optimised derivative of dorsomorphin with fewer off-target effects (Cuny et al. 2008 *Bioorg. Med. Chem. Lett.*) while K02 was identified more recently in

a screen for BMP inhibitors as an even more specific inhibitor of BMP receptors (Sanvitale et al. 2013 *PLoS ONE*).

Our reason for using both of these two independent inhibitors was simply to be an extra control to check that any effects observed were not due to the specific inhibitor but due to a genuine effect on BMP signalling itself. We agree that this logic was not properly communicated in the manuscript and we have now added a sentence to the Methods to explain this:

“Two independent inhibitors were used as controls to confirm that the observed phenotypes resulted from BMP signal perturbation rather than being specific to a particular inhibitor.”

We also added a sentence in the Results section to highlight the consistency of results across both inhibitor molecule treatments:

“Results were consistent across both independent BMP inhibitor molecules (LDN and K02), suggesting that the observed phenotypes were due to BMP signal disruption rather than off-target effects of a specific inhibitor.”

Page 11, Line 316: “...led to a loss of dorsal pSmad1/5...” – From the images presented, it looks like the dorsal signal persists, and additional signal is gained throughout the rest of the embryo.

This sentence was poorly worded, and was intended to mean that the specificity of pSmad1/5 to the dorsal side was lost. We have now rephrased to make it clearer:

“Conversely, treatment with recombinant mouse BMP4 (mBMP4) protein⁸⁷, referred to as BMP(+), expanded nuclear pSmad1/5 across the whole gastrula (Fig. 3f), validating the efficacy of the manipulations.”

Page 11, Line 319: “...altered gastrula morphology...” - The authors should describe how gastrula morphology is altered and what can be concluded for the role of BMP signaling in the process.

We have added a detailed description of the morphological changes, particularly the invagination process. We also noted that strong BMP signalling may promote ingression, but further studies are needed to clarify its role in brachiopod gastrulation.

The text now reads as follows:

“Notably, both BMP(–) and BMP(+) treatments disrupt gastrulation. BMP(–) treatment results in a shortened archenteron (Fig. 3d) or a complete delay in invagination when using a stronger BMP inhibitor (Fig. 3e). In contrast, BMP(+) treatment delays invagination but promotes ingression (Fig. 3f). These observations highlight the importance of the BMP gradient in gastrula development.”

Page 13, Line 371: “Gene ontology (GO) analysis [...] highlighted the ‘nervous system development’...” - How reliable is GO term analysis in this species? Phylogenetic trees should be presented, at least for the main protein families of interest to confirm this.

We used a standard method of orthology assignment for the gene ontology pipeline, employing bidirectional BLAST searches to the Swiss-Prot database from UniProt and maintaining only those with matching reciprocal best hits. To validate the accuracy of this procedure, we have now built trees of the key neural-related protein families Elav, Pax6, Sox1, Sox2. In every case, the trees support the annotation and naming of the protein sequences identified in the genome of *L. anatina* as bona fide orthologues.

Page 17, line 477: “...organiser genes cer1, noggin and shh are BMP-upregulated when they might be expected to be downregulated...” - The authors should provide confirmation of this with in situ staining. This also merits further discussion.

Given the fact we already had to repeat in situ and perform new *in situ*s for many other genes, we were unable to perform additional experiments in this case. In lieu of this, we have added a statement to highlight the uncertainty of this conclusion with the current data:

“These findings, while requiring further validation, suggest that although the central BMP–Chordin axis is conserved, its downstream effects are evolutionarily variable.”

In addition, we have completely re-written the section of the Discussion that focuses on this result to place it better into the context of the paper and published literature.

Page 17, Line 491: “Finally, we investigated how BMP ligand expression related to the BMP signaling gradient...” - This should be shown in Figure 1 or 2 and not the extended data.

This data is now shown in Figure 2 as suggested by the reviewer.

Point-by-point response to the reviewers' comments

Reviewer #1

Lewin et al. have significantly improved the manuscript, and I am happy the changes. There are some textual things, which, in my opinion, should be addressed before the acceptance, but I do not think another revision round is necessary. The data are excellent, and once the authors fix the text a little, the paper should be accepted.

We would like to thank the reviewer again for their careful review of our manuscript on both occasions. We are pleased that they are satisfied with the previous adjustments made and are particularly grateful in this round of review for their highlighting of terminological errors that we, as authors, had missed.

Here are some comments in the order of appearance in the text.

1. Throughout the manuscript, the authors state that both in vertebrates and invertebrates, the neurogenic domain forms on the “low BMP signaling” side of the DV axis and mention the ancestrality of the centralized nervous system in Bilateria as a proven fact. “CNS in low BMP zone” also marked on the root of their tree on Fig. 6. This is a misleading oversimplification. Insects with their ventral CNS do not represent the variety of NS types in ecdysozoans, and not every spiralian has a neat ventral nerve cord like *Platynereis*. Also ambulacrarian deuterostomes do not have a CNS. There are many invertebrate groups with much less centralized – sometimes even diffuse – nervous systems, but a dorsal BMP signaling maximum is clearly an ancestral feature of Bilateria. This automatically means that some neural tissue in such “non-CNS” animals must be forming in the “high BMP” area. See <https://www.sciencedirect.com/science/article/pii/S001216061730475X> and <https://www.biorxiv.org/content/10.1101/2025.06.08.658475v1.full> It is very important that the authors discuss the negative regulation of the potentially pro-neural markers in *Lingula*, however, they also must put this into context. Lewin et al. also need to elaborate on what is known about the embryonic origin of neurons in *Lingula* and the extent of centralization of its nervous system. Does the SoxB2- expressing area of the *Lingula* gastrula really represent neurectoderm? Or do Lewin et al. think that it is neurectoderm because it expresses SoxB1, SoxB2 and Pax6? How centralized is the NS of the *Lingula* larva and adult?

We are grateful to the reviewer for highlighting this oversimplification. We have revised the relevant paragraph in the Discussion to place our findings in a broader comparative context, including recent work by Knabl et al. (2025), as suggested by the reviewer, which was not available at the time of initial submission. We now explicitly highlight the potential link between variation in the effects of BMP signals on neural gene expression and variation in the extent of nervous system centralization. We also acknowledge the suggestion of Knabl et al., the role of BMP signals in the last common ancestor of bilaterians was likely anti-neural.

The added text in the Discussion is as follows:

“An alternative explanation recently suggested by Knabl et al. (2025)¹³¹ is that the primary function of BMP signalling is dorsal–ventral patterning, and the ‘anti-neural’ function observed in vertebrates, arthropods and now brachiopods is merely a ‘side-effect’. If this is the case, BMP signalling may have originally promoted neurogenesis in the cnidarian–bilaterian common ancestor lacking a centralized nervous system¹³¹. In bilaterians, multiple lineages may subsequently have evolved convergent mechanisms that utilise it in the opposite manner to pattern the central nervous system¹³². Differences in the relationship between BMP signalling and neural induction⁵ may therefore reflect general variation in nervous system organization, potentially linked to independent evolution of central nerve cords across bilaterians^{132,133}.”

Furthermore, we also recognise that anatomical data on early neurogenesis in *Lingula* are limited, particularly for embryonic stages. Detailed morphological descriptions are only available for juveniles (Yatsu, 1902) and adults (Temereva and Kuzmina, 2021). These studies indicate that the nervous system of brachiopods comprises two principal ganglia, the supraenteric and subenteric ganglia, located above and below the mouth, respectively, with extensive innervation of the lophophore tentacles. This organisation suggests a degree of anterior ventral neural concentration, although it does not correspond to a highly centralised nerve cord as seen in some other bilaterians.

Although establishing a clear connection between juvenile and early embryonic stages would be valuable, this is currently beyond the scope of our study. However, based on our molecular data, which primarily focus on the late gastrula stage, BMP signalling represses neurogenic gene expression, consistent with patterns reported in other brachiopods. Notably, Martín-Durán et al. (2017) showed that in *Novocrania anomala* and *Terebratalia transversa*, the BMP antagonist *chordin* is restricted to the ventral ectoderm, where it colocalises with the ventral midline neuronal marker *netrin* (*ntn*). This domain is characterised by the absence of nuclear pSmad1/5 activity, indicating locally reduced BMP signalling. Pharmacological inhibition of BMP signalling (DMH1 treatment) led to the expansion of ventral neurogenic markers, including *NK2.1*.

Consistent with these findings, our RNA-seq data from *Lingula* show that BMP suppression results in upregulation of *SoxB1*, *SoxB2* and *Pax6*. We therefore interpret the ventral low-BMP ectoderm as a conserved neurogenic territory in brachiopods, defined by molecular identity and signalling context. Importantly, this interpretation concerns early patterning and pro-neural specification and does not imply the presence of a morphologically centralised nervous system at the gastrula stage as we discussed above.

References:

- Yatsu, N. On the development of *Lingula anatina*. *J. Coll. Sci. Imp. Univ. Tokyo* **17**, 1–112 (1902).
- Martín-Durán, J. M., Passamanek, Y. J., Martindale, M. Q. & Hejnol, A. The developmental basis for the recurrent evolution of deuterostomy and protostomy. *Nat. Ecol. Evol.* **1**, 5 (2017).
- Temereva, E.N., Kuzmina, T.V. The nervous system of the most complex lophophore provides new insights into the evolution of Brachiopoda. *Sci. Rep.* **11**, 16192 (2021)

- Knabl, P. et al. The anti-neural role of BMP signaling is a side effect of its global function in dorsoventral patterning. *bioRxiv* 2025.06.08.658475 (2025).

2. Line 89-90: The authors talk about the “variability of ligand localisation” – please replace with “variability of ligand expression”.

We have now made this replacement.

3. In the new version of the manuscript, the authors replaced “basal deuterostomes” with “early-branching deuterostomes”. Ambulacrarians are as early-branching as the chordates. I would just stick to “ambulacrarian deuterostomes”.

We have now removed all cases of “early branching deuterostomes” from the text and replaced them with “ambulacrarian deuterostomes”.

4. Line 463: Lewin et al. write about “neurulation in spiralian”. Spiralian do not neurulate. Neurulation is a chordate-specific term.

We thank the reviewer for pointing out this mistake. We have replaced “neurulation” with “neural domain formation”.

5. In their discussion of the organiser (Lines 622 and further), the authors raise a question of whether the organiser may have been present outside chordates. The blastopore lip organiser is clearly much older than chordates – it exists in cnidarians. See <https://www.nature.com/articles/ncomms11694> However, molecularly, cnidarian organiser resembles the Nieuwkoop center more than the Spemann-Mangold organiser. If Lingula has a blastopore lip organiser, it would be important to know whether only Chordin-expressing side is capable of axis induction.

We thank the reviewer for highlighting this paper, which we had not cited but is highly relevant to our work. We have now rewritten this part of the Discussion to incorporate these ideas.

Reviewer #2

I appreciate the efforts the authors made to revise the manuscript; my concerns and questions have been adequately addressed. The revised focus on BMP signaling--- description of a BMP ‘seesaw’, separation of d-v axis formation and neural induction as potentially separate events, and a more nuanced description of BMP signaling and putative evolutionary scenarios throughout----has greatly improved the manuscript. I also think the addition of new gene expression data for ‘neural’ genes strengthens the hypothesis that BMP signaling affects neural fate in *L. anatina*. I also appreciate the clarifications throughout the manuscript in terms of methodology, reporting of results, and interpretation of those results. I think this manuscript makes important contributions to our understanding of the function of BMP signaling during development in Spiralia and evolution of BMP signaling across Bilateria and should be published. I have a few minor questions, but I don't need a response to these before publication.

We would like to take this opportunity to once again thank the reviewer for their time and effort in reviewing our manuscript. Their comments were instrumental in many improvements we were able to make to the manuscript, and we are delighted that they are satisfied with the result. We respond to their few remaining comments below.

Is there a chordin-like homolog in addition to chordin *L. anatina*?

We did not identify a chordin-like homolog in the *L. anatina* genome.

Are there distinct elav1 and elav2 homologs in *L. anatina*?

No, *L. anatina* has only one *elav* gene.

I am still not convinced that the neural tissue is increased/decreased after perturbation of BMP signaling in *L. anatina*. The expression of elav after manipulating BMP signaling is not very strong.

We have now rephrased this part of the text to further emphasise the weaker effect on *elav* compared to the other neural genes. The new text reads:

“*Elav* expression was slightly reduced under BMP(+), and showed a slight, though less clear, upregulation with K02 (Fig. 4c). This is consistent with the RNA-seq data, where the effect on *elav* is less pronounced than other neural genes.”

Fig. 2e – How do you know which side is ‘dorsal’ from a lateral view (is the future dorsal side morphologically distinct from the ventral side)?

The reviewer is correct that the dorsal and ventral sides are not morphologically distinguishable at the gastrula stage in lateral view. To clarify axial orientation, we first established the animal–vegetal axis as a geometric reference, defined by the position of the polar bodies at the animal pole and the site of invagination and blastopore formation at the vegetal hemisphere. The dorsal–ventral axis is spatially distinct from, and broadly orthogonal to, this animal–vegetal axis. Although *Lingula* embryos undergo extensive morphogenetic rearrangements from gastrulation to larval stages, the relative orientation of these axes remains consistent. Dorsal–ventral polarity is inferred from the opposing domains of BMP activity, consistent with conserved brachiopod dorsal–ventral patterning described by Martín-Durán et al. (2017), in which the BMP-high domain corresponds to the dorsal side (Fig. 2i–l; Reviewer Fig. 1). In *Lingula*, this dorsal region gives rise to the mantle and shell field, whereas the opposing ventral region forms the future mouth and tentacles. Dorsal and ventral identities were therefore assigned based on embryonic geometry, conserved molecular polarity and developmental fate.

● pSmad1/5 (BMP signalling readout) ● Hoechst (nuclei) ● CellMask (cytoplasmic membrane)

Reviewer Fig. 1 | Embryonic geometry and conserved molecular polarity in *Lingula*. **a**, Immunostaining of pSmad1/5 in an early gastrula embryo showing the BMP activity domain (confocal z-projection). The polar body marks the animal pole. The axis defined by the polar body and blastopore (animal–vegetal axis) is broadly orthogonal to the polarised pSmad1/5 signal, which delineates dorsal–ventral polarity consistent with conserved patterning in protostomes. Scale bar, 50 μm . **a'**, Schematic representation indicating the orientation of the animal–vegetal and dorsal–ventral axes in lateral view.